# A global long-term (1981–2019) daily land surface radiation budget product from AVHRR satellite data using a residual convolutional neural network

Jianglei Xu[1], Shunlin Liang[2], Bo Jiang[3]

[1]School of Remote Sensing and Information Engineering, Wuhan University, Wuhan 430079, China

[2]Department of Geographical Sciences, University of Maryland, College Park, MD 20742, USA

[3]Faculty of Geographical Science, Beijing Normal University, Beijing 100875, China

*Correspondence:* Shunlin Liang (sliang@umd.edu)

**Abstract.** The surface radiation budget, also known as all-wave net radiation ($R_n$), is a key parameter for various land surface processes including hydrological, ecological, agricultural and biogeochemical processes. Satellite data can be effectively used to estimate $R_n$, but existing satellite products have coarse spatial resolutions and limited temporal coverage. In this study, a point-surface matching estimation (PSME) method is proposed to estimate surface $R_n$ using a residual convolutional neural network (RCNN) integrating spatially adjacent information to improve the accuracy of retrievals. A global high-resolution (0.05 °), long-term (1981–2019), and daily mean $R_n$ product was subsequently generated from Advanced Very High-Resolution Radiometer (AVHRR) data. Specifically, the RCNN was employed to establish a nonlinear relationship between globally distributed ground measurements from 522 sites and AVHRR top of atmosphere (TOA) observations. Extended triplet collocation (ETC) technology was applied to address the spatial scale mismatch issue resulting from the low spatial support of ground measurements within the AVHRR footprint by selecting reliable sites for model training. The overall independent validation results show that the generated AVHRR $R_n$ product is highly accurate, with $R^2$, root-mean-square error (RMSE), and bias of 0.84, 26.77 $Wm^{-2}$ (31.54%), and 1.16 $Wm^{-2}$ (1.37%), respectively. Inter-comparisons with three other $R_n$ products, i.e., the 5 km Global Land Surface Satellite (GLASS), the 1 ° Clouds and the Earth's Radiant Energy System (CERES), and the 0.5 ° × 0.625 ° Modern-Era Retrospective analysis for Research and Applications, Version 2 (MERRA2), illustrate that our AVHRR $R_n$ retrievals have the best accuracy under most of the considered surface and atmospheric conditions, especially thick cloud or hazy conditions. However, the performance of the model needs to be further improved for the snow/ice cover surface. The spatiotemporal analyses of these four $R_n$ datasets indicate that the AVHRR $R_n$ product reasonably replicates the spatial pattern and temporal evolution trends of $R_n$ observations. The long-term record (1981-2019) of the AVHRR $R_n$ product shows its value in climate change studies. This dataset is freely available at https://doi.org/10.5281/zenodo.5546316 for 1981-2019 (Xu et al., 2021).

**1 Introduction**

Net radiation ($R_n$), which characterizes the surface radiation budget, is the difference between downward and upward radiation across the shortwave (0.3–3.0 μm) and longwave (3.0–100 μm) spectra. The surface radiation budget links the atmospheric climate system to the land surface. $R_n$ is thus a critical variable for studying Earth–atmosphere interactions, including meteorological, hydrological, and biological processes, and is also responsible for the redistribution of surface available energy (Sellers et al., 1997). Accurate characterization and quantification of spatial-temporal variations in surface $R_n$ are essential for both scientific and industrial applications, such as hydrological modeling and water resource management (Hao et al., 2019). However, because the spatial and temporal dynamics of surface $R_n$ are affected by multiple surface features (e.g., albedo, emissivity, and land surface temperature) and atmospheric parameters (e.g., clouds, aerosols, ozone, and water vapor) (Wang et al., 2015b), existing surface $R_n$ data suffer from large uncertainties (Jia et al., 2016; Jia et al., 2018; Jiang et al., 2018; Yang and Cheng, 2020). Therefore, there is an urgent need for long-term, high-resolution surface $R_n$ dataset to more properly understand its spatial pattern and temporal dynamics (i.e., seasonal and inter-annual variability).

Traditionally, historical $R_n$ and surface radiative components have been measured at ground meteorological stations. These ground-based measurements are widely used to study spatiotemporal variations in regional surface radiation and to evaluate gridded products (Jia et al., 2018; Zhang et al., 2020; Zhang et al., 2015). Nevertheless, the high cost of maintaining radiometers means that stations are sparely distributed, severely hindering our ability to study and understand the spatiotemporal variability of surface $R_n$ at global scale.

Alternatively, reanalysis products provide long-term global surface $R_n$ information (Zhang et al., 2016). The greatest advantage of reanalysis products is their global coverage over a long-term period; however, the large uncertainty and coarse spatial resolution of reanalysis products hinder their applications at regional spatial scale (Jia et al., 2018; Zhang et al., 2016; Zhang et al., 2020).

Retrieving $R_n$ from satellite data is another effective method (Liang et al., 2010; Liang et al., 2019). Currently, satellite-based $R_n$ retrieval methods can be broadly divided into two categories, physical methods based on radiative transfer (RT) and empirical statistical methods. RT-based physical methods are more applicable to a larger spatiotemporal extent because they consider the physical processes of solar radiation from the top of the atmosphere to the Earth's surface (Tang et al., 2019). The look-up-table (LUT) and parameterization methods are two typical physical schemes that are widely used to estimate surface radiation from satellite data. To address the low computational efficiency of the radiative transfer model (RTM), the LUT method was proposed to estimate the surface radiation from satellite top of atmosphere (TOA) observations, which combines the advantages of RTM-based simulations and statistical methods (Wang et al., 2015b, a; Wang et al., 2020b; Cheng and Liang, 2016; Cheng et al., 2017; Huang et al., 2011). This approach relies on several theoretical assumptions in the RTM simulation

process, such as water vapor amounts, aerosol types, plane-parallel homogeneous clouds, horizontal homogeneity and directional properties of the surface (Jiang et al., 2019b), which results in errors in the final radiation estimates (Hao et al., 2018; Cheng et al., 2017; Jiao et al., 2015). The parameterization scheme is another typical physical method that uses various surface and atmospheric parameter data to calculate surface radiation based on simplified RT (Huang et al., 2020; Qin et al., 2012). However, in the calculation of parameterized formulas, errors from each input variable accumulate in the final calculated surface radiation.

Conversely, different from physical methods, empirical statistical methods typically account for spatial-temporal variations in $R_n$ by establishing statistical relationships between satellite measures or sensed variables, including surface and atmospheric variables or TOA observations, and surface radiation measurements (Tang et al., 2017; Huang et al., 2020; Jiang et al., 2015; Bisht and Bras, 2010; Bisht et al., 2005) using linear or non-linear models. Machine learning (ML) has played an important role in the development of empirical statistical methods owing to its strong nonlinear fitting ability (Jiang et al., 2014; Jiang et al., 2016; Chen et al., 2020; Xu et al., 2020). Although statistical methods incorporate very little physical knowledge and have limited ability to expand their coverage, they are still widely employed owing to their low computational cost and easy implementation.

Several well-known global $R_n$ datasets have been generated from satellite data (Table 1), such as the Global Energy and Water Cycle experiment surface radiation budget (GEWEX-SRB) (Pinker and Laszlo, 1992), the Clouds and the Earth's Radiant Energy System (CERES) (Loeb et al., 2018), and the International Satellite Cloud Climatology Project (ISCCP) (Zhang et al., 2004). Although these products have been widely used in various fields, their coarse spatial resolution ($\geq 100$ km) cannot meet the requirements of high-resolution $R_n$ data. A high-resolution (5 km) global $R_n$ product was recently released (Jiang et al., 2018) from the Global Land Surface Satellite (GLASS) product suite (Liang et al., 2020). The GLASS $R_n$ product, available since 2000, was produced from the Moderate Resolution Imaging Spectroradiometer (MODIS) data and reanalysis products. The Fluxcom initiative recently published a gridded product of surface $R_n$ using multiple ML methods to merge energy flux measurements with remote sensing and meteorological data to estimate $R_n$ retrievals, but the product only provides available data on areas covered by vegetation and the dataset only spans 15 years (Jung et al., 2019b). A long-term, high-resolution, and accurate surface $R_n$ dataset is, therefore, still urgently needed to help more clearly understand the long-term spatiotemporal variation of global surface $R_n$.

In this study, deep learning ML methods were explored to produce a long-term, high-resolution surface $R_n$ dataset from AVHRR data. Reviewing recent studies on surface $R_n$ estimation using ML methods, many significant advancements have been made (Jiang et al., 2014; Husi et al., 2020; Wei et al., 2019; Wang et al., 2019); however, clear deviations still exist between satellite-derived estimations and ground-based measurements. Apart from the performance of the ML method itself,

many of these discrepancies are attributed to two aspects: first, the spatial representation mismatch between satellite data and ground-based measurements and, second, the neglect of spatial adjacent effects on surface radiation estimation.

The spatial scale mismatch between surface radiation for domain averages and ground point measurements with insufficient spatial representativeness (Jiang et al., 2019b; Barker and Li, 1997) has attracted attention for a long time in the development of ML (Yuan et al., 2020a) and in the evaluation of gridded products (Huang et al., 2016; Yang, 2020; Román et al., 2009). However, many current studies still use matched point-surface sample datasets to train ML model regardless of the difference in spatial representativeness of matched point-surface data. The triple collocation (TC) technique (Stoffelen, 1998) was considered as an appropriate upscaling approach for the impact of random measurement error on ground-based measurements in comparison to other complicated upscaling methods (e.g., the time stability approach and the block kriging algorithm (Crow et al., 2012)) (Yuan et al., 2020a). Furthermore, an extended triple collocation (ETC) method was proposed by Mccoll et al. (2014) and then applied to the validation activities of the Soil Moisture Active Passive (SMAP) level-2 surface soil moisture (SSM) product (Chen et al., 2017) and satellite surface albedo products (Wu et al., 2019). Therefore, the ETC technology is employed to limit the effect of upscaling errors of ground measurements on the final surface $R_n$ estimates at the satellite footprint scale.

Spatial adjacent effects should also be considered in the development of ML methods. With an increase in spatial resolution, horizontal inhomogeneities in the atmosphere have become increasingly important and reduce accuracy of surface radiation retrievals at higher spatial resolutions, especially in conjunction with high solar and viewing angles (Wyser et al., 2002); the correlation between satellite TOA observations and surface radiation measurements weakens, and surface radiation cannot be accessed directly from satellite TOA data for individual pixels. Convolutional neural networks (CNNs) were initially designed to perform image recognition tasks; they can be readily used to extract various high-level, hierarchical, and abstract spatial pattern features from original multispectral or hyperspectral satellite images (Yuan et al., 2020b; Ball et al., 2017). Using this approach, multiple environmental parameters and their spatially adjacent effects can be accounted for in the estimation of surface $R_n$ (Jiang et al., 2019b). Therefore, CNN represents a promising method for integrating potential spatially adjacent effects in surface radiation estimation (Jiang et al., 2020b; Jiang et al., 2019b).

Several studies have successfully employed CNNs and other deep neural networks to retrieve surface parameters, such as global solar radiation (Jiang et al., 2019a), precipitation (Wu et al., 2020), and land surface temperature (Yin et al., 2020), with varying success rates. However, no study has yet attempted to retrieve global surface $R_n$ using a CNN model. In this study, a residual CNN (RCNN) based point-surface matching estimation method (PSME) is proposed for estimating global land surface $R_n$. Specifically, the RCNN model links ground-based $R_n$ measurements with multiple image blocks of AVHRR TOA observation data, including reflectance in visible channels and brightness temperature in thermal infrared channels, along with

other additional auxiliary variables. These auxiliary variables include angular information, i.e., solar zenith angle (SZA), viewing zenith angle (VZA), and relative azimuth angle (RAA), and daily Modern-Era Retrospective analysis for Research and Applications, Version 2 (MERRA2) $R_n$ data (Jia et al., 2018; Zhang et al., 2016). Before training the RCNN, ETC technology is applied to select reliable sites at a global level to generate representative training samples, making the established statistical relationship more representative of surface $R_n$ variation at the AVHRR footprint scale. After validation and comparison, the best-trained model was subsequently implemented through the proposed PSME scheme to generate a global-scale 39-year daily surface $R_n$ dataset (1981–2019) with 0.05 ° resolution from the AVHRR data and MERRA2 reanalysis.

The remainder of this paper is organized as follows. Section 2 summarizes the characteristics of the data used for the reconstruction of the ETC triplet and PSME method development. Section 3 describes the ETC method for the selection of reliable sites and the PSME process for surface $R_n$ estimation using the RCNN model. The results for selected sites based on the ETC, the performance of the RCNN model and the long-term spatiotemporal variation of the $R_n$ dataset are presented in Sect. 4. The discussion is presented in Sect. 5. The data availability is described in Sect. 6. Finally, the conclusions are presented in Sect. 7.

**Table 1: Summary of available $R_n$ products.**

| Product name | Spatial resolution | Temporal resolution | Period | References |
|---|---|---|---|---|
| Reanalysis | | | | |
| NCEP/CFSR | ~38 km | 6 hours | 1979–2010 | Saha et al. (2010) |
| NASA/MERRA2 | 0.5 °×0.625 ° | 1 hour | 1979–present | Gelaro et al. (2017a) |
| ERA5 | 0.25 °×0.25 ° | 3 hours | 1950–present | Hersbach et al. (2020) |
| JRA55 | ~55 km | 3 hours | 1958–present | Kobayashi et al. (2015) |
| NCEP-DOE R2 | ~200 km | 6 hours | 1979–present | Kanamitsu et al. (2002) |
| Satellite products | | | | |
| CERES-SYN | 1 °×1 ° | 1 hour | 2000–present | Doelling et al. (2016) |
| GEWEX-SRB | 1 °×1 ° | 3 hour | 1983–2007 | Stackhouse Jr et al. (2000) |
| ISCCP-FD | 280 km | 8 days | 1983–2012 | Rossow and Zhang (1995) |
| FLUXCOM | 0.0833 °×0.0833 ° | 8 days | 2001–2015 | Jung et al. (2019a) |
| MODIS-TERRA | 0.05 °×0.05 ° | daily | 2001–2009 | Verma et al. (2016) |
| GLASS-MODIS | 0.05 °×0.05 ° | daily/daytime | 2000–2019 | Jiang et al. (2018) |

## 2 Datasets

### 2.1 Ground measurements

Ground measurements of daily surface $R_n$ were used for the RCNN model development. The in situ measurements used in this study cover the period from 2001 to 2019, and were obtained using various instruments (e.g., Kipp & ZonenCNR-1 and Eppley) at 522 globally distributed stations from 15 observation networks/programs, as shown in Fig. 1. Detailed information about

these observation networks/programs is listed in Table 2, and specific information about these sites is shown in Table S1. These stations are maintained by multiple global and regional observation network organizations, such as Global Fluxnet, the Greenland Climate Network (GCNET) and the Programme for Monitoring of the Greenland Ice Sheet (PROMICE) (Jiang et al., 2018). These stations vary in elevation from 4 to 5,063 m above sea level and are located in areas with different land covers, including forest, grassland, shrubland, wetland, cropland, ice/snow, and urban areas. The collective in situ measurements, therefore, represent an accurate and comprehensive dataset that is capable of accounting for surface $R_n$ spatial-temporal variation on a global scale.

The instruments applied to obtain surface radiation have different uncertainties. To be specific, the operational thermoelectric pyranometers are known for their high-accuracy performance, with a spectral response of 0.3-3.0 μm, a sensitivity of 7-14μVW$^{-1}$ m$^2$, a thermal effect of less than 5%, and an annual stability of 5% (Lu et al., 2011; Jiang et al., 2019b). The Eppley Precision Infrared Radiometers (PIR, 3.5-50 μm) and Kipp & Zonen CG 4 pyrgeometers (4.5-42 μm) are applied to measure the surface radiation with a uncertainty of ± 6% or 15 Wm$^{-2}$ at the 95% confidence level (Philipona et al., 1998). The largest uncertainty for surface radiation measurements are ~2% for pyrheliometers and ~5% for pyranometers (i.e., 15 Wm$^{-2}$), respectively (Augustine et al., 2000). Additionally, the radiation measurements obtained by Kipp & Zonen CNR1 and CNR4 instruments are with an expected accuracy of ±10% for daily totals (Wang and Dickinson, 2013). The radiation observations measured by Kipp & Zonen net radiometers (CNR1, 5-50 μm or CNR1-lite, 4.5-42 μm), are with uncertainty of ~10% at 95% confidence level for daily totals (Yamamoto et al., 2005). Besides, the uncertainties of the shortwave radiation measured by LI-COR Photodiode and $R_n$ observed by REBS Q*7 are about 5 (5-15%) and 10 Wm$^{-2}$ (5-50%), respectively, at monthly time scale (Box and Rinke, 2003; Steffen and Box, 2001). To deal with equipment and operational errors, daily mean surface $R_n$ measurements were calculated based on several strict processing rules successfully applied in previous studies (Jia et al., 2018; Jiang et al., 2014; Chen et al., 2020; Jiang et al., 2018).

To well illustrate the performance of the model in estimating global surface $R_n$, more sites from international observation networks should be determined as the validation sites rather than regional observation networks with similar climate regimes (e.g., ARM) to ensure the independence of the test dataset, which avoids overfitting in model training. In this study, more than 89% of validation sites come from the continental and international networks, including BSRN, FluxNet, CEOP, EOL, AsiFlux, and PROMICE. Additionally, similar and comprehensive surface and atmospheric conditions between the training and validation sites illustrate the good representations of the both training and independent test datasets in global surface $R_n$ variability (Fig. S1), which detects the ability of model in estimating global surface $R_n$. Note that some current regional and international networks are interconnected, for example, some ARM and all BSRN sites are included in the BSRN networks. When determining the training and validation sites, more attention should be paid to these duplicate sites in multiple

observation networks to ensure the independence of the validation sites from the training sites. Finally, as shown in Fig. 1, the

surface $R_n$ measurements from 448 stations were used to train the proposed RCNN model (red circles), while the measurements

from the remaining 75 stations (blue circles) were selected as the independent test dataset to evaluate the model performance.

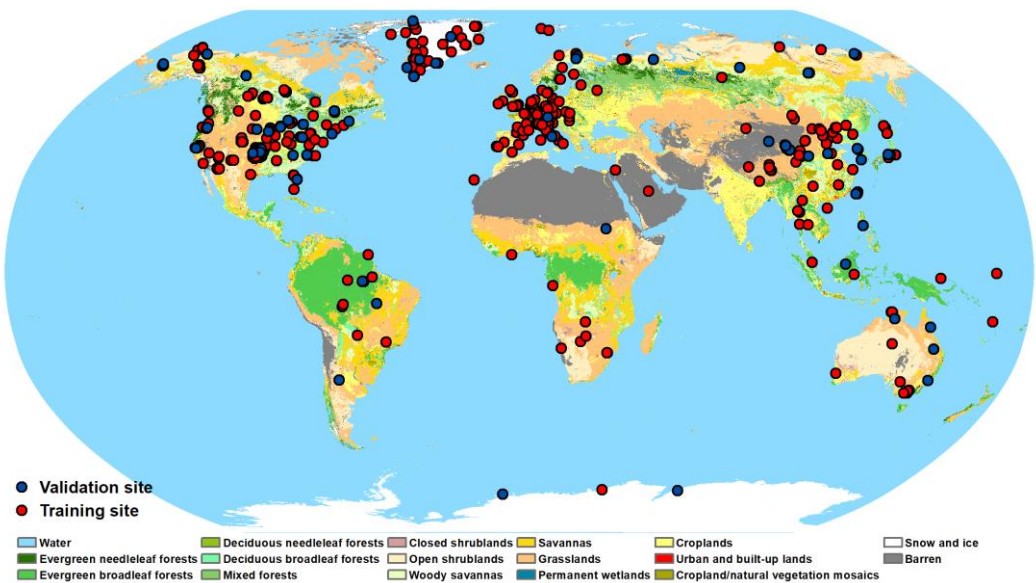

**Validation site**
**Training site**

| Water | Deciduous needleleaf forests | Closed shrublands | Savannas | Croplands | Snow and ice |
| Evergreen needleleaf forests | Deciduous broadleaf forests | Open shrublands | Grasslands | Urban and built-up lands | Barren |
| Evergreen broadleaf forests | Mixed forests | Woody savannas | Permanent wetlands | Cropland/natural vegetation mosaics | |

**Figure 1: Geographic locations of the 522 ground stations used in this study. The 448 stations marked with red circles are used to train the RCNN model, while the 74 other stations marked with blue circles are used for the independent validation of the resulting trained model. Background colors indicate different land cover according to International Geosphere–Biosphere Programme (IGBP) classification system.**

**Table 2: Information about the observation networks. ARM: Atmospheric Radiation Measurement, BSRN: Baseline Surface Radiation Network, CEOP-Int: Coordinated Enhanced Observing Period, CEOP: Coordinated Enhanced Observation Network of China, EOL: Earth Observing Laboratory, GAME.ANN: GEWEX Asian Monsoon Experiment, GCNET: Greenland Climate Network, IMAU-Ktransect: Institute for Marine and Atmospheric Research Ice and Climate, PROMICE: Programme for Monitoring for the Greenland Ice Sheet, SURFRAD: Surface Radiation Budget Network.**

| Network/Program | Instrument | Temporal Interval | Number of sites |
| --- | --- | --- | --- |
| ARM | Kipp&Zonen CNR-1 | 10 minutes | 33 |
| AsiaFlux | Kipp&Zonen CNR-1/EKO MS201 | 30 minutes | 29 |
| BSRN | Kipp&Zonen CG4/Eppley. PIR | 1 minutes | 15 |
| CEOP | Eppley. PIR/EKO MS202 | 30 minutes | 16 |
| CEOP-Int | Kipp&Zonen CG4/Eppley. PIR | 30 minutes | 8 |
| ChinaFlux | Kipp&Zonen CNR-1 | 30/60 minutes | 2 |
| EOL | Kipp&Zonen pyrgeometers, Eppley. PIR | 30/60 minutes | 17 |
| GCNET | Li Cor Photodiode & REBS Q*7 | 60 minutes | 18 |
| GAME.ANN | EKO MS0202F | 30 minutes | 3 |
| Global FluxNet | Kipp&Zonen CNR-1, etc. | 30 minutes | 308 |
| HiWATER | Kipp&Zonen CNR-1/CNR-4 | 10 minutes | 19 |
| IMAU-Ktransect | Kipp&Zonen CNR-1 | 60 minutes | 4 |
| LBA-ECO | Kipp&Zonen CG2/CNR-1 | 30 minutes | 8 |
| PROMICE | Kipp&Zonen CNR-1/CNR-4 | 10 minutes | 24 |
| SURFRAD | Eppley pyrgeometer | 1/3 minutes | 7 |

## 2.2 AVHRR data

AVHRR TOA observations at five spectral channels (a visible band (0.55–0.68 μm), a near-infrared band (0.75–1.1 μm), a middle-infrared band (3.55–3.93 μm), and two thermal bands (10.5–11.3 and 11.5–12.5 μm) were utilized for their comprehensive surface and atmospheric electromagnetic information. The National Aeronautics and Space Administration's (NASA) Land Long-term Data Record (LTDR) project produced a consistent long-term dataset by reprocessing Global Area Coverage (GAC) data, which were obtained from AVHRR sensors onboard the National Oceanic and Atmospheric Administration (NOAA) satellites (Pedelty et al., 2007). The primary reprocessing improvements included radiometric in-flight vicarious calibrations for the visible and near infrared channels, along with inverse navigation to relate a specific Earth location to each sensor's instantaneous field of view (Vermote and Kaufman, 1995; Pedelty et al., 2007). Multiple Climate Modeling Grid (CMG) data from AVHRR and MODIS instruments have been created for land climate studies (Xiao et al., 2017; Pedelty et al., 2007). In this study, we utilized a daily AVHRR TOA data product (AVH02C1) with a resolution of 0.05 ° from 1981 to 2019 to retrieve surface $R_n$ estimates. Additionally, solar/viewing geometry data (i.e., SZA, VZA, and RAA) were also incorporated into the model as the amount of solar radiation incident on the Earth's surface varies greatly under different geometric observation conditions. A summary of these gridded products and their attributes is presented in Table 3.

**Table 3: List of the satellite and reanalysis products used in this study.**

| Product names | Sensors | Spatial resolution | Temporal resolution | References |
|---|---|---|---|---|
| AVH02C1 | AVHRR | 0.05 °×0.05 ° | daily | Pedelty et al. (2007); (Vermote and Kaufman, 1995) |
| GLASS07B11 | MODIS | 0.05 °×0.05 ° | daily | Jiang et al. (2018); (Liang et al., 2020) |
| CERES-SYN | CERES/MODIS | 1 °×1 ° | 1 hour | Doelling et al. (2016); (Doelling et al., 2013) |
| MERRA2 | — | 0.5 °×0.625 ° | 3 hours | Gelaro et al. (2017a) |

## 2.3 GLASS product

The GLASS daily surface $R_n$ product from MODIS data, one part of the GLASS product suite (Liang et al., 2020), was produced using two sets of algorithms. The main algorithm primarily uses the well-documented conversion relationships between downward shortwave radiation and all-wave $R_n$ combinations (Wang and Liang, 2009; Jiang et al., 2015). It also incorporates a combination of other meteorological variables under various environmental conditions, such as different daytime lengths and land cover characteristics, which are designated based on the albedo and normalized difference vegetation index (NDVI). Multiple MARS learners were employed to establish efficient statistical relationships using GLASS downward

shortwave radiation and MERRA2 meteorological variables, allowing land surface $R_n$ to be estimated from these inputs across most spatial domains (Jiang et al., 2016; Jiang et al., 2015). Conversely, when surface solar radiation data were not available, the backup algorithm created a function that separately employed MODIS TOA observations to retrieve surface all-wave $R_n$ using the length ratio of daytime (LRD) classification method, which was accomplished by the genetic algorithm-artificial neural network (GA-ANN) (Chen et al., 2020). By using these two algorithms, the GLASS $R_n$ product can provide seamless global land surface $R_n$ estimates with a 0.05 °resolution. Several studies have used in situ measurements to conduct evaluation studies, illustrating high accuracy performance as well as good application potential (Jiang et al., 2018; Guo et al., 2020). Thus, we used the GLASS daily $R_n$ product covering 2000 to 2018 as a reference to help select reliable sites and validate the results from this study.

## 2.4 CERES-SYN product

The CERES instruments onboard the Terra, Aqua, and Suomi National Polar-Orbiting Partnership (Suomi NPP) satellites observe the TOA global energy budget by measuring shortwave reflected radiation, longwave Earth-emitted radiation and all wavelengths of radiation at a spatial resolution of 20 km at nadir (Wielicki et al., 1996). The CERES Synoptic (CERES-SYN) product combines CERES and MODIS observations with geostationary (GEO) data to provide hourly broadband TOA radiant flux and cloud properties (Doelling et al., 2013). The CERES-SYN product also contains computed TOA and in-atmosphere and surface fluxes based on the Fu-Liou radiation transfer model. Aerosol and atmospheric data were included as inputs to calculate the radiation flux. CERES-SYN fluxes were provided as a 1 °gridded product with an hourly temporal resolution. CERES-SYN surface $R_n$ data have been evaluated in many studies, which indicate that the product has high accuracy, although systematic overestimation exists in the surface net radiation flux data (Jia et al., 2018; Jiang et al., 2016; Jia et al., 2016). Thus, the CERES-SYN surface $R_n$ obtained from four surface radiative components was used as a reference for comparison.

## 2.5 MERRA2 reanalysis

MERRA2, produced by the NASA Global Modeling and Assimilation Office (GMAO), is the latest global atmospheric product and employs satellite observation data from 1980 to the present. The MERRA2 reanalysis assimilates space-based observations of aerosols and represents their interactions with other physical processes in the climate system. The goals of MERRA2 are to provide a regularly gridded, homogeneous record of the global atmosphere, and to incorporate additional climatic variables and conditions, including trace gas constituents (stratospheric ozone), improved land surface representation, and cryospheric processes (Gelaro et al., 2017b). The MERRA2 products have a 0.5 °×0.625 °spatial resolution and hourly temporal resolution. Previous studies (Jiang et al., 2018; Jia et al., 2018; Delgado-Bonal et al., 2020) have confirmed that the MERRA2-calculated surface $R_n$ and its radiative component provide outstanding accuracy and a reasonable spatial-temporal distribution compared

to other reanalysis data. Therefore, MERRA2 $R_n$ data calculated from four surface radiative components were also used in this study to help retrieve accurate high-resolution surface $R_n$ estimates by providing average atmospheric information.

## 3 Methods

The entire workflow of the RCNN-based PSME method is shown in Fig. 2. First, the ETC technology was applied to the triplet system constructed from ground-, satellite-, and model-based $R_n$ data to identify reliable sites at which measurements can well represent the dynamic variation of surface $R_n$ at a 5 km scale. Then, AVHRR TOA reflectance, brightness temperature, angular information and MERRA2 $R_n$ were matched with the ground-based $R_n$ measurements, both spatially and temporally. Specifically, the site-measured $R_n$ data were collocated with the 5 km AVHRR grid product covering the site. If one grid in the AVH02C1 product covered multiple sites, the mean values from these sites' measurements were used to match the grid data. Subsequently, the matched input-output training samples were fed into the RCNN to train the model. Reference $R_n$ measurements taken from reliable sites were used to evaluate the model's performance and, subsequently, identify the best option to produce surface $R_n$ by tuning the hyper-parameters of the RCNN. Finally, surface $R_n$ retrievals were generated using the best-trained model for the global scale, and CERES-SYN and GLASS $R_n$ products were applied to perform inter-comparisons to illustrate the accuracy and spatiotemporal variation of the surface $R_n$ retrievals.

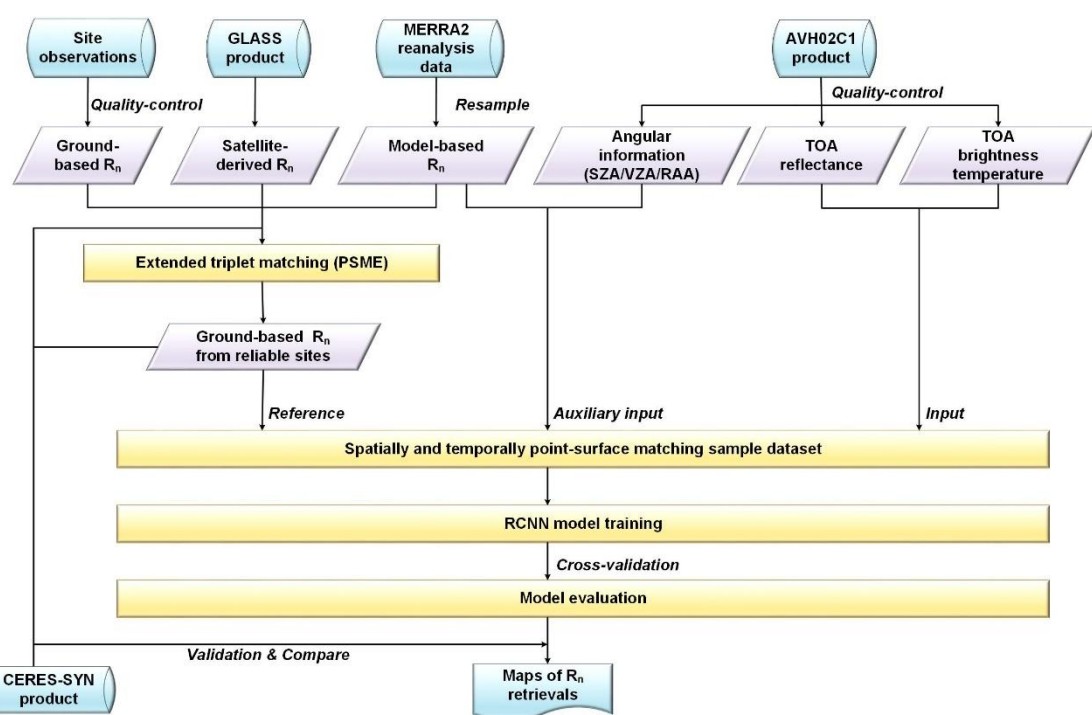

**Figure 2: Workflow of RCNN-based point-surface matching estimation (PSME) method for surface $R_n$ retrievals. TOA: Top of Atmosphere; RCNN: Residual Convolution Neural Network; GLASS: Global Land Surface Satellite; CRES-SYN: Clouds and the Earth's Radiant Energy System Synoptic; MERRA2: Modern-Era Retrospective analysis for Research and Applications, Version 2; SZA: Solar Zenith Angle; VZA: Viewing Zenith Angle; RAA: Relative Azimuth Angle.**

**3.1 Extended triple collocation (ETC)**

To address the spatial scale mismatch issue owing to the small spatial support of sparse ground measurements in comparison to gridded satellite data, which introduces large uncertainty into the collaborative inversion process, triplet collocation (TC) technology was employed (Stoffelen, 1998; Yuan et al., 2020a). Specifically, based on the availability of three collocated, independent measurement systems describing the same geophysical variable, TC was designed to estimate the unknown error standard deviations (or RMSEs) of three mutually independent measurement systems, without treating any one system as

perfectly observed "truth" (Stoffelen, 1998; Gruber et al., 2016). To perform the TC, the following assumptions were made: 1) each of the triplet is related with the unknown truth of the geophysical variable in the linear form; 2) zero cross-correlation across each of the triplet; 3) zero error cross-correlation between the triplet and the true signal state ($T$); and 4) the signal and error statistics are stationary (Chen et al., 2017). Following the first assumption, the independent triplet systems ($X_i$, $X_j$, and $X_k$) are related to the unknown true quantity in a linear error model:

$$X_i = \beta_i + \alpha_i T + \varepsilon_i \tag{1}$$

where $\alpha_i$ and $\beta_i$ are the additive and multiplicative bias terms, respectively; and $\varepsilon_i$ is the mean-zero random error. Similar calibration constants ($\alpha_j$, $\beta_j$, $\alpha_k$, and $\beta_k$) and random error terms ($\varepsilon_j$ and $\varepsilon_k$) are also defined for $X_j$ and $X_k$.

The objective of TC is to find a solution that individually estimates the variance of the random error term ($\varepsilon_i$) for each of the triplet based on the listed assumptions (Stoffelen, 1998; Yuan et al., 2020a). However, to obtain the calibration constants,

one dataset is chosen from the three collocated measurement systems as the reference dataset, and the other two are rescaled into the same reference data space. This results in a dependency of the error variance of the other two datasets on the climatology of the scaling reference (Draper et al., 2013; Yuan et al., 2020a). To deal with this issue, ETC technology was proposed by Mccoll et al. (2014), based on the same assumptions as TC, to estimate an additional evaluation metric independent of the reference dataset, i.e., a correlation coefficient ($\rho_{(T,X_i)}$) of each measurement system with respect to the

unknown target variable as formulated below:

$$\rho(T,X_i) = sign(\pm)\sqrt{\frac{Cov(X_i,X_j)Cov(X_i,X_k)}{Cov(X_i,X_i)Cov(X_j,X_k)}} \tag{2}$$

where $\rho_{(T,X_i)}$ is correct up to the sign ambiguity, as the measurement systems will almost always be positively correlated to the unobserved truth.

Following the ideals of Chen et al. (2017) and Yuan et al. (2020a), ETC was applied to determine the reliable site measurements

over the AVHRR data footprint scale (i.e., 5 km grid). Specifically, the triplet dataset was first constructed using ground-based measurements (i.e., site measurements), satellite-derived retrievals (GLASS $R_n$) and downscaled model-based simulations (MERRA2 $R_n$) depending on the conversion ratio of GLASS $R_n$ between original (0.05 °) and aggregated (0.5 °) spatial resolutions, as they belong to different measurement systems and are not dependent on each other. Then, $\rho_{(T,X_i)}$ was

calculated using Eq. (2) for individual site, illustrating the fraction of 5 km satellite footprint-scale $R_n$ dynamics captured by point-scale ground-based measurements. An appropriate threshold of $\rho_{(T,X_i)}$ was determined to select reliable sites with the greatest representativeness within a 5 km footprint to obtain sample datasets for the model training. After testing a series of thresholds between 0.2 and 0.9 at intervals of 0.1, a threshold of 0.9 was selected, above which the sites were assigned as 'reliable' (also see Sect. 5). Based on these reliable sites, errors of upscaling reliable site-based measurement to a 5-km scale can be weakened to a certain degree due to its better representativeness within the AVHRR footprint.

## 3.2 RCNN-based PSME

### 3.2.1 RCNN

As the spatial resolution of satellite sensors increases, the spatial adjacent effects induced by spatially inhomogeneous atmospheric constitutes (or clouds) fields become more significant, for example, clouds affect the distribution of surface radiation in a region larger than the resolution of an individual pixel. One spatial adjacent effect is the diffusion of radiation that removes part of radiation from an atmospheric column and transfer it to neighboring columns. Two other effects are related to the solar and viewing geometry, such as a shift of the apparent position of clouds and their shadows. Surface $R_n$ is no longer accurately estimated with retrieval algorithms based on the individual pixel approximation (IPA). Comprehensive information within a certain spatial extent centered at reliable sites needs to be considered to help retrieve surface $R_n$.

Loosely inspired by the human visual cortex, CNNs were originally applied to analyze common visual imagery using convolution instead of general matrix multiplication (Ball et al., 2017). CNN model can extract features hierarchically from input multi-channel images using multiple filters. Therefore, the most important feature information regarding reliable site-based $R_n$ measurements can be effectively extracted by CNN within a certain spatial extent rather than on IPA, to help retrieve $R_n$, which weakens the spatial adjacent effects to a certain extent. A general CNN consists of multiple layers of operations, such as convolution, pooling, normalization, and nonlinear activation functions. In the convolutional layers, a series of convolution (*Conv*) operations are performed using convolutional kernel weights and biases on the input images within the receptive field. The result of the locally weighted sum (feature map) is then passed through a nonlinear layer, such as a rectified linear unit (*ReLU*), which increases the nonlinear properties of the decision function and the overall network (Romanuke, 2017). The pooling layer, a form of non-linear down-sampling, merges semantically similar features into one, thereby reducing the amount of computation in the network (Géron, 2019). Additionally, a batch normalization layer is placed between the convolutional layers and nonlinearities to speed up the training of the CNN and reduce the sensitivity to network initialization. By stacking two or three stages of convolution, nonlinearity, and pooling, followed by more fully connected layers, a typical CNN architecture is built.

To improve network performance in complicated tasks, a deeper CNN architecture is needed; however, a deeper neural network is difficult to train well because a degradation problem occurs with deeper networks converge (He et al., 2016). Specifically,

as the network depth increase, the training accuracy becomes saturated and then degrades rapidly. To address the degradation problem, He et al. (2016) proposed a residual block to improve the gradient flow through the network , which enables the training of deeper networks. Residual blocks were employed in our CNN architecture to construct the RCNN. The structure of the RCNN proposed in this study is shown schematically in Fig. 3. Table 4 lists the detailed configurations of the proposed RCNN.

The input signals of the RCNN included AVHRR TOA reflectance and brightness temperature, angular information (SZA, VZA, and RAA), and daily MERRA2 $R_n$ with a spatial size of $15 \times 15$ pixels (these specifications are further discussed in Sect. 5). To avoid introducing new errors, nearest-neighbor interpolation was used to resample MERRA2 $R_n$ to 0.05 ° to match the spatial resolution of the other predictors. The output signal was ground-based $R_n$ from the reliable sites. Essentially, the RCNN model uses a convolution operation taken at stages of the feature map and residual learning block to recognize spatial patterns

centered on a reliable site. Then, the multiple layer perceptron links abstract spatial patterns with ground-based measurements to construct a strong non-linear relationship to reproduce the spatial and temporal variation of surface $R_n$. This approach had been carried out in previous studies (Jiang et al., 2019b; Jiang et al., 2020a; Jiang et al., 2019a).

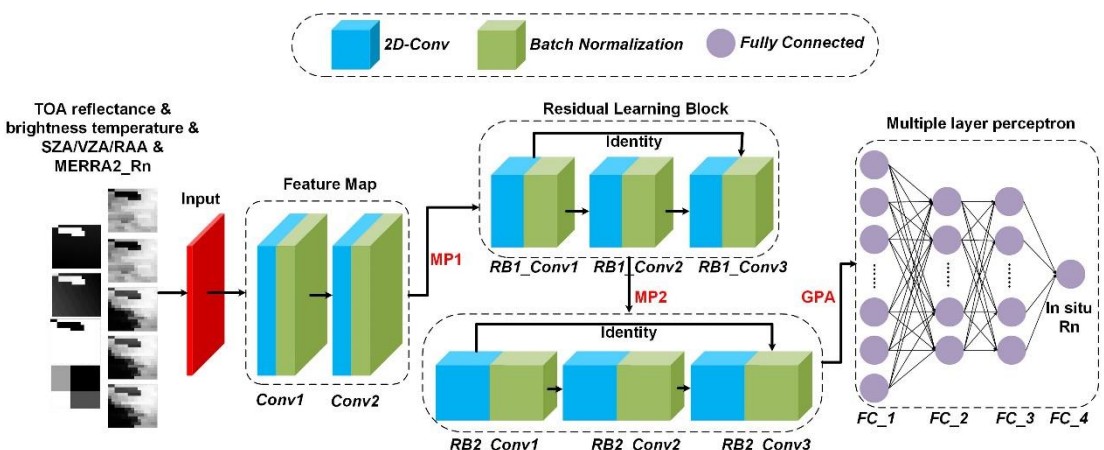

**Figure 3: Depiction of the RCNN network. The model uses AVHRR TOA observations, angular information (SZA, VZA, and RAA), and MERRA-2 $R_n$ as inputs, which are used to calculate daily surface $R_n$ values as output. *Conv* represents the convolution operation; *MP* and *GAP* are the max-pooling and global average-pooling operations, respectively; *RB*: residual block; *FC*: fully-connected layer.**

**Table 4: Detailed configuration of the RCNN. *eLU*: exponential linear unit; *DROP*: dropout layer.**

| Module | Unit | Input size | Kernel num. | Kernel size | Stride | Activation function | Output size |
|--------|------|-----------|-------------|-------------|--------|---------------------|-------------|
| | Input | $9 \times 15 \times 15$ | — | — | — | — | $9 \times 15 \times 15$ |
| Feature | *Conv1* | $9 \times 15 \times 15$ | 32 | $3 \times 3$ | [1, 1] | *ReLU* | $32 \times 15 \times 15$ |
| Mapping | *Conv2* | $32 \times 15 \times 15$ | 32 | $3 \times 3$ | [1, 1] | *ReLU* | $32 \times 15 \times 15$ |
| | *MP1* | $32 \times 15 \times 15$ | — | $2 \times 2$ | [2, 2] | — | $32 \times 7 \times 7$ |

| | | | | | | | |
|---|---|---|---|---|---|---|---|
| Residual | RB1_Conv1 | 32 ×7 ×7 | 64 | 3 ×3 | [1, 1] | ReLU | 64 ×7 ×7 |
| Learning | RB1_Conv2 | 64 ×7 ×7 | 64 | 3 ×3 | [1, 1] | ReLU | 64 ×7 ×7 |
| Block | RB1_Conv3 | 64 ×7 ×7 | 64 | 3 ×3 | [1, 1] | ReLU | 64 ×7 ×7 |
| | MP2 | 64 ×7 ×7 | — | 2 ×2 | [2, 2] | — | 64 ×3 ×3 |
| | RB2_Conv1 | 64 ×3 ×3 | 128 | 3 ×3 | [1, 1] | ReLU | 128 ×3 ×3 |
| | RB2_Conv2 | 128 ×3 ×3 | 128 | 3 ×3 | [1, 1] | ReLU | 128 ×3 ×3 |
| | RB2_Conv3 | 128 ×3 ×3 | 128 | 3 ×3 | [1, 1] | ReLU | 128 ×3 ×3 |
| | GPA | 128 ×3 ×3 | — | — | [0, 0] | — | 128 ×1 ×1 |
| Multiple | FC_1 | 128 | — | — | — | eLU | 128 |
| Layer | DROP | 128 | — | — | — | — | 128 |
| perceptron | FC_2 | 128 | — | — | — | eLU | 64 |
| | FC_3 | 64 | — | — | — | eLU | 64 |
| | FC4 | 64 | — | — | — | eLU | 1 |


**3.2.2 RCNN model training and evaluation**

Sample data from the reliable sites in the training site group (red circles as shown in Fig. 1) were used to train the RCNN model; datasets from reliable sites in the independent validation site group (blue circles shown in Fig. 1) served as test datasets to independently evaluate the model's performance. Specifically, in the training process, 10-fold cross-validation (CV) was used to test the model's predictive power. All of the sample datasets from the reliable training sites were randomly shuffled

and divided into ten groups. One group of these data was then removed as a hold-out or validation dataset and the remaining nine groups of data were treated as the training datasets. The training datasets were used to fit the RCNN model, and the validation datasets were applied to evaluate the trained model's performance to fine-tune the model's parameters. The process was repeated ten times to ensure that each group of data validated the model, and the remaining nine groups of data were trained. Finally, the evaluation results were presented by summarizing and averaging the ten evaluation scores. After

determining the hyper-parameter settings using the CV, the model was trained again using datasets from all the reliable training sites, which was then independently evaluated using the test datasets from the reliable validation sites.

The following five evaluation metrics were used to evaluate the performance of the RCNN model and the $R_n$ retrievals: bias, relative bias (rbias), RMSE, relative RMSE (rRMSE) and the coefficient of determination ($R^2$). Detailed information regarding the application of these metrics can be found in Yang et al. (2018).

**4 Results**

**4.1 Identification of reliable sites**

The number of reliable and unreliable sites for each observation network, identified by a threshold of 0.9 for the ETC-derived correlation coefficient, is listed in Table 5. A total of 262 sites could be considered reliable, accounting for ~50% of the sites.

Furthermore, no site was considered reliable for some observation networks/programs, namely ChinaFlux, GCNET, GAME.ANN, HiWATER, IMAU-Ktransect, and LBA-ECO. It is not surprising that sites from the ChinaFlux network were assigned as unreliable; several studies have revealed that the reliability of site measurements from China is questionable and should be examined carefully before use (Zhang et al., 2015; Tang et al., 2013; Tang et al., 2011). The GCNET network is located in inner Greenland, where systematic measurements errors are common due to difficulties in instrument maintenance and operation-related failures. Sites from the GAME.ANN network are located in the Tibetan Plateau (TP) region, where abnormal climate and complex terrain make it difficult to accurately measure variations in $R_n$. Similar issues also exist in in situ measurements from the IMAU-Ktransect, HiWATER and LBA-ECO networks. In contrast, some of the international observational networks, such as BSRN (Ohmura et al., 1998) and FluxNet (Wilson et al., 2002), provide many ground-based measurements with sufficient spatial representativeness for $R_n$ at 5 km resolution. In addition, the ARM (Stokes and Schwartz, 1994) and SURFRAD (Augustine et al., 2000) networks were classified as containing reliable sites. In situ measurements from the SURFRAD (Augustine et al., 2000) network were well known in surface radiation budget studies because of their high data quality, and have been widely utilized as a result (Wang et al., 2015b; Hao et al., 2019; Wang et al., 2015a; Qin et al., 2012). Overall, compared to other networks, the sites from ARM, BSRN, SURFRAD, and FluxNet networks were mostly identified as reliable sites, illustrating the superiority of these observation networks.

The spatial and proportion distributions of the reliable training and validation sites for different surface types are presented in Fig. 4. The most reliable sites are distributed in the United States, Europe, and East Asia. In turn, many sites located in South America, Africa, Eurasia, and the Polar Regions were identified as unreliable. The reasons that these sites were classified as unreliable are closely related to the complex surface and atmospheric environment and poor instrument maintenance in their corresponding regions. Most grassland and cropland sites were classified as reliable (~66% and ~62%, respectively), whereas the fewest reliable sites were classified in ice/snow-covered areas (~14%). In addition, sites neighboring the sea were mostly identified as unreliability due to the presence of large water bodies within the satellite footprint. Thus, the processing of identifying reliable sites highlights the need to pay more attention to such areas for surface radiation estimations.

**Table 5: Summary of the selected reliable and unreliable sites based on ETC for each observation network**

| Network/Program | Number of reliable sites | Number of unreliable sites |
| --- | --- | --- |
| ARM | 33 | 0 |
| AsiaFlux | 12 | 17 |
| BSRN | 8 | 7 |
| CEOP | 5 | 11 |
| CEOP-Int | 4 | 4 |
| ChinaFlux | 0 | 2 |
| EOL | 2 | 15 |
| GCNEET | 0 | 18 |

| | | |
|---|---|---|
| GAME.ANN | 0 | 3 |
| Global FluxNet | 185 | 123 |
| HiWATER | 0 | 19 |
| IMAU-Ktransect | 0 | 4 |
| LBA-ECO | 0 | 8 |
| PROMICE | 6 | 18 |
| SURFRAD | 7 | 0 |

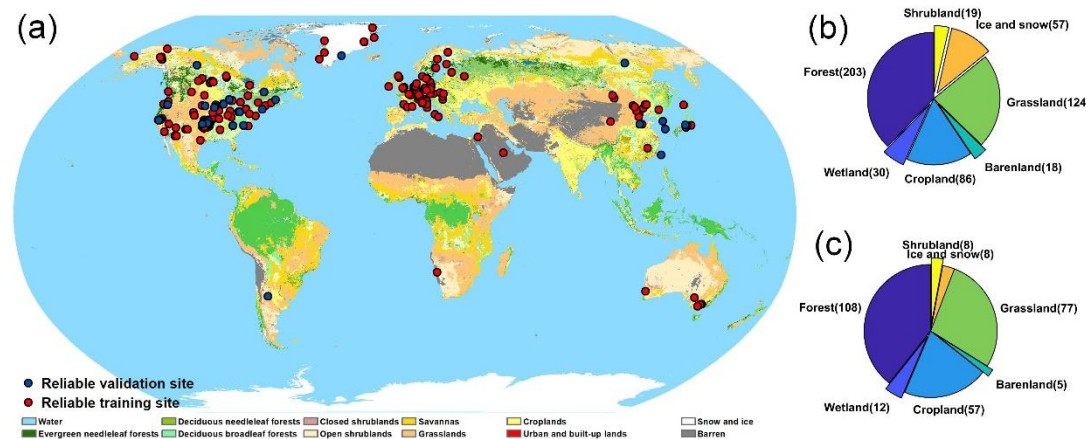

**Figure 4: (a) Spatial distribution of reliable sites and the absolute numbers of (b) all sites and (c) reliable sites under different surface types.**

### 4.2 Assessment of the RCNN model

Ten-CV was used to evaluate the performance of the RCNN model at reliable and all training sites, respectively, and the evaluated performances are summarized in Table 6. Note that the model fitting result represents the model with the best fitting accuracy over the 10 CV rounds, while the cross-validation results are the averages of the 10-round combination. The RCNN model showed a high fitting accuracy at the reliable training sites with $R^2$, RMSE (rRMSE) and bias (rbias) values of 0.90, 20.61 $Wm^{-2}$ (25.71%) and 0.42 $Wm^{-2}$ (0.53%), respectively. Compared to the model fitting accuracy across all sites, the result for the reliable sites was improved, with $R^2$ values increased by 0.04 and rRMSE values reduced by 8.24%. The implementation of ETC for the selection of reliable sites ensures more consistent spatial representativeness of ground-based measurements and AVHRR data, which improves the accuracy of $R_n$ retrievals. Indeed, the CV-derived average accuracy is extremely similar to the model fitting accuracy, illustrating that the trained RCNN model is highly robust. Additionally, an unbiased estimation was achieved by the RCNN model with CV-derived biases close to zero.

Figure 5 shows the overall training accuracy and test accuracy for the RCNN model at reliable training and independent validation sites. The over-training accuracy of the RCNN model is close to that of the CV-derived result. Between the model training to the test phase, the $R^2$ score dropped by 0.06 and RMSE increased by 5.93 $Wm^{-2}$ (5.96%), which indicates that slight over-fitting by the proposed model. However, with the highest cross-validated $R^2$ of 0.90, and the lowest RMSE of 21.01 $Wm^{-}$

[2], the RCNN model trained using ground-based $R_n$ measurements obtained at the reliable sites is considered as the best-trained model, which was selected for the subsequent analysis.


**Table 6: Ten-fold cross-validation performance of the RCNN models.**

| Used sites | Model fitting | | | | | Cross-validation | | | | |
|---|---|---|---|---|---|---|---|---|---|---|
| | $R^2$ | RMSE | rRMSE | bias | rbias | $R^2$ | RMSE | rRMSE | bias | rbias |
| All sites | 0.86 | 25.79 | 33.95 | -0.05 | -0.07 | 0.86 | 25.86 | 33.97 | -0.31 | -0.41 |
| Reliable sites | 0.90 | 20.61 | 25.71 | 0.42 | 0.53 | 0.90 | 21.01 | 26.18 | 0.42 | 0.52 |

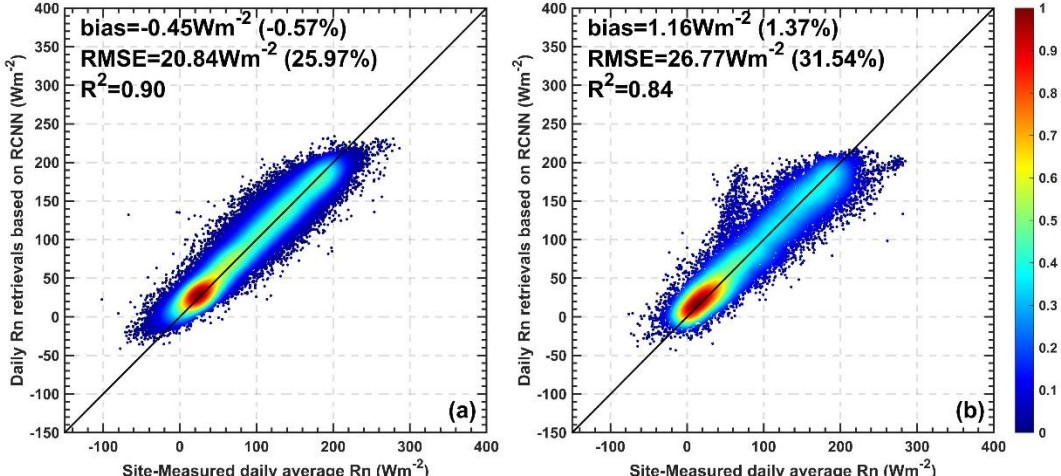

**Figure 5: Scatterplots of (a) mode training (fitting) accuracy and (b) model test accuracy for the reliable training and independent**
**validation sites. The color bar illustrates the normalized density of samples.**

### 4.3 Evaluation of the RCNN-based AVHRR $R_n$ retrievals

#### 4.3.1 Inter-comparisons of $R_n$ products

Figure 6 shows the validation results of the four datasets at the reliable sites, including the AVHRR, CERES-SYN, MERRA2,
and GLASS $R_n$ estimates. Comparatively, the RCNN-derived AVHRR $R_n$ retrievals show the best performance with $R^2$, RMSE, and bias of 0.90, 21.08 Wm$^{-2}$ (26.22%), -0.38 Wm$^{-2}$ (-0.47%), respectively, followed by the GLASS $R_n$ estimates with corresponding values of 0.89, 22.47 Wm$^{-2}$ (27.95%) and -2.96 Wm$^{-2}$ (-3.68%), respectively. The CERES $R_n$ estimates show a notable overestimation at higher values against the in situ measurements with a bias of 7.25 Wm$^{-2}$ (9.04%). In addition, a greater uncertainty exists in the CERES $R_n$ compared to the AVHRR and GLASS $R_n$ estimates, with an RMSE of 25.11 Wm$^{-}$
$^2$. The CERES-SYN cloud product, an input for the calculation of flux products, underestimated low-level clouds (by 11.8% and 20.9% for day and night, respectively) over the sun-glint ocean and polar regions both during daytime and nighttime (Xi et al., 2018; Xu et al., 2020). Additionally, when the aerosol optical depths (AODs) used to calculate CERES-SYN surface solar radiation are compared with the ground-based observations, the calculated shortwave radiation is 1–2% higher (Fillmore et al., 2021). Therefore, large uncertainties in these atmospheric input parameters may lead to serious overestimations of the

CERES-SYN $R_n$. In addition, the MERRA2 $R_n$ shows the lowest accuracy, with an RMSE of 30.88 Wm⁻², reflecting the reanalysis model's inability to accurately describe the evolution of cloud properties (Betts et al., 2006). Comparatively, the AVHRR retrievals show a better accuracy than the other three products. In addition, our estimates have a higher spatial resolution compared to the CERES-SYN and MERRA2 data.

Compared to the validation results at the reliable sites, the accuracy evaluation at all sites shows the ability of the RCNN to
accurately capture $R_n$ variation at a global scale, even though some measurements from unreliable sites added large uncertainties to the final evaluation. Figure S2 shows a comparison of results for the four datasets at all of the sites. Overall, the AVHRR and GLASS $R_n$ retrievals were still better than those of CERES-SYN and MERRA2; however, the accuracy of AVHRR $R_n$ decreases slightly, with $R^2$, RMSE, and bias values of 0.85, 26.74 Wm⁻² (35.70%) and 1.20 Wm⁻² (1.60%), which is comparable to the GLASS $R_n$ retrievals, with values of 0.85, 26.79 Wm⁻² (35.77%) and -0.82 Wm⁻² (-1.10%), respectively.
Therefore, even if the RCNN model is trained using measurements from less reliable sites, it still accurately reproduces surface $R_n$ distributions at global scale. In the following analysis, the GLASS $R_n$ retrievals were used as the main comparison because of their high accuracy and reasonable spatiotemporal variation (Jia et al., 2018; Jiang et al., 2018).

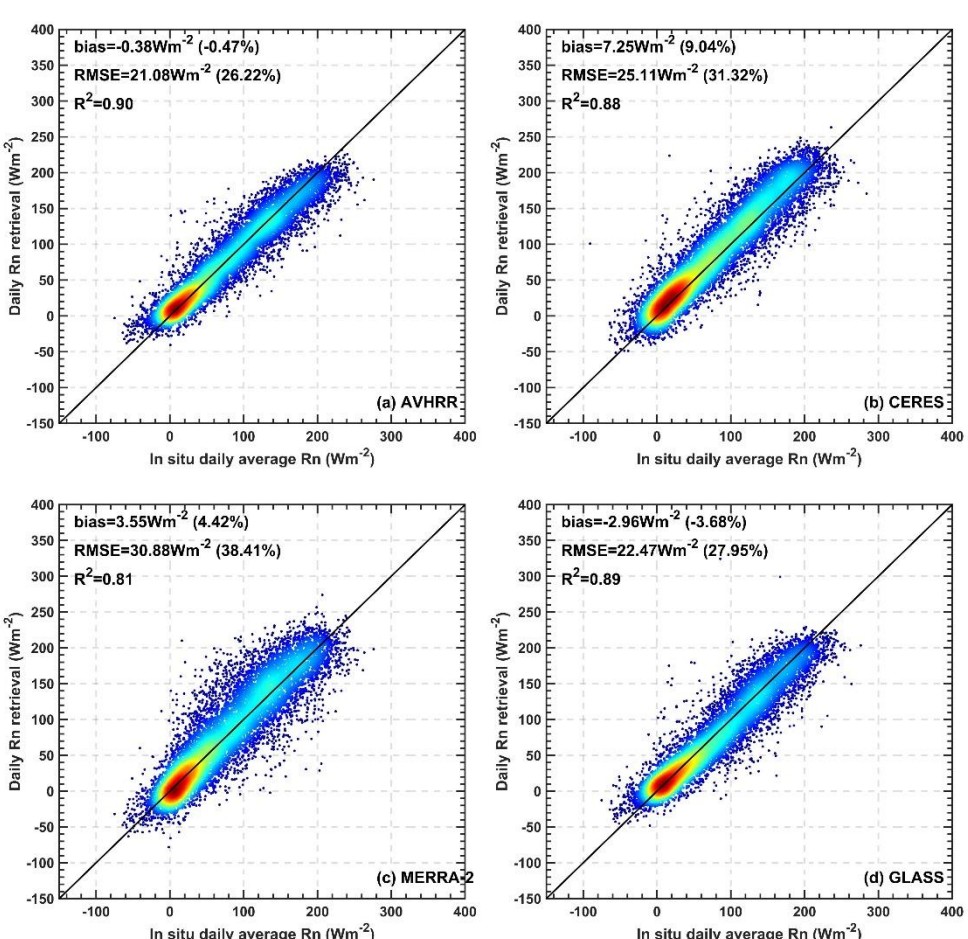

**Figure 6: Scatterplots of product validation for (a) AVHRR, (b) CERES-SYN, (c) MERRA2, and (d) GLASS at the reliable sites.**

Inter-comparison results for the AVHRR and GLASS $R_n$ retrievals against the ground-based measurements over each network are displayed in Fig. 7. The AVHRR $R_n$ retrievals performed slightly better than the GLASS $R_n$ retrievals in most of the observation networks. Specifically, the AVHRR $R_n$ retrievals show lower RMSE values over seven networks, except for the EOL and PROMICE networks. However, the RMSE differences over the EOL and PROMICE networks are very small—only 1.41 and 0.22 $Wm^{-2}$, respectively. The EOL is a small regional network, and its measurements thus only reflect local-scale $R_n$ variation. However, only two out of 17 sites were identified as reliable for the model training and, therefore, the RCNN model cannot capture specific $R_n$ dynamics within such a small spatial extent. Similar reasons also account for the poor performance at the PROMICE network because most of the sites in the GCNET and PROMICE networks are identified as unreliable sites. Thus, the RCNN model has less knowledge of $R_n$ dynamics for snow and ice surfaces. The most significant difference for RMSE was observed over the ARM network, for which the mean RMSE value decreased by 2.0 $Wm^{-2}$ for the AVHRR $R_n$ retrievals relative to the GLASS $R_n$ retrievals. Additionally, these two datasets showed very similar performance based on their $R^2$ values.

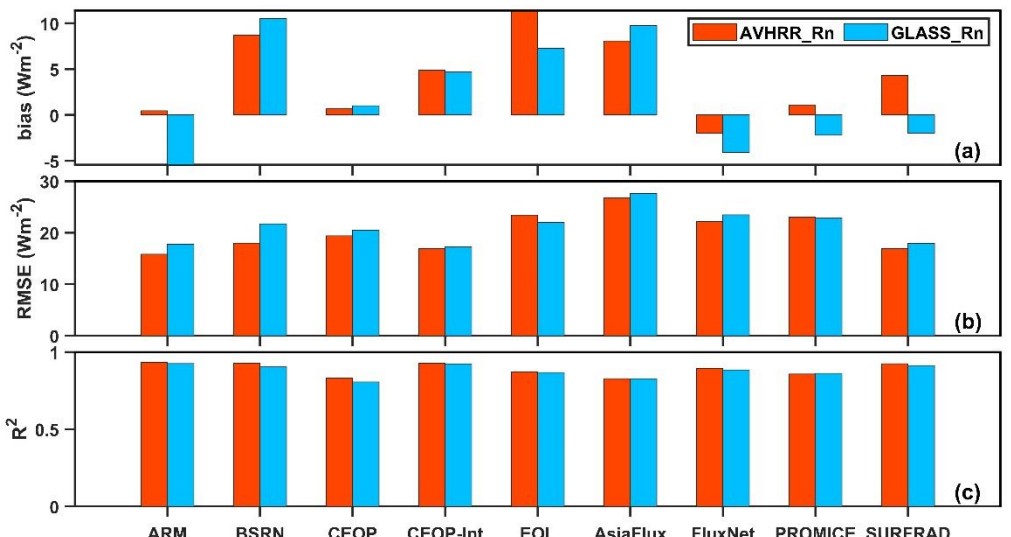

**Figure 7: The average performance of the AVHRR and GLASS $R_n$ retrievals against ground-based measurements at the reliable sites over each network.**

To further improve understanding of the temporal variations of the AVHRR $R_n$ retrievals, coincident time series from all the $R_n$ datasets were inter-compared over seven sites representing different surface types, as shown in Fig. 8. Overall, all four datasets broadly captured the true dynamics of $R_n$ under the different surface types. Comparatively, the AVHRR and GLASS $R_n$ retrievals are more consistent with in situ measurements than the CERES-SYN and MERRA2 products. Specifically, the MERRA2 and CERES-SYN $R_n$ retrievals show higher values compared to the in situ measurements at the BSRN_DRA site, especially during 140–200 day period. In comparison, the AVHRR and GLASS $R_n$ values closely match the ground-based measurements, and thereby better reflect the true temporal variation in $R_n$. At the Lath_UK-AMo site, four of the datasets slightly overestimated during the summer; however, the AVHRR and GLASS $R_n$ retrievals still performed best. Moreover,

large discrepancies occurred at the PM-SCO_U site under the snow/ice surface for the four datasets. Notably, MERRA2 $R_n$ values do not reflect true variations for snow and ice surfaces, especially during the 150–250 day period. Comparatively, the satellite-derived retrievals better capture $R_n$ dynamics, although the CERES-SYN product still exhibits overestimation. Because less reliable sites were screened at the global scale to train the RCNN model under the snow/ice surfaces, the AVHRR $R_n$ values only capture the general $R_n$ trend in the snow/ice areas. The GLASS $R_n$ retrievals are also most consistent with the in situ measurements among the four datasets, although small biases still exist.

The overall evaluation results of the AVHRR and GLASS $R_n$ retrievals for different surface types are displayed in Table 7. Generally, both datasets achieved high accuracy, with RMSEs ranging from 20 to 25 Wm$^{-2}$. The AVHRR $R_n$ retrievals show the better performance for most of the surface types, except for snow/ice, as previously discussed; however, the difference in the RMSEs between the AVHRR and GLASS $R_n$ retrievals for the snow/ice cover type is small (0.49 W/m$^{-2}$). Together, these results further indicate that the RCNN model can generate accurate $R_n$ estimates for different land cover types.

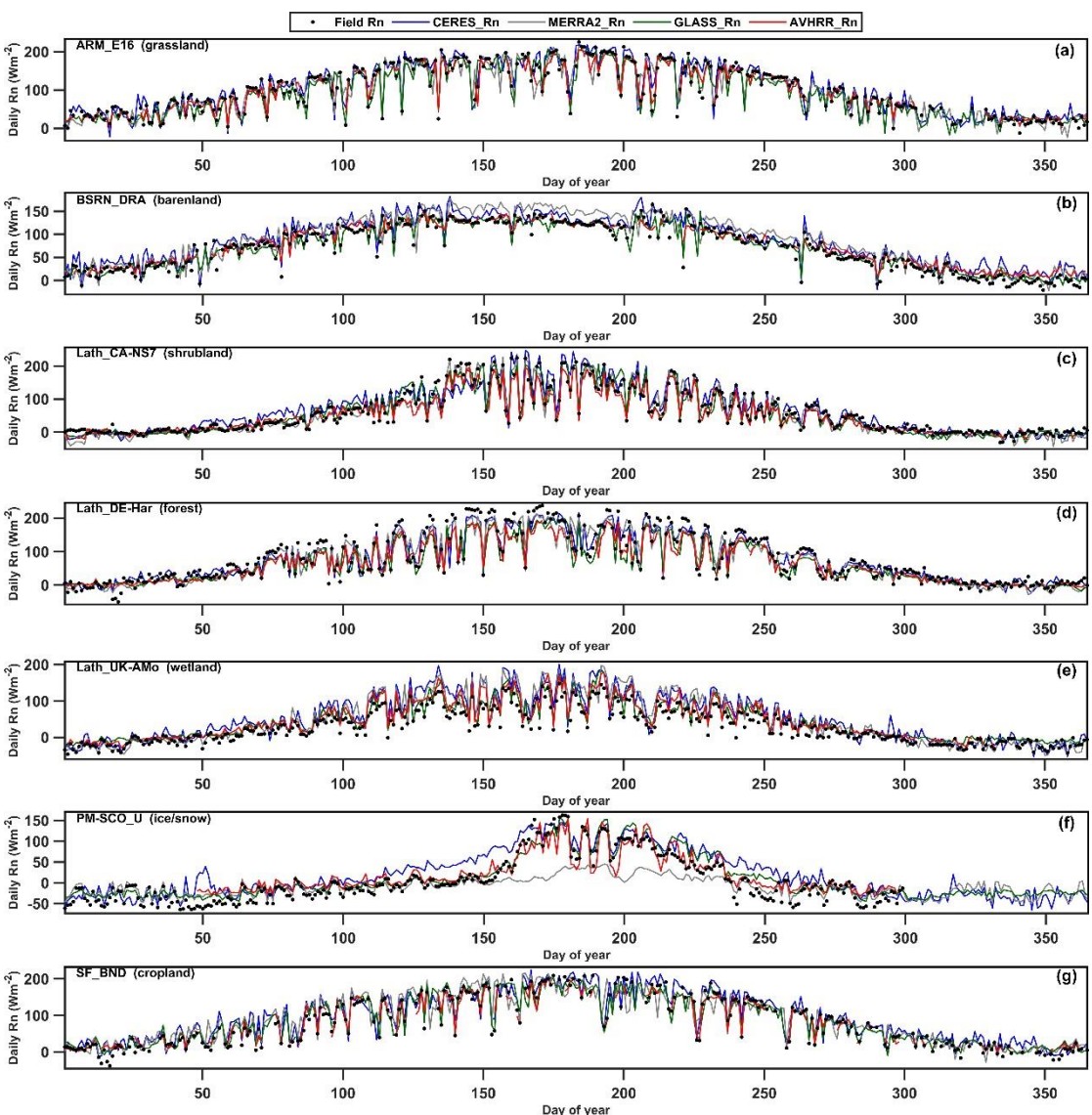

**Figure 8: Coincident time series of the AVHRR, GLASS, MERRA2, CERES-SYN $R_n$ retrievals and ground-based measurements over seven sites representing different surface cover types for (a) ARM_E06 (38.061 °, -99.134 °), (b) BSRN_DRA (36.626 °, -116.018 °,**

 **(c) Lath_CA-NS7 (56.635 °, -99.948 °), (d) Lath_DE-Har (47.934 °, 7.601 °), (e) Lath_UK-AMo (55.791 °, -3.238 °), (f) PM-SCO_U (72.393 °, -27.233 °) and (g) SF_BND (40.050 °, -88.37 °).**

**Table 7: Evaluation of the AVHRR and GLASS $R_n$ retrievals for different surface cover types.**

| Surface types | AVHRR $R_n$ | | | | | GLASS $R_n$ | | | | |
|---|---|---|---|---|---|---|---|---|---|---|
| | $R^2$ | RMSE | rRMSE | bias | rbias | $R^2$ | RMSE | rRMSE | bias | rbias |
| Forest | 0.90 | 22.39 | 27.11 | -4.20 | -5.09 | 0.89 | 23.76 | 28.75 | -5.22 | -6.32 |
| Crop | 0.90 | 20.96 | 26.56 | 3.03 | 3.84 | 0.89 | 22.12 | 28.02 | -1.37 | -1.73 |
| Grass | 0.91 | 18.76 | 22.80 | 4.06 | 4.93 | 0.90 | 20.48 | 24.88 | -0.34 | -0.41 |
| Shrub | 0.90 | 18.52 | 21.93 | -0.88 | -1.04 | 0.89 | 20.09 | 23.78 | -2.03 | -2.40 |
| Ice/snow | 0.85 | 24.46 | 76.37 | 1.84 | 5.75 | 0.86 | 23.97 | 74.84 | -0.59 | -1.84 |
| Barren | 0.91 | 14.46 | 19.84 | 2.70 | 3.70 | 0.84 | 19.27 | 26.43 | 3.37 | 4.62 |
| Wetland | 0.92 | 22.86 | 27.42 | -6.91 | -8.28 | 0.91 | 23.86 | 28.62 | -7.84 | -9.41 |

### 4.3.2 Analysis of influencing factors

Variation in surface $R_n$ is mainly affected by atmospheric conditions, but also, to a lesser degree, by surface characteristics (He et al., 2015). Under clear-sky conditions, AOD and column water vapor (CWV) are the main atmospheric constituents that modulate surface shortwave and longwave radiation, and further affect spatiotemporal variations of surface $R_n$. In contrast, clouds and CWV control surface $R_n$ dynamics under cloud-sky conditions, especially clouds that have significant impacts on shortwave and longwave radiation. Therefore, AOD, CWV, and cloud optical thickness (COT, as a surrogate for cloud optical properties) derived from MERRA2 were employed to analyze the sensitivity of the accuracy of the AVHRR and GLASS $R_n$ retrievals to variations in these influencing factors. In addition, $R_n$ retrieval performance at different elevations was also evaluated.

All the evaluation results are displayed in Figure 9. The AVHRR $R_n$ retrievals were always better than the GLASS $R_n$ retrievals under all conditions of the four influencing factors, except for elevations ranging 800–1000 and 1200–1500 m, which demonstrates the superiority of our algorithm. Specifically, as the COT increases (i.e., increasing cloud thickness), the AVHRR and GLASS $R_n$ RMSE values increase accordingly but still remain relatively low for both datasets (< 27 Wm$^{-2}$). Note that the differences in RMSE between the two datasets also increased with increasing COT (Fig, 9(a)). A small COT indicates relative clear-sky conditions, which results in surface total solar radiation dominated by direct solar radiation. Therefore, the performance of the RCNN model and the MARS models used for the GLASS $R_n$ product (Jiang et al., 2016) is comparable with regard to the accuracy of their $R_n$ retrievals. However, when the absorption and scattering effects are enhanced for direct solar radiation from TOA, depending on the IPA, it is difficult to retrieve the total surface $R_n$ accurately using the MARS model because the spatially adjacent effects (i.e., 3-D effects from clouds) are not considered. Although the RMSEs of the AVHRR retrievals also increase, the rate of increase is lower than that of the GLASS $R_n$ retrievals. This is because the RCNN

model recognizes spatial textural and contextual information and comprehensively considers atmospheric conditions within a certain area rather than on IPA, which to some extent addresses the spatially adjacent effect on the accuracy of AVHRR $R_n$ retrievals.

Aerosols also have absorption and scattering influences on solar radiation, and therefore, a similar conclusion can be drawn. For example, when AOD increases from 0.3 (a non-clean atmosphere), the difference in the performance of the two retrieved model becomes more pronounced. The AVHRR $R_n$ retrievals maintain a stable level of uncertainty (RMSEs = 22–24 $Wm^{-2}$), while the errors GLASS $R_n$ retrieval errors increase dramatically (up to 29 $Wm^{-2}$). This illustrates the importance of integrating the spatial adjacent effect into the inversion model under hazy atmospheric conditions.

In the case of CWV, which has a strong influence on longwave radiation, when the condition is $< 50$ $kgm^{-2}$, the accuracy differences between the AVHRR and GLASS $R_n$ values are small ($< 1$ $Wm^{-2}$). However, as CWV increases, the RMSEs of the GLASS $R_n$ retrievals increase dramatically, while the AVHRR $R_n$ estimates maintain a high level of accuracy (RMSEs = ~23 $Wm^{-2}$).

With respect to elevation, there was no notable difference between the two datasets; our estimates were better than GLASS $R_n$ under different elevation ranges, except for the 800–1,000 and 1,200–1,500 m bins, although these differences were less than 1 $Wm^{-2}$. The lower accuracy of AVHRR $R_n$ values for these two elevation ranges is attributable to the less reliable sites used for the RCNN training. In addition, the AVHRR $R_n$ retrievals show steady and very low (close to zero) biases under different conditions, while the biases of the GLASS $R_n$ retrievals show a high degree of variation. This illustrates that the RCNN model has a greater capability for unbiased surface $R_n$ estimation.

Overall, the RCNN-derived $R_n$ retrievals show a high accuracy under different atmospheric and surface conditions relative to the GLASS $R_n$ retrievals and especially for thick-cloud and hazy atmospheric conditions. In such cases, spatially adjacent information is important for accurately estimating the surface $R_n$ retrievals. Although previous studies have proposed several methods for integrating spatial information to retrieve surface and atmospheric variables, such as $PM_{2.5}$ (Li et al., 2020b; Wang et al., 2020a), ozone (Li and Cheng, 2021), and nitrogen dioxide (Li et al., 2020a), these methods only considered discrete surrounding points within a certain area to train the model using IPA. This artificially destroys the natural correlation between the target and the surroundings. Our RCNN model can automatically recognize complete spatial information centered at an interesting location and, thus, is a more reasonable and effective method.

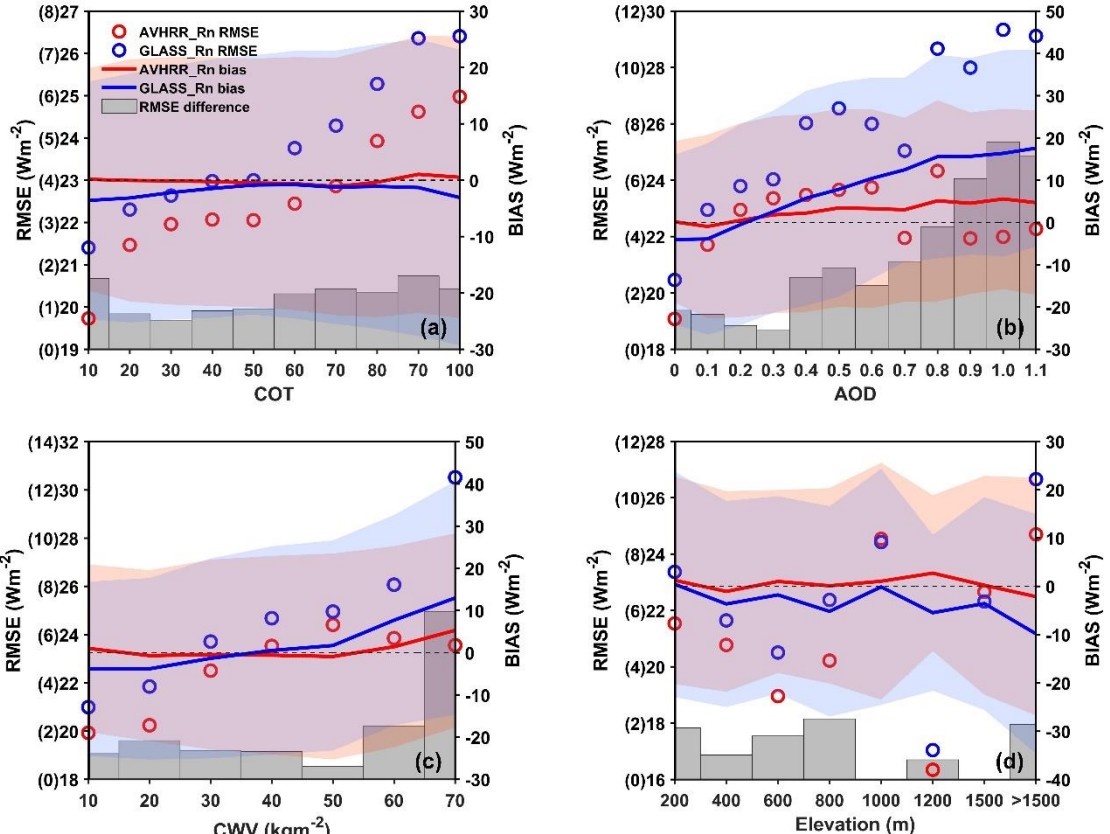

Figure 9: Accuracy changes in AVHRR and GLASS $R_n$ retrievals under different conditions for (a) cloud optical thickness (COT), (b) aerosol optical depth (AOD), (c) column water vapor (CWV), and (d) elevation. The values in parentheses on the left-axis correspond to the RMSE differences denoted by bar charts. The shaded area show the variance ranges of the biases.

### 4.3.3 Spatiotemporal analysis

The global-scale spatial distributions of mean AVHRR and GLASS surface $R_n$ for January and July 2008 are displayed in Fig. 10. The missing values in the Polar Regions reflects the unavailability of valid data at the five bands in the case of the AVH02C1 product. The overall distribution of surface $R_n$ for the two datasets is very similar, although slight differences exist in some regions, such as the TP region, the Sahara Desert, and Greenland. AVHRR $R_n$ retrievals are notably larger than the GLASS $R_n$ retrievals in the TP region. Based on the results shown in Fig. 9(d), greater confidence can be placed in the AVHRR $R_n$ retrievals for high-elevation regions relative to GLASS $R_n$ retrievals; however, the AVHRR $R_n$ retrievals are relatively lower in Greenland. Because few sites from the GCNET and PROMICES networks were classified as reliable for the model training, the RCNN model has less knowledge about the spatiotemporal variations of $R_n$ in Greenland compared to other regions. The validation results in Fig. 8 and Table 7 for the ice/snow surface cover type further confirm that GLASS $R_n$ product may offer a better performance in Greenland region. Therefore, new algorithms and data are required for the Polar Regions to address this problem.

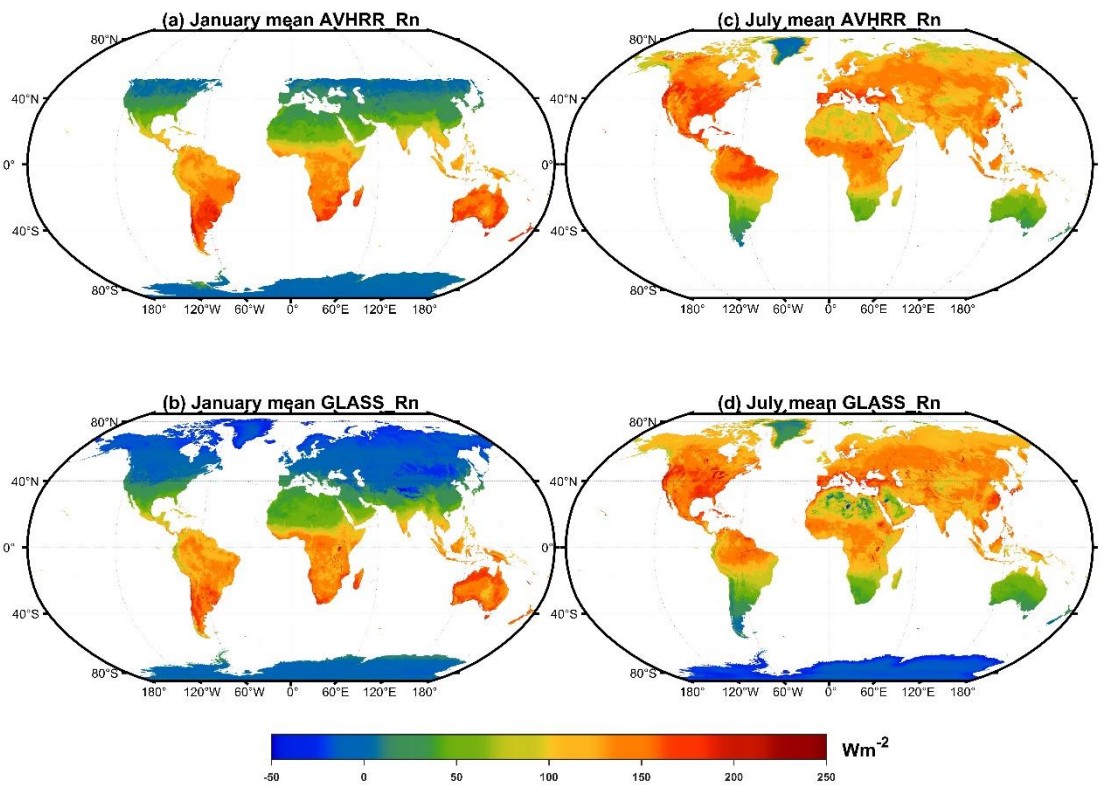

**Figure 10: Spatial distribution of monthly mean AVHRR and GLASS $R_n$ retrievals in January (a, b) and July (c, d) 2008.**

The spatiotemporal consistency of the AVHRR and GLASS daily $R_n$ retrievals against COT was examined at a global scale in January and July 2008, respectively, as shown in Fig. 11. Overall, the spatial consistency is high for the two datasets. Specifically, in January, as the COT increases, the daily mean $R_n$ values and the absolute differences between the two datasets also increase. As shown in Fig. 9(a), when COT increases, the AVHRR $R_n$ retrievals are more accurate. Thus we believe that the large discrepancies under high COT conditions are mainly attributed to the uncertainty of GLASS $R_n$ retrievals. In July,

surface daily mean $R_n$ remained relatively stable under different COT conditions, and the absolute differences between the two datasets also remain steady, with a mean absolute difference of about 20 $Wm^{-2}$.

Based on the previous analysis, spatially adjacent information is important for surface $R_n$ estimation when COTs values are large; however, if the cover of the entire cloud layer is small compared to the scale of the AVHRR footprint, the spatial adjacent effects will be significantly weakened in the inversion process, even if the corresponding COT is large. IPA-provided

information includes the properties of the entire cloud layer. Figure S3 shows the spatial distribution of the monthly mean cloud cover fraction (CF) at the global scale in January and July, and the corresponding differences in CFs (Jan.-July). In January, the CFs are higher than in July over most land regions except in Central Africa, Southern Asia, Southern Australia and Antarctica. However, most regions had smaller CFs in July. The differences in CFs for the two months are also marked; the positive differences demonstrate that more than 72% of the land pixels had a higher CFs in January than in July. The spatial

adjacent effects induced by clouds are more significant on surface $R_n$ in January than in July. Therefore, when large and thick

cloud layers exist, such as in the Polar Regions, CNN is a better choice for surface $R_n$ estimation, especially for downward longwave radiation (DLR) because the temperature of cloud-base, which is an essential variable in the parameterized calculation of DLR, is difficult to retrieve from multispectral remote sensing (Yang and Cheng, 2020).

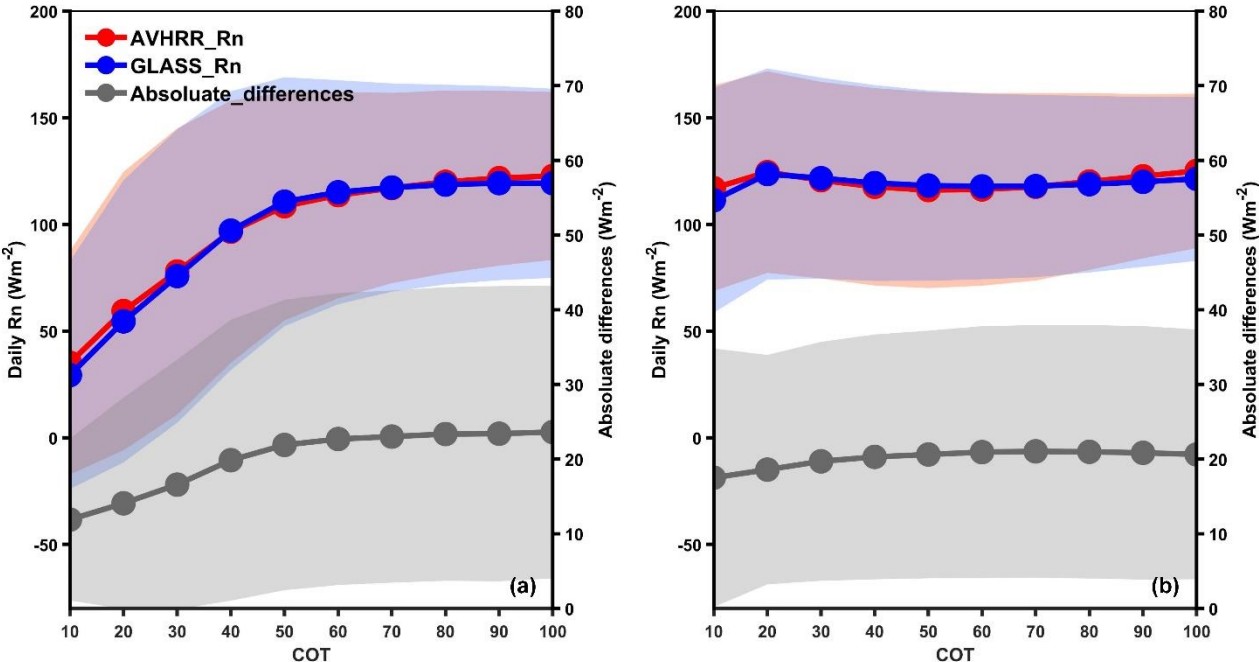

**Figure 11: Variations in the spatial and temporal consistency of AVHRR and GLASS daily $R_n$ retrievals against cloud optical thickness (COT) in (a) January and (b) July 2008. The absolute difference is defined as $\left|Rn_{avhrr} - Rn_{glass}\right|$. The shading represents the variation range (stand deviation) of global daily AVHRR and GLASS $R_n$ retrievals and their absolute differences.**

### 4.3.4 Temporal analysis

To examine the temporal reliability of the generated AVHRR $R_n$ dataset, a long-term analysis of surface $R_n$ for the four datasets was carried out, the results of which are shown in Fig. 12. In view of missing values for the Polar Regions, we focused on surface $R_n$ within the $\pm 60\,°$latitudes region. Overall, the AVHRR $R_n$ retrievals are highly consistent with MERRA2 $R_n$ values during the period of 1981 to 2019 as well as CERES and GLASS $R_n$ retrievals after 2000. However, the MERRA2 and CERES $R_n$ values are generally higher than AVHRR $R_n$ retrievals. Inter-comparison results illustrated that the CERES and MERRA2 $R_n$ values are overestimated against ground-based measurements. GLASS $R_n$ temporal profile is more consistently correlated with the AVHRR $R_n$ retrievals.

Note that the LTDR project only uses afternoon satellite to generate the AVHRR product to do with the high uncertainty of the atmospheric correction algorithm when applied to low sun elevation pixels present in morning (am) satellites. Afternoon satellites include NOAA-7, NOAA-9, NOAA-11, NOAA-14, NOAA-16, NOAA-18, NOAA-19, and NOAA-20. The use of these satellites alone inevitably leads to small gaps in the data in exchange for a higher accuracy in the atmospheric correction. The time series is not fully complete and presents some observational gaps. Specifically, some large discrepancies occur,

during some periods including 1994–1995, 1999–2000, 2007–2008, and 2018–2019. These periods correspond to the alternative update times of the NOAA-series satellites. For example, NOAA-11 was successfully succeeded by NOAA-14 from 1994 to 1995. Important gaps and noise were found in the images from March to September and empty data from September to December, due to NOAA-11 orbital degradation. NOAA-16 replaced NOAA-14 in 2000 for monitoring of the Earth's surface and atmosphere. During these periods of satellite replacement, the corresponding AVHRR data also contain large gaps. Similarly, NOAA-20 was launched on November 18, 2017, yet the quality of the AVHRR TOA observations from this platform was poor due to important gaps in the images and the presence of artefacts. This explains the abnormal temporal variations in the AVHRR $R_n$ profile in these years shown in Fig. 12(a). Therefore, effective multi-source data fusion algorithms and spatial gap-filling technology are urgently needed to improve the quality and coverage of the AVHRR $R_n$ dataset.

The temporal variations in monthly $R_n$ anomalies for the four datasets are shown in Fig. 12(b). High temporal consistency exists between AVHRR $R_n$ anomalies and the other three datasets. Specifically, the correlation coefficient for the AVHRR and MERRA2 $R_n$ anomalies for 1981–2019 is 0.952, and for the period after 2000, is 0.957 and 0.956 for the AVHRR and CERES, and AVHRR and GLASS, $R_n$ anomalies, respectively. Thus, the RCNN-derived AVHRR $R_n$ dataset is temporally stable and reliable when the other three $R_n$ datasets are used as a comparative baseline. In fact, the LTDR project has adapted a calibration method that can be consistently applied across the AVHRR instruments onboard various NOAA satellites to account for sensor degradation (Vermote and Kaufman, 1995), which enables a temporally reliable $R_n$ dataset to be produced. Following the approach, overall, our AVHRR $R_n$ dataset is more accurate and shows reasonable spatiotemporal variations compared to the other three datasets. This dataset will play an important role in climate change study.

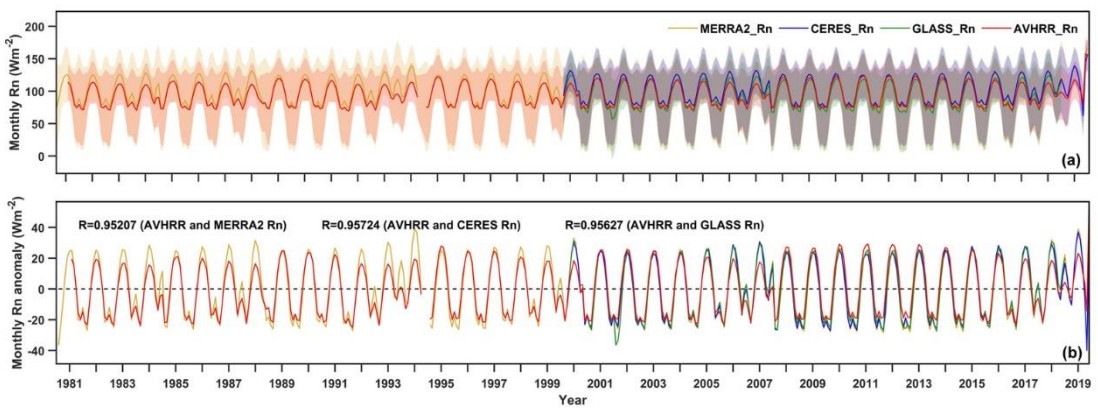

Figure 12: Long-term temporal variation of (a) monthly average $R_n$ and (b) monthly $R_n$ anomalies for the AVHRR, CERES, GLASS and MERRA2 datasets, respectively. The shading represents the variation range (stand deviation) of the global monthly mean $R_n$.

## 5 Discussion

### 5.1 Determination of an appropriate spatial scale

To provide appropriate AVHRR sub-image blocks containing sufficient information for the RCNN model to generate high-accuracy retrievals, the spatial adjacent effects on surface $R_n$ under different valid spatial extents should be examined. For this, we used a simple multivariate linear regression (MLR) model (see supplementary data for further details). The spatial sizes of the sub-images denoted as B3 … B19 vary from $3 \times 3$ to $19 \times 19$, respectively, with an interval of 2 pixels. The true areas correspond to approximately $15 \times 15$ km$^2$ (B3) to $135 \times 135$ km$^2$ (B19) on the ground. The results are shown in Fig. 13. Overall,

the average R increases from 0.61 to 0.708, and RMSE decreases from 50.12 to 46.17, respectively, for the MLR model. As the valid spatial extent increases, essential and complete spatial features are exposed and incorporated into the MLR model, which helps to continuously improve the model's retrieval accuracy. The spatial extent of B13 (approximately $65 \times 65$ km$^2$) is the smallest size that exhibits convergent R and RMSE values, and the spatial extent at the B15 reaches a more stable state for surface $R_n$ estimations. This finding is in line with the results of Jiang et al. (2020b), showing that scale effects have a

considerable impact on solar radiation retrieval accuracy, and distances of approximately 20 to 40 km from the central point (corresponding to areas of $40 \times 40$ km$^2$ to $80 \times 80$ km$^2$), are optimal spatial scale. In addition, previous studies (Hakuba et al., 2013; Huang et al., 2016) recommended a threshold distance of approximately 30 km, equal to a $13 \times 13$ grid region with a spatial resolution of 0.05 °, for shortwave radiation estimation. Therefore, a $15 \times 15$ grid area was selected for the input sub-images to generate AVHRR $R_n$ retrievals.

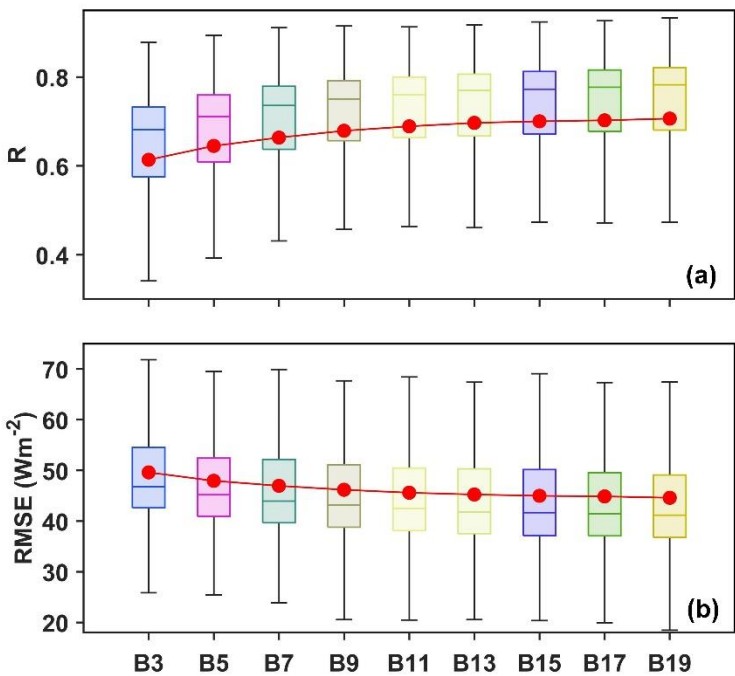

**Figure 13: Variations of (a) R and (b) RMSE indices for each spatial scale in the MLR model. The red lines in the subplots are the average curves of indices at the different spatial scales.**

**5.2 Role of daily mean MERRA2 R$_n$**

The RCNN model uses instantaneous satellite-sensed signals to directly estimate daily mean R$_n$ retrievals. Though some previous studies (Chen et al., 2020; Wang et al., 2015a; Wang and Liang, 2017; Xu et al., 2020) directly estimated daily-averaged surface radiation from instantaneous satellite observations, like MODIS, the idea is theoretically flawed because the AVHRR sensor only offers instantaneous "snapshots", which cannot capture daily mean information about the diurnal cycles of the atmosphere and clouds. King et al. (2013) acknowledged that the frequency of cloud variations is high at different times and locations based on twin MODIS cloud products. In view of the wide satellite overpass times over a particular location, e.g., equatorial crossing time generally ranges from 1300 to 1730 in local time, representing different instantaneous atmospheric conditions for different AVHRR sensors, daily mean MERRA2 R$_n$ is incorporated into the input collection to provide daily mean information about the surface, atmosphere and clouds for the RCNN model.

Figure 14 shows the effect of the daily mean MERRA2 R$_n$ on the final AVHRR R$_n$ retrievals at different AVHRR overpass times in local time. The improved effect is slightly more significant during the afternoon than in the morning when more over-land clouds are present (King et al., 2013). This improvement is also more pronounced during the night. The AVHRR R$_n$ retrievals can only be obtained when solar radiation is available (Fig. 10) because of the missing values in the AVH01C1 product; therefore, the results during the night are based on the validation results for high latitudes, which demonstrate that daily mean information about the diurnal cycles of the atmosphere and clouds is more important for daily surface radiation estimation at high latitudes than that at middle and low latitudes. Shupe et al. (2011) found annual cloud occurrence fractions are 58%–83% at the Arctic observatories, with a clear annual cycle wherein clouds are least frequent in the winter and most frequent in the late summer and autumn.

Additionally, MERRA2 downward shortwave radiation (DSR) was used as a replacement for MERRA2 R$_n$ to test its contribution to daily mean surface R$_n$ estimations when using instantaneous satellite data. The results presented in Fig. S4 show that the improved effect of daily MERRA2 DSR is not comparable with that achieved using daily MERRA2 R$_n$, and the former is closer to the results obtained when only instantaneous AVHRR observations are used. Therefore, MERRA2 R$_n$ is a meaningful input for the RCNN model. Moreover, the AVHRR R$_n$ retrievals could also be further improved by using more accurate daily mean R$_n$ data, such as GLASS R$_n$, or other parameters that accurately represent daily mean atmospheric and cloud variations.

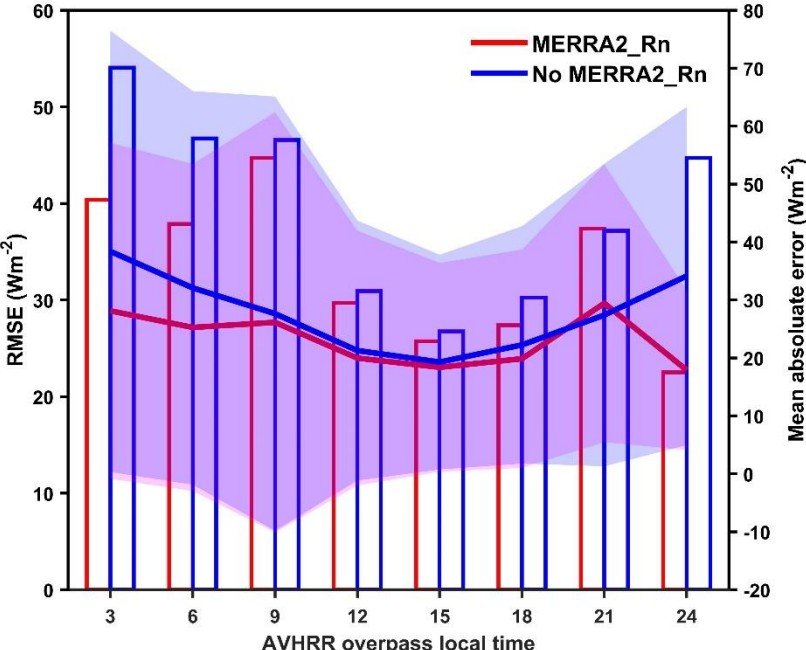

Figure 14: Effect of daily mean MERRA2 $R_n$ on AVHRR $R_n$ retrievals at different satellite crossing times in local time over sites. The bars indicate RMSE and lines indicate absolute biases. The shading shows the variation range of absolute bias.

## 5.3 Determination of a threshold for the ETC-derived correlation coefficient

The threshold for the ETC-derived correlation coefficient between in situ measurements and the unknown truth within the 5 km AVHRR grid in Eq. (2) affects the selection of reliable sites and the subsequent PSME process. A series of thresholds for the ETC-derived coefficients were considered, ranging from 0.2 to 0.9 with an interval of 0.1. In each case, the corresponding measurements from the selected reliable sites were fed into the RCNN model to train and subsequently generate AVHRR $R_n$ retrievals. Then, the training and test accuracies of the RCNN were calculated over all of the reliable sites for comparison. Another important consideration is the representativeness of the RCNN for global $R_n$ estimation, given that the number of reliable training sites decreases with higher thresholds. Thus, the trained RCNN model was again evaluated at all sites, including reliable and unreliable sites, to examine the global representativeness. The number of reliable sites (training and test accuracies) and the associated global accuracy are presented in Fig. 15. As the threshold increases, the number of reliable sites decreased. The training and test relative RMSEs of the RCNN model showed a general decreasing trend, especially above a threshold of 0.5, which illustrates that the selection of reliable sites and the measurements from these sites have better representativeness for the AVHRR footprint scale using ETC. This helps address the spatial scale mismatch issue and improve the accuracy of AVHRR retrievals at a 5 km resolution. In addition, a trade-off between the RCNN's fitting accuracy at the reliable sites and the global accuracy at all sites needs to be considered. Even when a threshold of 0.9 was used, the global accuracy of the RCNN was only slightly lower, which explains why this threshold was applied in Sect. 3.1.

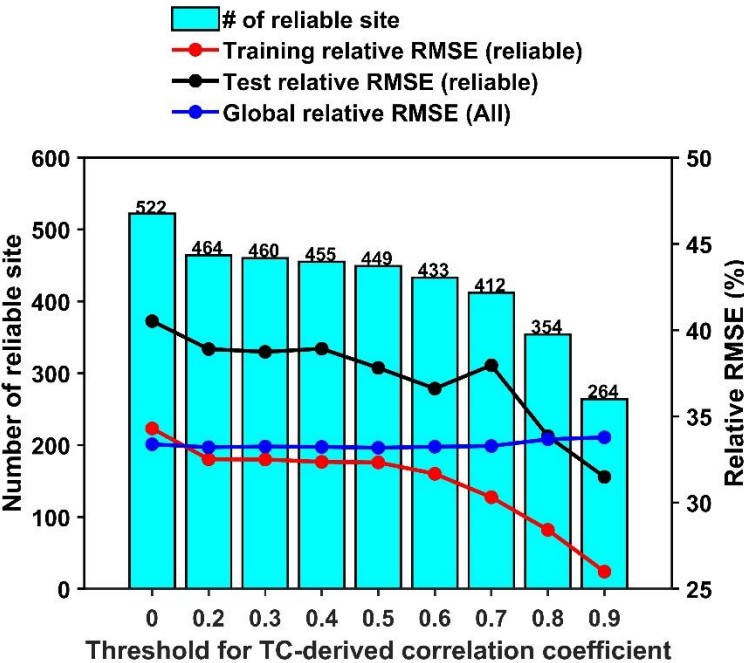

**Figure 15: Effect of extended triplet collocation (ETC)-derived correlation coefficients on the number of reliable sites and the corresponding RCNN's training, test, and global accuracies.**

## 5.4 Orbital drift of the NOAA-series satellites

The orbital drift problem of the NOAA-series satellites has attracted the attention of users applying AVHRR-derived high-level remote sensing products, such as land surface temperature (LST) (Ma et al., 2020; Liu et al., 2019) and TOA albedo (Song et al., 2018). As shown in Fig. 16, the orbital drift makes the true equatorial crossing time (ECT) of the NOAA-series afternoon satellites range from 13:00 to 17:30 in the solar time system. Previous geophysical variable retrievals based on AVHRR data are instantaneous values at satellite overpass times, which need to be corrected to a specific time, such as LST at 14:30 and albedo at noon local time. However, the RCNN model uses instantaneous AVHRR TOA observations at different satellite overpass times to directly retrieve daily surface $R_n$ estimates, which differs from previous studies.

Additionally, daily MERRA2 $R_n$ and instantaneous SZA values closely related to satellite transit times are taken as inputs; therefore, the RCNN model can automatically learn the relationships between instantaneous satellite data at different overpass times and corresponding daily surface $R_n$ measurements. Moreover, the results of the long-term temporal analysis of the AVHRR $R_n$ dataset provide more evidence to ensure that the quality of the long-term AVHRR daily $R_n$ datasets is not affected by orbital-drift. As such, orbital drift does not affect our long-term AVHRR $R_n$ dataset.

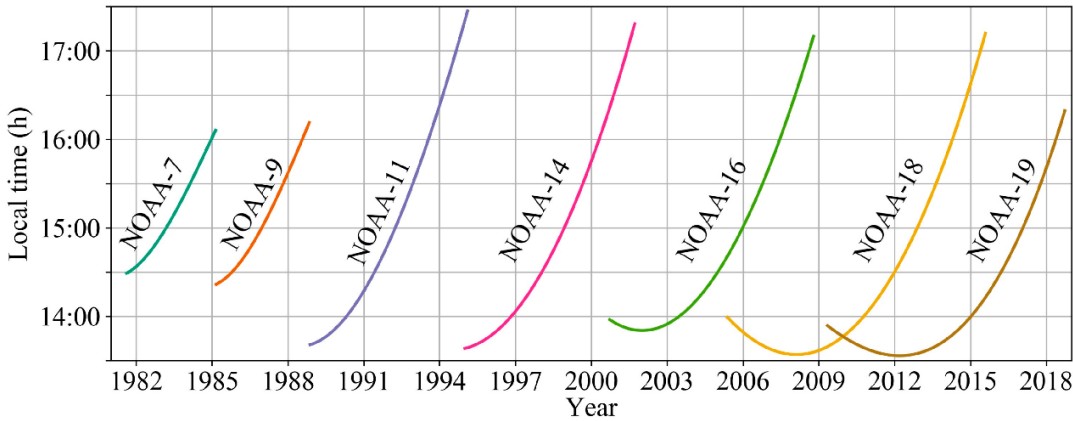

**Figure 16: Equatorial Crossing Time (ECT) for the National Oceanic and Atmospheric Administration (NOAA)-series afternoon Satellites. Figure obtained from Liu et al. (2019).**

## 6 Data availability

Global surface $R_n$ retrieved from NOAA/AVHRR data from 1981 to 2019 are freely available at https://doi.org/10.5281/zenodo.5546316 for 1981-2019 (Xu et al., 2021).

The AVH02C1 product data were downloaded from Level-1 and Atmosphere Archive & Distribution System Distributed Active Archive Center (https://ladsweb.modaps.eosdis.nasa.gov/, last access: 2 July 2021, NASA, 2021). CERES-SYN product was downloaded from CERES team (https://ceres.larc.nasa.gov/, last access: 2 July 2021, NASA, 2021). GLASS $R_n$ product was provided by GLASS team at (http://www.glass.umd.edu/, last access: 2 July 2021, Beijing Normal University and University of Maryland, College Park, 2021). MERRA2 reanalysis was downloaded from the Global Modelling and Assimilation Office (https://gmao.gsfc.nasa.gov/, last access: 2 July 2021, NASA, 2021). The download links of ground-based measurements from different observational networks were referenced to Jiang et al. (2018).

## 7 Conclusions and outlook

A long-term (1981–2019) global daily surface $R_n$ product with spatial resolution of 0.05 °was generated from historical NOAA-series AVHRR data using a RCNN-based PSME method. The specific steps employed were as follows: (1) selecting reliable sites from all sites based on ETC to generate the sample dataset; (2) training and independent testing of the proposed RCNN model; (3) evaluating the AVHRR $R_n$ retrievals against in situ measurements and performing inter-comparisons with three other $R_n$ products (GLASS, CERES-SYN, and MERRA2); and (4) generating and evaluating the long-term AVHRR $R_n$ product.

ETC was first applied to select reliable sites to prepare a sample dataset with better spatial representativeness at the AVHRR footprint scale (i.e., 5 km). In total, 262 sites were classified as reliable sites from a total of 522 sites and used as a sample

dataset for the RCNN model. The proportions of the selected reliable sites representing cropland and grassland surfaces were highest (~66% and ~62%, respectively), while those representing ice/snow surface were lowest (~14%). The sample dataset from the 262 sites ensured that the trained RCNN model had both a good fitting accuracy for the reliable sites and global accuracy across all sites.

A simple MLR model was used to examine the spatial adjacent effects on surface $R_n$ estimation, and a spatial extent of 15 $\times$ 15 pixels (75 $\times$ 75 km$^2$) was then determined as the input size of the RCNN to provide sufficient spatial information. Based on 10-fold CV, the trained RCNN model achieved an $R^2$ of 0.90, with an RMSE of 20.84 Wm$^{-2}$ (25.97%), and a bias of -0.45 Wm$^{-2}$ (-0.57%); the corresponding independent validation values were 0.84, 26.77 Wm$^{-2}$ (31.54%), and 1.16 Wm$^{-2}$ (1.37%) at the reliable sites, respectively. These results demonstrate the overall ability of the RCNN model to accurately predict surface

$R_n$.

The results of an inter-comparison between the AVHRR $R_n$ retrievals and three other products illustrated that our retrievals show a better accuracy against in situ measurements, with an $R^2$ of 0.90, RMSE of 21.08 Wm$^{-2}$ (26.22%), and bias of -0.38 W/m$^2$ (-0.47%) at the reliable sites, and an $R^2$ of 0.85, RMSE of 26.74 Wm$^{-2}$ (35.70%), and bias of 1.20 Wm$^{-2}$ (1.60%) across all sites. At the same time, the AVHRR $R_n$ retrievals show better performance for different observational networks and surface

cover types, except for the snow/ice surface cover. Under different elevations and atmospheric conditions, the AVHRR $R_n$ retrievals performed better than the GLASS $R_n$ equivalent, especially in the presence of thick clouds and hazy atmospheric conditions because of the integration of spatially adjacent information into the inversion process in the RCNN model. In addition, the spatiotemporal variation of the AVHRR $R_n$ retrievals is similar to that of the GLASS $R_n$ values, demonstrating the ability of the RCNN model to generate a long-term global $R_n$ product.

The long-term global $R_n$ dataset generated by the RCNN model displays high accuracy and reasonable spatiotemporal variation at the global scale, which is suited to many applications including, for example, studies to understand the radiation budget and global climate change. Besides, compared to current satellite-derived $R_n$ products, e.g., CERES-SYN and GLASS (2000-present), a more long record (1981-2019) of the AVHRR $R_n$ dataset shows its value in climate change studies. However, further research is needed to solve some problems to further improve the data quality of the AVHRR $R_n$ dataset. First, new

algorithms and satellite data are needed to estimate surface $R_n$ in the Polar Regions, such as MODIS data (Chen et al., 2020). Second, an effective data gap-filling method or multi-source data-fusion algorithm is required to fill the data gaps over land, especially during periods of satellite replacement work. Third, coupled with spatially adjacent information, real-time temporal information, or historical information should be incorporated to further improve the accuracy of the $R_n$ retrievals.

As a type of machine learning, deep learning involves using data-driven models to find potential relationships and patterns,

and offers high adaptability to training data sample inputs. The predictive ability of a data-driven model completely depends

on the limitations of the training dataset and in the case of $R_n$, the ability of the model to accurately portray spatiotemporal dynamics in areas where the availability of training data is relatively poor, such as for AVHRR $R_n$ retrievals for ice/snow-covered surfaces. To address this problem, more physical knowledge is needed to fully utilize data-driven modeling to estimate surface $R_n$ under different atmospheric and surface conditions. In particular, more attention should be paid to understanding inherent physical processes in addition to obtaining optimal estimation by coupling physical process models with the versatility of data-driven machine learning (Reichstein et al., 2019).

**Author contributions**

JX and SL contributed to the design of this study and developed the overall methodology. JX and BJ collected and preprocessed the data. JX carried out the experiment and produced the product. JX wrote the first draft. All authors revised the manuscript.

**Competing interests**

All authors declare that they have no conflicts of interest.

**Acknowledgements**

The authors would like to thank the LTDR project for providing AVH02C1 product, CERES team for offering CERES-SYN flux product, NASA team for MERRA2 product and GLASS team for GLASS $R_n$ product. We express our gratitude to each observation network project for providing ground-based radiation measurements, including ARM, BSRN, AsiaFlux, GCNET, Global FluxNet, PROMICE and other networks/programs.

**Financial support**

This work was supported by the Chinese Grand Research Program on Climate Change and Response (project 2016YFA0600103) and the National Natural Science Foundation of China (grant 41971291).

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
