# Peer review of "A global long-term (1981–2019) daily land surface radiation budget product from AVHRR satellite data using a residual convolutional neural network"

_Earth System Science Data, 2021_

## Author Comment (AC2)

**Response to referee #1**

Title: A global long-term (1981–2019) daily land surface radiation budget product from AVHRR satellite data using a residual convolutional neural network

MS_No: ESSD-2021-250

Thanks very much for taking your time to review this manuscript. We really appreciate all your valuable comments and constructive suggestions! The specific responses to your all comments are listed below one by one.

**General comments:**

**Comment 1:**

Getting finer resolution from coarse resolution data is not easily accepted without clear explanation. You explain that "spatial adjacent effect" is accomplished by applying reanalysis data and angular information of the satellite measurement and solar position. The spatial adjacent effect seems to be the novel advantage of the CNN method over other existing methods, but the explanation lacks detail and examples to help the reader better understand the upscaling process. Does it use reanalysis vertical profiles to correct the path of upwelling radiation to the satellite? Or is it some kind of statistical approach?

**Response 1:**

We use CNN model to upscale the *in-situ* measurements at "points" to a 0.05° spatial resolution. We selected these sites whose measurements can well represent the average state of surface $R_n$ at a 5-km geospatial extent using the ETC method. Errors of the upscaling process can be weakened to a certain degree because of a good spatial representation of selected ground-based measurements within AVHRR footprint. MERRA2 reanalysis has a spatial resolution of 0.5°×0.625°. Therefore, MERRA2 data are resampled to the 0.05° resolution using the nearest neighbor method to avoid introducing new errors.

thresholds between 0.2 and 0.9 at intervals of 0.1, a threshold of 0.9 was selected, above which the sites were assigned as 'reliable' (also see Sect. 5). Errors of upscaling reliable site-based measurement can be weakened to a certain degree due to its better representativeness within the AVHRR footprint.

We do not use the reanalysis vertical profiles to correct the path of upwelling radiation to the satellite. Instead, we provide comprehensive information within a determined optimal geospatial extent for CNN to automatically extract the most important features related to reliable site-based $R_n$ measurements based on multiple filters. The influence of spatial adjacent effect on surface radiation is highly related to the viewing geometry of sun-target-sensor (Wang et al., 2017), surface, and atmospheric conditions, e.g., the presence of clouds (Wyser et al., 2002), by multiple scattering, reflection, and absorption in the entire atmosphere column on pixel scale. Surface net radiation is generally inferred from satellite-observed radiance based on the independent pixel approximation in the past retrieval algorithms. However, with the increased spatial resolution, the spatial adjacent effects (or 3-D radiative effects) caused by clouds, water vapor, and aerosols become more significant and are not ignored in the inversion process. To address the spatial adjacent effect on surface $R_n$, a proper geospatial extent centered on the site was determined by the MLR method using AVHRR TOA observations. Comprehensive surface and atmospheric information within the determined spatial extent is necessarily considered in the inversion process of surface $R_n$. CNN is a tool to properly process the input data in a form of the multi-dimensional matrix. Therefore, CNN can extract the most essential feature from the input spatial data within the determined spatial extent to relate with site-based measurements, which is better than using the information on individual pixels. The operation weakens spatial adjacent effect on the surface $R_n$ at the center pixel to a certain degree. The input features include AVHRR TOA observations representing comprehensive surface and atmospheric information, viewing geometry, and MERRA2 $R_n$ addressing the difference in temporal scales. In this way, more accurate $R_n$ values are obtained at the center pixel. More explanations are included in the revised manuscript.

As the spatial resolution of satellite sensors increases, the spatial adjacent effects induced by spatially inhomogeneous atmospheric constitutes (or clouds) fields become more significant, for example, clouds affect the distribution of surface radiation in a region larger than the resolution of an individual pixel. One spatial adjacent effect is the diffusion of radiation that removes part of radiation from an atmospheric column and transfer it to neighboring columns. Two other effects are related to the solar and viewing geometry, such as a shift of the apparent position of clouds and their shadows. Surface $R_n$ is no longer accurately estimated with retrieval algorithms based on the individual pixel approximation (IPA). Comprehensive information within a certain spatial extent centered at reliable sites needs to be applied to help retrieve surface $R_n$. CNN model can extract features hierarchically from input multi-channel images using multiple filters. Therefore, the most important feature information regarding reliable site-based $R_n$ measurements can be effectively extracted by CNN within a certain spatial extent rather than on IPA, to help retrieve $R_n$, which weakens the spatial adjacent effects to a certain extent.
* * *
**Specific comments:**

**Comment 1:**

Include the temporal resolution of the Rn estimates here.

**Response 1:**

Thanks for your suggestion. The temporal resolution of the Rn estimates has been included in line 14 as follows.

network (RCNN) integrating spatially adjacent information to improve the accuracy of retrievals. A global high-resolution (0.05°), long-term (1981–2019), and daily mean $R_n$ product was subsequently generated from Advanced Very High-Resolution

Radiometer (AVHRR) data. Specifically, the RCNN was employed to establish a nonlinear relationship between globally
* * *
**Comment 2:**

20-24 The statement beginning with "Inter-comparisons with three…" is not true. In section 4.3.3 you state: "The validation results in Fig. 8 and Table 7 for the ice/snow surface cover type further confirm that GLASS Rn product may offer a better performance in Greenland region."

**Response 2:**

We are extremely grateful for you to point out the problem. We have revised the statement in the
revised manuscript as follows.

and bias of 0.84, 26.66 Wm$^{-2}$ (31.66%), and 1.59 Wm$^{-2}$ (1.89%), respectively. Inter-comparisons with three other R$_n$ products, i.e., the 5 km Global Land Surface Satellite (GLASS), the 1° Clouds and the Earth's Radiant Energy System (CERES), and the 0.5° × 0.625° Modern-Era Retrospective analysis for Research and Applications, Version 2 (MERRA2), illustrate that our AVHRR R$_n$ retrievals have the best accuracy under most of the considered surface and atmospheric conditions, especially thick cloud or hazy conditions. However, the performance of the model needs to be further improved for the snow/ice cover surface. The spatiotemporal analyses of these four R$_n$ datasets indicate that the AVHRR R$_n$ product reasonably replicates the

**Comment 3:**

40 "radiation" is not needed in front of "radiometers"

**Response 3:**

Thanks for your nice suggestion. The "radiation" has been deleted in the revised manuscript.

ground-based measurements are widely used to study spatiotemporal variations in regional surface radiation and to evaluate gridded products (Jia et al., 2018; Zhang et al., 2020; Zhang et al., 2015). Nevertheless, the high cost of maintaining radiation radiometers means that stations are sparely distributed, severely hindering our ability to study and understand the

**Comment 4:**

132-161 This section describes the instruments used in the various networks which range from good thermopile pyranometers and pyrgeometers to not-so-good net radiometers. You only provide performance measures for the thermopile pyranometers, which are generally good. You don't
provide any performance information on the net radiometers, which are notoriously bad, especially the REBS model. According to Table 2, net radiometers dominate your observational dataset. You should provide performance measures of the net radiometers. Also, Table 2 is incomplete. For some you specify "Eppley PIR," and others just "Eppley."

**Response 4:**

Thanks for your kind suggestion. After extensively reading literature and corresponding websites, the uncertainty of each instrument is listed as follows, and the corresponding content is also included in the revised manuscript.

To be specific, the operational thermoelectric pyranometers are known for their high-accuracy performance, with a spectral response of 0.3-3.0 μm, a sensitivity of 7-14μVW$^{-1}$ m$^2$, a thermal effect of less than 5%, and an annual stability of 5% (Lu et al., 2011; Jiang et al., 2019). The Eppley Precision Infrared Radiometers (PIR, 3.5-50 μm) and Kipp & Zonen CG 4 pyrgeometers (4.5-42 μm) are applied to measure the surface radiation with a uncertainty of ± 6% or 15 Wm$^{-2}$ at the 95% confidence level (Philipona et al., 1998). The largest uncertainty for surface radiation measurements is ~2% for pyrheliometers and ~5% for pyranometers (i.e., 15 Wm$^{-2}$), respectively (Augustine et al.,

2000). Additionally, the radiation measurements obtained by Kipp & Zonen CNR1 and CNR4 instruments are with an expected accuracy of ±10% for daily totals (Wang and Dickinson, 2013). The radiation observations measured by Kipp & Zonen net radiometers (CNR1, 5-50 μm or CNR1-lite, 4.5-42 μm), are with uncertainty of ~10% at 95% confidence level for daily totals (Yamamoto et al., 2005). Besides, the uncertainties of the shortwave radiation measured by LI-COR Photodiode and R$_n$ observed by REBS Q*7 are about 5 (5-15%) and 10 Wm$^{-2}$ (5-50%), respectively, at monthly time scale (Box and Rinke, 2003; Steffen and Box, 2001).

To be specific, the operational thermoelectric pyranometers are known for their high-accuracy performance, with a spectral response of 0.3-3.0 μm, a sensitivity of 7-14μVW$^{-1}$ m$^2$, a thermal effect of less than 5%, and an annual stability of 5% (Lu et al., 2011; Jiang et al., 2019b). The Eppley Precision Infrared Radiometers (PIR, 3.5-50 μm) and Kipp & Zonen CG 4 pyrgeometers (4.5-42 μm) are applied to measure the surface radiation with a uncertainty of ± 6% or 15 Wm$^{-2}$ at the 95% confidence level (Philipona et al., 1998). The largest uncertainty for surface radiation measurements are ~2% for pyrheliometers and ~5% for pyranometers (i.e., 15 Wm$^{-2}$), respectively (Augustine et al., 2000). Additionally, the radiation measurements obtained by Kipp & Zonen CNR1 and CNR4 instruments are with an expected accuracy of ±10% for daily totals (Wang and Dickinson, 2013). The radiation observations measured by Kipp & Zonen net radiometers (CNR1, 5-50 μm or CNR1-lite, 4.5-42 μm), are with uncertainty of ~10% at 95% confidence level for daily totals (Yamamoto et al., 2005). Besides, the uncertainties of the shortwave radiation measured by LI-COR Photodiode and R$_n$ observed by REBS Q*7 are about 5 (5-15%) and 10 Wm$^{-2}$ (5-50%), respectively, at monthly time scale (Box and Rinke, 2003; Steffen and Box, 2001). To deal with equipment and operational errors, daily mean surface R$_n$ measurements were calculated based on several strict

| Network/Program | Instrument | Temporal Interval | Number of sites | |
|---|---|---|---|---|
| ARM | Kipp&Zonen CNR-1 | 10 minutes | 34 | |
| AsiaFlux | Kipp&Zonen CNR-1/EKO MS201 | 30 minutes | 31 | |
| BSRN | Kipp&Zonen CG4/Eppley PIR | 1 minutes | 21 | |
| CEOP | Eppley PIR/EKO MS202 | 30 minutes | 16 | |
| CEOP-Int | Kipp&Zonen CG4/Eppley PIR | 30 minutes | 8 | |
| ChinaFlux | Kipp&Zonen CNR-1 | 30/60 minutes | 3 | |
| EOL | Kipp&Zonen pyrgeometers, Eppley PIR | 30/60 minutes | 17 | |
| GCNET | Li Cor Photodiode & REBS Q*7 | 60 minutes | 18 | |
| GAME.ANN | EKO MS0202F | 30 minutes | 3 | |
| Global FluxNet | Kipp&Zonen CNR-1, etc. | 30 minutes | 314 | |
| HiWATER | Kipp&Zonen CNR-1/CNR-4 | 10 minutes | 19 | |
| IMAU-Ktransect | Kipp&Zonen CNR-1 | 60 minutes | 4 | |
| LBA-ECO | Kipp&Zonen CG2/CNR-1 | 30 minutes | 8 | |
| PROMICE | Kipp&Zonen CNR-1/CNR-4 | 10 minutes | 24 | |
| SURFRAD | Eppley pyrgeometer | 1/3 minutes | 7 | |

**Comments 5:**

By "thermal effect," are you referring to the thermal offset of single black detector pyranometers? If so, there are references for this measurement error.

**Response 5:**

The "thermal effect" refers to the thermal offset of the thermopile pyranometers. The measurements error is referred to studies of Jiang et al. (2019) and Lu et al. (2011)

**Comment 6:**

169. What does "along with inverse navigation to relate a specific Earth location to each sensor's instantaneous field of view" mean?

**Response 6:**

The sentence of "along with inverse navigation to relate a specific Earth location to each sensor's instantaneous field of view" means geometric correction which is one of the three components of the AVHRR Land Pathfinder II processing system. The other two components are radiometric in-flight vicarious calibrations for the visible and near-infrared channels and atmospheric correction, respectively (Pedelty et al., 2007). Specifically, navigation is a process that relates an Earth location to an instantaneous field of view (IFOV) of the sensor. The inverse navigation refers that the nearest IFOV scan number and position are determined for each grid cell of a predetermined geographic grid, which is a preprocessing for generating a consistent, long-term AVHRR data set at a resolution of 0.05° (El Saleous et al., 2000).

**Comment 7:**

180-194 In this description of the GLASS product, Rn is estimated from downward shortwave radiation, and other variables using multiple MARS learners. Where do the input data come from?

**Response 7:**

The shortwave radiation, albedo, and NDVI data are from GLASS products (Liang et al., 2020; Xiao et al., 2017). Other meteorological varibales come from MERRA2 reanalysis (Gelaro et al., 2017). The related information is included in the revised manuscript.

index (NDVI). Multiple MARS learners were employed to establish efficient statistical relationships using GLASS downward shortwave radiation and MERRA2 meteorological variables, allowing land surface $R_n$ to be estimated from these inputs across most spatial domains (Jiang et al., 2016; Jiang et al., 2015). Conversely, when surface solar radiation data were not available,

**Comment 8:**

217 The last phrase of this sentence "the diurnal variation of daily surface Rn." Does not make sense.

**Response 8:**

Thanks for your nice suggestion. The last phrase of "the diurnal variation of daily surface Rn" has been deleted in the revised manuscript.

> to other reanalysis data. Therefore, MERRA2 $R_n$ data calculated from four surface radiative components were also used in this study to help retrieve accurate high-resolution surface $R_n$ estimates by providing average atmospheric information

**Comment 9:**

What does "when deeper networks converge" mean?

**Response 9:**

A deeper network converages, meaning the training and test errors no longer decrease with increasing training epochs. However, a degradation problem may expose that training accuray gets saturated and then degrades rapidly with network depth increasing. In other words, adding more layers to a suitably network leads to higher training error (He and Sun, 2015; Srivastava et al., 2015), though the deeper network starts converaging. The residual learning framework proposed by He et al. (2016) is thus applied to deal with the degradation problem.

**Comment 10:**

This sentence does not make sense. Do you mean "Reliable and unreliable sites from each
observation network, separated by a threshold ETC-derived correlation coefficient of 0.9, are listed in Table 5"?

**Response 10:**

Thanks for your kind suggestion. Your understanding of this sentence is right! We have revised the sentence according to your expression. The number of reliable and unreliable sites for each observation
network, identified by a threshold of 0.9 for the ETC-derived correlation coefficient, is listed in Table 5.

>  The number of reliable and unreliable sites for each observation network, identified by a threshold of 0.9 for the ETC-derived correlation coefficient, is listed in Table 5. A total of 275 sites could be considered reliable, accounting for ~48% of the sites. Furthermore, no site was considered reliable for some observation

**Comment 11:**

334 - 337 Please include references for the ARM, SURFRAD, BSRN, and FluxNet networks.

**Response 11:**

We are grateful for the suggestion. The corresponding references for the four networks are included in the revised manuscript. ARM (Stokes and Schwartz, 1994), SURFRAD (Augustine et al., 2000), BSRN (Ohmura et al., 1998), and FluxNet (Wilson et al., 2002).

networks. In contrast, some of the international observational networks, such as BSRN (Ohmura et al., 1998) and FluxNet (Wilson et al., 2002), provide many ground-based measurements with sufficient spatial representativeness for $R_n$ at 5 km resolution. In addition, the ARM (Stokes and Schwartz, 1994) and SURFRAD (Augustine et al., 2000) networks were classified as containing reliable sites. In situ measurements from the SURFRAD (Augustine et al., 2000) network were well

**Comment 12:**

I assume the color bar represents a normalized count scaled to the most frequent count. Regardless,
explain the color bar in the caption.

**Response 12:**

Thanks for your suggestion. The color bar illustrates the normalized density of samples. The corresponding explain has been included in the caption of the figure.

Figure 5: Scatterplots of (a) mode training (fitting) accuracy and (b) model test accuracy for the reliable training and independent validation sites. The color bar illustrates the normalized density of samples.

**Comment 13:**

"under snow and ice surfaces" ? Perhaps use "for snow and ice surfaces" ?

**Response 13:**

Thanks for your suggestion. We have revised the phrase according to your comment.

PROMICE network for most of the large sites in the GCNET and PROMICE networks identified as unreliable sites. Thus, the RCNN model has less knowledge of $R_n$ dynamics under for snow and ice surfaces. The most significant difference for RMSE was observed over the ARM network, for which the mean RMSE value decreased by 2.1 Wm$^{-2}$ for the AVHRR $R_n$ retrievals

**Comment 14:**

453 Change phrasing to "…especially clouds that have significant impacts on shortwave…"

**Response 14:**

Thanks for your suggestion. We have revised the phrase according to your comment.

clouds and CWV control surface $R_n$ dynamics under cloud-sky conditions,  especially clouds that have significant impacts on shortwave and longwave radiation. Therefore, AOD, CWV, and cloud optical thickness (COT, as a surrogate for cloud optical properties) were

**Comment 15:**

Where do cloud optical thickness (COT) and cloud water vapor (CWV) data come from?

**Response 15:**

The COT and CWV data come from MERRA2 reanalysis. We have added the information in the revised manuscript.

radiation. Therefore, AOD, CWV, and cloud optical thickness (COT, as a surrogate for cloud optical properties) derived from MERRA2 were employed to analyze the sensitivity of the accuracy of the AVHRR and GLASS $R_n$ retrievals to variations in these influencing factors. In addition, $R_n$ retrieval performance at different elevations was also evaluated.

**Comment 16:**

462 The sentence beginning with "Therefore, the performance…" does not make sense. Perhaps the end of that sentence should read: "…is comparable with regard to the accuracy of their Rn retrievals."

**Response 16:**

Thanks for your kind suggestion. We have revised the phrase according to your comment.

clear-sky conditions, which results in surface total solar radiation dominated direct solar radiation. Therefore, the performance of the RCNN model and the MARS models used for the GLASS $R_n$ product (Jiang et al., 2016) is comparable  with regard to the accuracy of their $R_n$ retrievals. However, when the absorption and scattering effects are enhanced for direct solar radiation from TOA, depending on the individual pixel approximation (IPA), it is difficult to retrieve

**Comment 17:**

– 519 I don't understand your Figure 11. The AVHRR and GLASS Rn's as a function of COT are nearly on top of each other, yet the bias plotted on the same charts is significant. What am I missing here?

Regardless, in the caption please define the bias and what the shading represents.

**Response 17:**

The absolute bias is defined as the absolute difference between daily mean AVHRR and GLASS $R_n$, i.e., $\left| Rn_{avhrr} - Rn_{glass} \right|$. To more easily understand, the bias is replaced by the difference. The shading represents the variation range (stand deviation) of global daily AVHRR and GLASS $R_n$ retrievals and their absolute differences. The related information has been added in the figure caption.

**Figure 11: Variations in the spatial and temporal consistency of AVHRR and GLASS daily R$_n$ retrievals against cloud optical thickness (COT) in (a) January and (b) July 2008. The absolute difference is defined as** $|Rn_{avhrr} - Rn_{glass}|$**. The shading represents the variation range (stand deviation) of global daily AVHRR and GLASS R$_n$ retrievals and their absolute differences.**

The distributions of the absolute differences between daily mean AVHRR and GLASS R$_n$ values in January and July 2008 are shown in Figure 1. There exist large absolute differences in both January and July. It is reasonable that large absolute differences occur in Figure 11 in the manuscript. The reason for the AVHRR and GLASS R$_n$ values being nearly on top of each other (solid lines) is the effect of averaging operation over multiple land pixels within a certain COT range. However, the large variation range (shading) of daily R$_n$ retrievals means a single pixel within a certain COT range may have a large absolute difference under some specific conditions.

[Figure]

Figure 1: The distributions of absolute differences in January (left) and July (right), respectively.

**Comment 18:**

Please state how the difference is defined. Jan.- July or July – Jan.

**Response 18:**

The difference of cloud fraction is defined as Jan.-July. We have included the information in the revised manuscript.

information includes the properties of the entire cloud layer. Figure S2 shows the spatial distribution of the monthly mean cloud cover fraction (CF) at the global scale in January and July, and the corresponding differences in CFs (Jan.-July). In January, the CFs are higher over most land regions except in northern Africa, southern America, northern Austria and southern

**Comment 19:**

Do you mean "northern Australia" and "South America" ?

**Response 19:**

Note the areas where the CFs in July is larger than that in January 2008 (blue areas, Figure 2). The areas
include Central Africa, Southern Asia, Southern Australia and Antarctica. We have revised the
corresponding part in the manuscript.

cloud cover fraction (CF) at the global scale in January and July, and the corresponding differences in CFs (Jan.-July). In

January, the CFs are higher than in July over most land regions except in

 Central Africa, Southern Asia, Southern Australia and Antarctica. However, most regions had smaller CFs

[Figure]

Figure 2: Spatial distribution of the global CFs differences between in January and in July 2008.

**Comment 20:**

"produced by NOAA" ? Should this read "replaced by NOAA"?

**Response 20:**

Thanks for your careful work. We have revised the mistake in the revised manuscript.

periods correspond to the alternative update times of the NOAA-series satellites. For example, NOAA-11 was successfully

 succeeded by NOAA-14 from 1994 to 1995. Similarly, NOAA-16 replaced NOAA-14 in 2000 for monitoring of the

Earth's surface and atmosphere. During these periods of satellite replacement, the corresponding AVHRR data contain large

**Comment 21:**

555-560 The four timeseries in Fig. 12 after 2017 for all data sets may be well correlated but are obviously
wrong and could not be used for climate studies. What does the shading represent in Fig. 12?

**Response 21:**

The LTDR project only uses afternoon satellite to generate the long-term AVHRR dataset because the
atmospheric correction algorithm would produce high uncertainty when applied to low sun elevation
pixels from the morning (am) satellites. Afternoon satellites include NOAA7, NOAA9, NOAA11,
NOAA14, NOAA16, NOAA18 and NOAA19 (Figure 3). The use of these satellites alone inevitably leads to small gaps in the data in exchange for a higher accuracy in the atmospheric correction. The time series is not fully complete and presents some observational gaps. The most important two were found in 1994 and from 2018 onwards. In the first case, important gaps and noise were found in the images from March to September and empty data from September to December, due to NOAA11 orbital degradation. From 2018 onwards the data quality has been degrading due to important gaps in the images and the presence of artefacts (Otón et al., 2021). This is why the $R_n$ timeseries after 2017 seem to be abnormal. At several studies, authors suggest that 1994, 2018, 2019, and 2020 are not used due to the poor quality of AVHRR data (Hansen et al., 2020; Tian et al., 2015). The corrsponding contents were included in the revised manuscript.

The shading represents the variation range of global monthly mean $R_n$. The information has been included in the revised manuscript.

**Figure 12: Long-term temporal variation of (a) monthly average $R_n$ and (b) monthly $R_n$ anomalies for the AVHRR, CERES, GALSS and MERRA2 datasets, respectively. The shading represents the variation range (stand deviation) of the global monthly mean $R_n$.**

Note that the LTDR project only use afternoon satellite to generate the AVHRR product to do with the high uncertainty of the atmospheric correction algorithm when applied to low sun elevation pixels present in morning (am) satellites. Afternoon satellites include NOAA-7, NOAA-9, NOAA-11, NOAA-14, NOAA-16, NOAA-18, NOAA-19, and NOAA-20. The use of these satellites alone inevitably leads to small gaps in the data in exchange for a higher accuracy in the atmospheric correction. The time series is not fully complete and presents some observational gaps. Specifically, some large discrepancies occur,

[Figure]

Figure 3: Local overpass time of all NOA satellites containing the AVHRR sensor. Figure obtained from Clerbaux et al. (2020).

**Comment 22:**

Has the AVHRR calibration across all satellites been applied to the AVHRR data shown in Fig. 12?

**Response 22:**

The LTDR product performs geolocation, calibration, and atmospheric and surface anisotropy correction for all AVHRR sensors aboard the NOAA afternoon (pm) satellites (Vermote and Saleous, 2006; Vermote and Kaufman, 1995; Otón et al., 2021; Franch et al., 2017). The calibration method proposed by Vermote and Kaufman (1995) is applied consistently across the AVHRR instruments onboard various NOAA satellites. Relevant information refers to https://landweb.modaps.eosdis.nasa.gov/cgi-bin/ltdr/ltdr/ltdrPage.cgi?fileName=avhrr_calib.

**Comment 23:**

Do you mean 0.708?

**Response 23:**

Thanks for your carefule examination. I looked back at the statistics, and the best R is 0.708, not 7.08. I
have revised the mistake in the manuscript.

> correspond to approximately $15 \times 15$ km$^2$ (B3) to $135 \times 135$ km$^2$ (B19) on the ground. The results are shown in Fig. 13. Overall,
>
> the average R increases from 0.61 to  0.708, and RMSE decreases from 50.12 to 46.17, respectively, for the MLR model.
>
> As the valid spatial extent increases, essential and complete spatial features are exposed and incorporated into the MLR model,

**Comment 24:**

591 What is a wide overpass time?

**Response 24:**

The wide overpass time refers to a broad range of local time for satellite crossing as shown in Fig. 3 from
13:00 to 20:00 LT for afternoon satellites.

**Comment 25:**

What Study?

**Response 25:**

Shupe et al. (2011) found annual cloud occurrence fractions are 58%–83% at the Arctic observatories,
with a clear annual cycle wherein clouds are least frequent in the winter and most frequent in the late
summer and autumn. We have included the study in the revised manuscript.

> about the diurnal cycles of the atmosphere and clouds is more important for daily surface radiation estimation at high latitudes
>
> than that at middle and low latitudes. Shupe et al. (2011) found annual cloud occurrence fractions are 58%–83% at the Arctic
>
> observatories, with a clear annual cycle wherein clouds are least frequent in the winter and most frequent in the late summer
>
> and autumn.

**Technical corrections:**

**Comment 26:**

You don't need ", respectively" in this sentence.

**Response 26:**

Thanks for your suggestion. I have deleted the ", respectively" in the revised manuscript.

AVHRR TOA observations at five spectral channels (a visible band (0.55–0.68 μm), a near-infrared band (0.75–1.1 μm), a middle-infrared band (3.55–3.93 μm), and two thermal bands (10.5–11.3 and 11.5–12.5 μm, respectively) were utilized for

**Comment 27:**

"BSRN_DRA" site.

**Response 27:**

Thanks for your suggestion. I have revised the "BSRA_DRA" to "BSRN_DRA".

MERRA2 and CERES-SYN $R_n$ retrievals show higher values compared to the in situ measurements at the BSRA_DRA

BSRN_DRA site, especially during 140–200 day period. In comparison, the AVHRR and GLASS $R_n$ values closely match the

**Comment 28:**

"very low" not "vary low"

**Response 28:**

Thanks for your suggestion. We have revised the "vary low" to "very low".

Wm$^{-2}$. The lower accuracy of AVHRR $R_n$ values for these two elevation ranges is attributable to the less reliable sites used for the RCNN training. In addition, the AVHRR $R_n$ retrievals show steady and vary very low (close to zero) biases under different conditions, while the biases of the GLASS $R_n$ retrievals show a high degree of variation. This illustrates that the

**Comment 29:**

1999-2000

**Response 29:**

Thanks for your careful examination. We have corrected the mistake.

some large discrepancies occur, during some periods including 1994–1995, 1999–2000, 2007–2008, and 2018–2019. These periods correspond to the alternative update times of the NOAA-series satellites. For example, NOAA-11 was successfully

**References**

*Augustine, J. A., DeLuisi, J. J., and Long, C. N.: SURFRAD–A national surface radiation budget network*

for atmospheric research, Bulletin of the American Meteorological Society, 81, 2341-2358, 2000.

Box, J. E. and Rinke, A.: Evaluation of Greenland ice sheet surface climate in the HIRHAM regional climate model using automatic weather station data, Journal of Climate, 16, 1302-1319, 2003.

Clerbaux, N., Akkermans, T., Baudrez, E., Velazquez Blazquez, A., Moutier, W., Moreels, J., and Aebi, C.: The Climate Monitoring SAF Outgoing Longwave Radiation from AVHRR, Remote Sensing, 12, 929, 2020.

El Saleous, N., Vermote, E., Justice, C., Townshend, J., Tucker, C., and Goward, S.: Improvements in the global biospheric record from the Advanced Very High Resolution Radiometer (AVHRR), International Journal of Remote Sensing, 21, 1251-1277, 2000.

Franch, B., Vermote, E. F., Roger, J.-C., Murphy, E., Becker-Reshef, I., Justice, C., Claverie, M., Nagol, J., Csiszar, I., Meyer, D., Baret, F., Masuoka, E., Wolfe, R., and Devadiga, S.: A 30+ Year AVHRR Land Surface Reflectance Climate Data Record and Its Application to Wheat Yield Monitoring, Remote Sensing, 9, 296, 2017.

Gelaro, R., McCarty, W., Suárez, M. J., Todling, R., Molod, A., Takacs, L., Randles, C. A., Darmenov, A., Bosilovich, M. G., and Reichle, R.: The modern-era retrospective analysis for research and applications, version 2 (MERRA-2), Journal of climate, 30, 5419-5454, 2017.

Hansen, M., Song, X., DiMiceli, C., Carroll, M., Sohlberg, R., Kim, D., and Townshend, J.: MEaSURES Vegetation Continuous Fields ESDR Algorithm Theoretical Basis Document (ATBD) Version 2.0,    2020.

He, K. and Sun, J.: Convolutional neural networks at constrained time cost, Proceedings of the IEEE conference on computer vision and pattern recognition, 5353-5360,

He, K., Zhang, X., Ren, S., and Sun, J.: Deep Residual Learning for Image Recognition, 2016 IEEE Conference on Computer Vision and Pattern Recognition (CVPR), 770-778, 10.1109/CVPR.2016.90, 2016.

Jiang, H., Lu, N., Qin, J., Tang, W., and Yao, L.: A deep learning algorithm to estimate hourly global solar radiation from geostationary satellite data, Renewable & Sustainable Energy Reviews, 114, 109327, 2019.

Liang, S., Cheng, J., Jia, K., Jiang, B., Liu, Q., Xiao, Z., Yao, Y., Yuan, W., Zhang, X., and Zhao, X.: The Global LAnd Surface Satellite (GLASS) product suite, Bulletin of the American Meteorological Society, 1-37, 2020.

Lu, N., Qin, J., Yang, K., and Sun, J.: A simple and efficient algorithm to estimate daily global solar radiation from geostationary satellite data, Energy, 36, 3179-3188, 2011.

Ohmura, A., Dutton, E. G., Forgan, B., Fröhlich, C., Gilgen, H., Hegner, H., Heimo, A., König-Langlo, G., McArthur, B., and Müller, G.: Baseline Surface Radiation Network (BSRN/WCRP): New precision radiometry for climate research, Bulletin of the American Meteorological Society, 79, 2115-2136, 1998.

Otón, G., Lizundia-Loiola, J., Pettinari, M. L., and Chuvieco, E.: Development of a consistent global long-term burned area product (1982–2018) based on AVHRR-LTDR data, International Journal of Applied Earth Observation and Geoinformation, 103, 102473, https://doi.org/10.1016/j.jag.2021.102473, 2021.

Pedelty, J., Devadiga, S., Masuoka, E., Brown, M., Pinzon, J., Tucker, C., Vermote, E., Prince, S., Nagol, J., Justice, C., Roy, D., Junchang, J., Schaaf, C., Jicheng, L., Privette, J., and Pinheiro, A.: Generating a Long-term Land Data Record from the AVHRR and MODIS Instruments, 2007 IEEE International Geoscience and Remote Sensing Symposium, 23-28 July 2007, 1021-1025, 10.1109/IGARSS.2007.4422974,

Philipona, R., Fröhlich, C., Dehne, K., DeLuisi, J., Augustine, J., Dutton, E., Nelson, D., Forgan, B.,

*Novotny, P., and Hickey, J.: The Baseline Surface Radiation Network pyrgeometer round-robin calibration experiment, Journal of Atmospheric and Oceanic Technology, 15, 687-696, 1998.*

*Shupe, M. D., Walden, V. P., Eloranta, E., Uttal, T., Campbell, J. R., Starkweather, S. M., and Shiobara, M.: Clouds at Arctic atmospheric observatories. Part I: Occurrence and macrophysical properties, Journal of Applied Meteorology and Climatology, 50, 626-644, 2011.*

*Srivastava, R. K., Greff, K., and Schmidhuber, J.: Highway networks, arXiv preprint arXiv:1505.00387, 2015.*

*Steffen, K. and Box, J.: Surface climatology of the Greenland ice sheet: Greenland Climate Network 1995–1999, Journal of Geophysical Research: Atmospheres, 106, 33951-33964, 2001.*

*Stokes, G. M. and Schwartz, S. E.: The Atmospheric Radiation Measurement (ARM) Program:*
*Programmatic background and design of the cloud and radiation test bed, Bulletin of the American Meteorological Society, 75, 1201-1222, 1994.*

*Tian, F., Fensholt, R., Verbesselt, J., Grogan, K., Horion, S., and Wang, Y.: Evaluating temporal consistency of long-term global NDVI datasets for trend analysis, Remote Sensing of Environment, 163, 326-340, 2015.*

*Vermote, E. and Kaufman, Y.: Absolute calibration of AVHRR visible and near-infrared channels using ocean and cloud views, International Journal of Remote Sensing, 16, 2317-2340, 1995.*

*Vermote, E. and Saleous, N.: Calibration of NOAA16 AVHRR over a desert site using MODIS data, Remote sensing of Environment, 105, 214-220, 2006.*

*Wang, K. and Dickinson, R. E.: Global atmospheric downward longwave radiation at the surface from*
*ground-based observations, satellite retrievals, and reanalyses, Reviews of Geophysics, 51, 150-185, 2013.*

*Wang, T., Shi, J., Husi, L., Zhao, T., Ji, D., Xiong, C., and Gao, B.: Effect of solar-cloud-satellite geometry on land surface shortwave radiation derived from remotely sensed data, Remote Sensing, 9, 690, 2017.*

*Wilson, K., Goldstein, A., Falge, E., Aubinet, M., Baldocchi, D., Berbigier, P., Bernhofer, C., Ceulemans,*
*R., Dolman, H., and Field, C.: Energy balance closure at FLUXNET sites, Agricultural and Forest Meteorology, 113, 223-243, 2002.*

*Wyser, K., O'Hirok, W., Gautier, C., and Jones, C.: Remote sensing of surface solar irradiance with corrections for 3-D cloud effects, Remote Sensing of Environment, 80, 272-284, https://doi.org/10.1016/S0034-4257(01)00309-1, 2002.*

*Xiao, Z., Liang, S., Tian, X., Jia, K., Yao, Y., and Jiang, B.: Reconstruction of long-term temporally continuous NDVI and surface reflectance from AVHRR data, IEEE Journal of Selected Topics in Applied Earth Observations and Remote Sensing, 10, 5551-5568, 2017.*

*Yamamoto, S., Saigusa, N., Gamo, M., Fujinuma, Y., Inoue, G., and Hirano, T.: Findings through the AsiaFlux network and a view toward the future, Journal of Geographical Sciences, 15, 142-148, 2005.*

---

## Author Comment (AC3)

**Response to referee #2**

Title: A global long-term (1981–2019) daily land surface radiation budget product from AVHRR satellite data using a residual convolutional neural network

MS_No: ESSD-2021-250

Thanks very much for taking your time to review this manuscript. We really appreciate all your valuable comments and constructive suggestions! The specific responses to your all comments are listed below one by one.

10

**Major comments:**

**Comment 1:**

One of the real advantages I see with this dataset is the long record—since the dataset starts in 1981 15 and has an accuracy equal to or exceeding other satellite-based estimates, this extends observation-based estimates of the surface radiation budget significantly. That could be of significant value for long-term climate studies. ***The authors could highlight this advantage more strongly in the abstract and conclusions.***

**Response 1:**

20 Thanks for your kind suggestion. We have highlighted the advantage of the long-term record of AVHRR $R_n$ dataset, and the related content has been included in the revised manuscript.

> spatial pattern and temporal evolution trends of $R_n$ observations. The long-term record (1981-2019) of the AVHRR $R_n$ product also shows its value in climate change studies. This dataset is freely available at https://doi.org/10.5281/zenodo.5546316 for
>
> global climate change. Besides, compared to current satellite-derived $R_n$ products, e.g., CERES-SYN and GLASS (2000-present), a more long record (1981-2019) of the AVHRR $R_n$ dataset shows its value in climate change studies. However,

25

**Comment 2:**

Evaluation and training are done against multiple networks, but some of these networks are interconnected, for example, some ARM and all SURFRAD sites are included in the BSRN. As I look at the list of sites in Table S1, it appears that some of these stations are included multiple times. 30 For example, BSRN_DRA is the same station as SF_DRA because the SURFRAD Desert Rock stations is submitted to the BSRN global network. ***This is particularly a problem if any of the independent validation stations are also included in the training dataset. Please look into this duplication.***

**Response 2:**

35      Thanks for your careful examination. After thoroughly reviewing the list of training (460) and
        validation (77) sites based on sites' geographic coordinates (i.e, latitude, longitude, elevation), we
        found several duplicate sites in the training sites group, including one ARM (ARM_E13) site and
        six SURFRAD (SF_TBL, SF_DRA, SF_PSU, SF_SXF, SF_FPK, SF_GCM) sites have
        corresponding duplicated sites in the BSRN network in the training group. Besides, several sites
40      from the AsiaFlux, including FxMt_GCK, FxMt_MSE, FxMt_QHB, FxMt_TMK, FxMt_TSE,
        FxMt_QHB, may be identical to the corresponding sites of the Global FluxNet. However, the same
        site from different observation networks has different time periods of record, e.g., BSRN_DRA
        (1998-2017) and SF_DRA (1999-2019). After these duplicated sites were removed from the training
        dataset, the training statistics were almost the same probably because the duplicated samples are
45      relatively small compared to the total sample population. Therefore, we deleted theses duplicated
        sites from Table S1 and Table 2, and the corresponding Figure 5(a) unchanged. The corresponding
        content was also revised.

        In addition, we found out that there are three same sites in both the training and independent
        validation groups, as shown in Table 1, although they are nominally administrated by different
50      observation networks. The three training sites of Lath_CN-Ha2, Lath_KR-Hnm, and Lath_ID-Pag are
        of the Global FluxNET. For corresponding three validation sites, the CF_HB site belongs to the
        ChinaFlux network; the FxMt_HFK and the FxMt_PDF sites are of the AsiaFlux network. The
        ChinaFlux and AsiaFlux networks are sub-network of the FluxNET project. Therefore, we believe
        that the respective three sites in the training group and the validation group are the same.

55      **Table 1** Summary of duplicate site in both training and validation sites groups.

| Training site name | Latitude (°) | Longitude (°) | Elevation (m) | Validation site name | Latitude (°) | Longitude (°) | Elevation (m) |
|---|---|---|---|---|---|---|---|
| Lath_CN-Ha2 | 37.6086 | 101.3269 | 3203 | CF_HB | 37.6099 | 101.3224 | 3205 |
| Lath_KR-Hnm | 34.55 | 126.57 | 7 | FxMt_HFK | 34.55 | 126.57 | 13.74 |
| Lath_ID-Pag | 2.345 | 114.036 | 30 | FxMt_PDF | 2.345 | 114.0364 | 30 |

        Besides, we also found that several validation sites have extremely similar geographic coordinates
        to the training sites (Table 2). These sites are from the same observation network at local scale.
        These sites do not belong to the same site at both training and validation sites groups, e.g., the
60      Lath_US-Tw1 in the training group and Lath_US-Tw1 (-2, -3) in the validation group. To deal with
        the issue, we have adopted the method that the mean values from these sites' measurements within
        5-km extent were used to match the grid data, as mentioned in section 3 (Line 224).

        **Table 2** Summary of sites with the similar geographic coordinates in training and validation groups.

| Training site name | Latitude (°) | Longitude (°) | Elevation (m) | Validation site name | Latitude (°) | Longitude (°) | Elevation (m) |
|---|---|---|---|---|---|---|---|
| FGI_MET0 002 | 67.361866 | 26.637728 | 179 | FGI_VUO000 2 | 67.361883 | 26.643233 | 180 |
| HAWS17 | 38.8451 | 100.36972 | 1559.63 | HAWS16 | 38.84931 | 100.36411 | 1564.31 |
| Lath_CA-SCB | 61.3089 | -121.2984 | 280 | Lath_CA-SCC | 61.3079 | -121.2992 | 285 |
| Lath_US-Tw1 | 38.1074 | -121.6469 | -9 | Lath_US-Tw2 | 38.1047 | -121.6433 | -5 |
| Lath_US-Tw1 | 38.1074 | -121.6469 | -9 | Lath_US-Tw3 | 38.1159 | -121.6467 | -9 |
| Lath_US-Tw1 | 38.1074 | -121.6469 | -9 | Lath_US-Tw4 | 38.10298 | -121.6414 | -5 |
| IMAU-S10 | 67.0005 | -47.0167 | 1850 | PM-KAN_U | 67.0003 | -47.0253 | 1840 |

65  To keep the independence of validation dataset from training samples, we removed duplicate three sites of the CF_HB, the FxMt_HFK, and the FxMt_PDF from the validation group. Note that we only use measurements of the sites with ETC coefficient of more than 0.9 to weaken upscaling errors of ground-based measurements. The ETC coefficients of the CF_HB and the FxMt_PDF are 0.7492 and 0.0337, respectively. Measurements from the two sites were previously not used in the
70  validation activity. Therefore, we only need to delete the FxMt_HFK site with an ETC coefficient of 0.9225 from the validation group to evaluate the performance of the RCNN model again. Figure 1 shows the evaluated result based on the independent validation sites without/with the FxMt_HFK site. The uncertainty of $R_n$ retrievals at validation sites changes slightly with RMSE values from 26.66 $Wm^{-2}$ to 26.77$Wm^{-2}$.

75  Therefore, previous independent evaluation of $R_n$ retrievals at validation sites is reliable although duplicate three sites are used in training and validation dataset simultaneously. We have revised Figure 5(b) and the corresponding content in the revised manuscript.

[Figure]

**Figure 1:** Evaluated results of RCNN model using independent validation dataset (a) without FxMt_HFK and (b) with FxMt_HFK site measurements, respectively.

**Comment 3:**

I am curious whether the results shown in Figure 7 reflect the fact that some of these networks are included in training the AVHRR dataset. It isn't clear to me from the description whether training stations were also used in this analysis, or whether this only includes independent testing stations and stations that didn't meet the reliability requirements. But even if these validation stations are independent from training data, the network of measurements around the ARM Southern Great Plains sites, for example, may be more similar to each other than a site that is located in a much different climate regime (e.g. independent sites ARM_E06 and ARM_E41 sites). That could lead to overfitting. ***It would be helpful to understand how independent this validation dataset is.***

**Response 3:**

The collected sites come from several local observation networks and international networks. Generally, sites of the local networks are located at the small region (e.g., ARM, HiWATER), while sites of the international networks are distributed over the globe (e.g., BSRN, FluxNet). To fully utilize these networks, we follow the idea of determination of training and validation sites in the GLASS $R_n$ estimation algorithm (Jiang et al., 2016). For an observation network with multiple sites, we randomly selected several sites to serve as independent sites and remaining sites are used as training sites. Regarding to a local network with less sites, all sites are used as training sites to ensure the representativeness of the training dataset in characterizing spatiotemporal variation of surface $R_n$. Based on the strategy, the training and validation sites are finally determined. Therefore, the training and validation dataset both have great representation that reflect different surfaces (land cover types, elevations) and atmospheric properties (climate zone), which is important for evaluating model's robustness. Figure 2 shows the proportional distribution of training and validation sites under different conditions.

[Figure]

**Figure 2:** Proportional distributions of (a-c) training sites and (d-f) validation sites under different climates, elevation ranges, and land covers, respectively. The value in the brackets is total number of sites under specific condition.

The result in Figure 7 is obtained only using the independent validation dataset with ETC coefficients > 0.9 (reliable). We can see that AVHRR and GLASS $R_n$ retrievals have comparable accuracies over most observation networks, and the overall validation result also illustrates that the difference of uncertainty in these two $R_n$ datasets is small (< 1.63%). Specifically, the RMSE differences between AVHRR and GLASS $R_n$ are -2.03 (ARM), -1.31 (BSRN), -1.34 (CEOP), -0.32 (CEOP-Int), 1.41 (EOL), -0.84 (AsiaFlux), -1.32 (FluxNet), 0.22 (PROMICE), and -0.99 (SURFRAD) $Wm^{-2}$, respectively. The performance of RCNN model over ARM network is better than other networks as ARM is a local network with extremely similar conditions for training and validation, which may reveal a false performance of RCNN model. However, some results from BSRN, FluxNet, EOL networks can reflect more information about RCNN model robustness at a larger spatiotemporal extent.

For the small regional network, measurements only reflect the spatiotemporal variation of $R_n$ at a local extent. It is unsuitable to select many sites from local networks as validation sites to evaluate RCNN's independent performance when we want to retrieve surface $R_n$ at global scale. Therefore, more sites from the international networks should be used as the validation sites. Fortunately, the number of site from the local networks is small in the validation group. Most validation sites were used come from the networks at continental or global scales. Specifically, the number of sites from the continental and global networks is more than 89%, including BSRN (2), CEOP (5), EOL (5), AsiaFlux (10), FluxNet (39), and PROMICE (7). Conversely, the number of sites from local networks is small with a proportion < 10%, including ARM (2), HiWATER (1), GAME.ANN (1). Besides, based on the response 2, there is no duplicate site in training and validation site groups, except the CF_HB, the FxMt_HFK, and the FxMt_PDF. Therefore, the independence of validation dataset is adequate to evaluate the overall performance of RCNN model at validation sites.

135 **Comment 4:**

Does Figure 14 show local time? Please label for clarity.

**Response 4:**

Thanks for your nice suggestion. Figure 14 shows the local time of NOAA-series satellites crossing. We have added the information in the caption of figure 14 and the corresponding phrase.

Figure 14 shows the effect of the daily mean MERRA2 $R_n$ on the final AVHRR $R_n$ retrievals at different AVHRR overpass

140 times in local time. The improved effect is slightly more significant during the afternoon than in the morning when more over-

**Figure 14: Effect of daily mean MERRA2 $R_n$ on AVHRR $R_n$ retrievals at different satellite crossing times in local time. The bars indicate RMSE and lines indicate absolute biases. The shading shows the variation range of absolute bias.**

**Minor comments:**

145 **Comment 1:**

Line 50: "RT-based physical methods show a great generalization" I am not sure what this phrase means, please revise for clarity.

**Response 1:**

The phrase refers that different from empirical methods, the application of RT-based physical
150 methods is not subjected to the limitation of training samples at a regional scale; in other words, the RT-based physical models are more applicable to a larger spatiotemporal extent. The phrase has been revised in the revised manuscript.

empirical statistical methods.  The RT-based physical methods are more applicable to a larger spatiotemporal extent because they consider the physical processes of solar radiation from the top

155

**Comment 2:**

Line 310: should it be: "data was *then* removed"?

**Response 2:**

Thanks for your careful examination. We have corrected the mistake in the revised manuscript.

and divided into ten groups. One group of these data was  then removed as a hold-out or validation dataset and the

remaining nine groups of data were treated as the training datasets. The training datasets were used to fit the RCNN model,
160

**Comment 3:**

Line 346: "for in surface radiation estimations." Wording doesn't seem quite right here.

165 **Response 3:**

Thanks for your careful work. We have revised the phrase in the revised manuscript.

identified as unreliability due to the presence of large water bodies within the satellite footprint. Thus, the processing of identifying reliable sites highlights the need to pay more attention to such areas for  surface radiation estimations.

170 **Comment 4:**

Lines 360-361: This sentence is awkwardly written and should be revised. Changing consistently to consistent, and site to sites would improve readability.

**Response 4:**

Thanks for your kind suggestion. We have revised the phrase according to your comments.

implementation of ETC for the selection of reliable  sites ensures more  consistent spatial representativeness of ground-based measurements and AVHRR data, which improves the accuracy of $R_n$ retrievals. Indeed, the CV-derived average

175

**Comment 5:**

Line 483: should be very instead of "vary"

180 **Response 5:**

Thanks for your careful examination. We have corrected the mistake.

for the RCNN training. In addition, the AVHRR $R_n$ retrievals show steady and  very low (close to zero) biases under different conditions, while the biases of the GLASS $R_n$ retrievals show a high degree of variation. This illustrates that the

185 **Comment 6:**

Line 545: should "produced" be replaced?

**Response 6:**

Thanks for your careful examination. We have revised the phrase.

> alternative update times of the NOAA-series satellites. For example, NOAA-11 was successfully  succeeded by
>
> NOAA-14 from 1994 to 1995. Important gaps and noise were found in the images from March to September and empty data

190

**Comment 7:**

Line 563: GLASS is misspelled GALSS

**Response 7:**

195    Thanks for your careful examination. We have corrected the misspelling.

> **Figure 12: Long-term temporal variation of (a) monthly average $R_n$ and (b) monthly $R_n$ anomalies for the AVHRR, CERES, **
>
> **GLASS and MERRA2 datasets, respectively. The shading represents the variation range (stand deviation) of the global monthly**

**Comment 8:**

200    Line 572: I think that 7.08 must be 0.78. Please check.

**Response 8:**

Thanks for your careful examination. After looking back at the evaluated result, the R-value is 0.78, not 7.08. We have corrected the mistake.

> the average R increases from 0.61 to  0.708, and RMSE decreases from 50.12 to 46.17, respectively, for the MLR model.
>
> As the valid spatial extent increases, essential and complete spatial features are exposed and incorporated into the MLR model,

205

**Comment 9:**

Line 690: I think "satellite replacement works" should be satellite replacement work if you are referring to times when there is no satellite data because it the satellites are being worked on.

210    **Response 9:**

Thanks for your kind suggestion. We have revised the phrase according to your comments.

filling method or multi-source data-fusion algorithm is required to fill the data gaps over land, especially during periods of satellite replacement work. Third, coupled with spatially adjacent information, real-time temporal information, or historical information should be incorporated to further improve the accuracy of the $R_n$ retrievals.

215 **Comment 10:**

Line 697: should be "covered surfaces".

**Response 10:**

Thanks for your careful work. We have corrected the mistake.

covered  surfaces. To address this problem, more physical knowledge is needed to fully utilize data-driven modeling to estimate surface $R_n$ under different atmospheric and surface conditions. In particular, more attention should be paid to

220

**Reference:**

Jiang, B., Liang, S., Ma, H., Zhang, X., Xiao, Z., Zhao, X., Jia, K., Yao, Y., and Jia, A.: GLASS daytime all-wave net radiation product: Algorithm development and preliminary validation, Remote Sensing, 8, 225  222, 2016.

---

## Author Response (AR3)

Dear Editor:

I have updated tables and necessary performance metrics throughout the manuscript according to advice of Review #2.

Thank you and two anonymous reviewers for your careful work. These valuable suggestions greatly improve the quality of the manuscript. I look forward to your reply.

Sincerely,

Jianglei Xu
School of Remote Sensing and Information Engineering
Wuhan University
jiangleixu@whu.edu.cn

---

## Author Response (AR5)

**Response to referee #1**

Title: A global long-term (1981–2019) daily land surface radiation budget product from AVHRR satellite data using a residual convolutional neural network

MS_No: ESSD-2021-250

Thanks very much for taking your time to review this manuscript. We really appreciate all your valuable comments and constructive suggestions! The specific responses to your all comments are listed below one by one.

**General comments:**

**Comment 1:**

Getting finer resolution from coarse resolution data is not easily accepted without clear explanation. You explain that "spatial adjacent effect" is accomplished by applying reanalysis data and angular
15   information of the satellite measurement and solar position. The spatial adjacent effect seems to be the novel advantage of the CNN method over other existing methods, but the explanation lacks detail and examples to help the reader better understand the upscaling process. Does it use reanalysis vertical profiles to correct the path of upwelling radiation to the satellite? Or is it some kind of statistical approach?

20   ### Response 1:

We use CNN model to upscale the *in-situ* measurements at "points" to a 0.05° spatial resolution. We selected these sites whose measurements can well represent the average state of surface $R_n$ at a 5-km geospatial extent using the ETC method. Errors of the upscaling process can be weakened to a certain degree because of a good spatial representation of selected ground-based measurements
25   within AVHRR footprint. MERRA2 reanalysis has a spatial resolution of 0.5°×0.625°. Therefore, MERRA2 data are resampled to the 0.05° resolution using the nearest neighbor method to avoid introducing new errors.

'reliable' (also see Sect. 5). Based on these reliable sites, errors of upscaling reliable site-based measurement to a 5-km scale
290   can be weakened to a certain degree due to its better representativeness within the AVHRR footprint.

30   We do not use the reanalysis vertical profiles to correct the path of upwelling radiation to the satellite. Instead, we provide comprehensive information within a determined optimal geospatial extent for CNN to automatically extract the most important features related to reliable site-based $R_n$ measurements based on multiple filters. The influence of spatial adjacent effect on surface radiation is highly related to the viewing geometry of sun-target-sensor (Wang et al., 2017), surface, and
35   atmospheric conditions, e.g., the presence of clouds (Wyser et al., 2002), by multiple scattering, reflection, and absorption in the entire atmosphere column on pixel scale. Surface net radiation is generally inferred from satellite-observed radiance based on the independent pixel approximation in the past retrieval algorithms. However, with the increased spatial resolution, the spatial adjacent effects (or 3-D radiative effects) caused by clouds, water vapor, and aerosols become more

40 significant and are not ignored in the inversion process. To address the spatial adjacent effect on surface $R_n$, a proper geospatial extent centered on the site was determined by the MLR method using AVHRR TOA observations. Comprehensive surface and atmospheric information within the determined spatial extent is necessarily considered in the inversion process of surface $R_n$. CNN is a tool to properly process the input data in a form of the multi-dimensional matrix. Therefore, CNN

45 can extract the most essential feature from the input spatial data within the determined spatial extent to relate with site-based measurements, which is better than using the information on individual pixels. The operation weakens spatial adjacent effect on the surface $R_n$ at the center pixel to a certain degree. The input features include AVHRR TOA observations representing comprehensive surface and atmospheric information, viewing geometry, and MERRA2 $R_n$ addressing the difference in

50 temporal scales. In this way, more accurate $R_n$ values are obtained at the center pixel. More explanations are included in the revised manuscript.

290 As the spatial resolution of satellite sensors increases, the spatial adjacent effects induced by spatially inhomogeneous atmospheric constitutes (or clouds) fields become more significant, for example, clouds affect the distribution of surface radiation in a region larger than the resolution of an individual pixel. One spatial adjacent effect is the diffusion of radiation that removes part of radiation from an atmospheric column and transfer it to neighboring columns. Two other effects are related to the solar and viewing geometry, such as a shift of the apparent position of clouds and their shadows. Surface $R_n$ is no longer

295 accurately estimated with retrieval algorithms based on the individual pixel approximation (IPA). Comprehensive information within a certain spatial extent centered at reliable sites needs to be applied to help retrieve surface $R_n$.

Loosely inspired by the human visual cortex, CNNs were originally applied to analyze common visual imagery using convolution instead of general matrix multiplication (Ball et al., 2017). CNN model can extract features hierarchically from

input multi-channel images using multiple filters. Therefore, the most important feature information regarding reliable site-

300 based $R_n$ measurements can be effectively extracted by CNN within a certain spatial extent rather than on IPA, to help retrieve $R_n$, which weakens the spatial adjacent effects to a certain extent. A general CNN consists of multiple layers of operations,

55 **Specific comments:**

**Comment 1:**

14 Include the temporal resolution of the Rn estimates here.

**Response 1:**

Thanks for your suggestion. The temporal resolution of the Rn estimates has been included in line

60 14 as follows.

network (RCNN) integrating spatially adjacent information to improve the accuracy of retrievals. A global high-resolution (0.05°), long-term (1981–2019), and daily mean $R_n$ product was subsequently generated from Advanced Very High-Resolution

15 Radiometer (AVHRR) data. Specifically, the RCNN was employed to establish a nonlinear relationship between globally

**Comment 2:**

20-24 The statement beginning with "Inter-comparisons with three…" is not true. In section 4.3.3 you state: "The validation results in Fig. 8 and Table 7 for the ice/snow surface cover type further confirm that GLASS Rn product may offer a better performance in Greenland region."

**Response 2:**

We are extremely grateful for you to point out the problem. We have revised the statement in the revised manuscript as follows.

> (MERRA2), illustrate that our AVHRR $R_n$ retrievals have the best accuracy under  most of the considered surface and atmospheric conditions, especially thick cloud or hazy conditions. However, the performance of the model needs to be further
>
> 25   improved for the snow/ice cover surface. The spatiotemporal analyses of these four $R_n$ datasets indicate that the AVHRR $R_n$

**Comment 3:**

40 "radiation" is not needed in front of "radiometers"

**Response 3:**

Thanks for your nice suggestion. The "radiation" has been deleted in the revised manuscript.

> gridded products (Jia et al., 2018; Zhang et al., 2020; Zhang et al., 2015). Nevertheless, the high cost of maintaining
>
> radiometers means that stations are sparely distributed, severely hindering our ability to study and understand the
>
> 45   spatiotemporal variability of surface $R_n$ at global scale.

**Comment 4:**

132-161 This section describes the instruments used in the various networks which range from good thermopile pyranometers and pyrgeometers to not-so-good net radiometers. You only provide performance measures for the thermopile pyranometers, which are generally good. You don't provide any performance information on the net radiometers, which are notoriously bad, especially the REBS model. According to Table 2, net radiometers dominate your observational dataset. You should provide performance measures of the net radiometers. Also, Table 2 is incomplete. For some you specify "Eppley PIR," and others just "Eppley."

**Response 4:**

Thanks for your kind suggestion. After extensively reading literatures and corresponding websites, the uncertainty of each instrument is listed as follows.
*To be specific, the operational thermoelectric pyranometers are known for their high-accuracy performance, with a spectral response of 0.3-3.0 μm, a sensitivity of 7-14μVW⁻¹ m², a thermal effect*

*of less than 5%, and an annual stability of 5% (Lu et al., 2011; Jiang et al., 2019). The Eppley* 95 *Precision Infrared Radiometers (PIR, 3.5-50 μm) and Kipp & Zonen CG 4 pyrgeometers (4.5-42 μm) are applied to measure the surface radiation with a uncertainty of ± 6% or 15 Wm$^{-2}$ at the 95% confidence level (Philipona et al., 1998). The largest uncertainty for surface radiation measurements is ~2% for pyrheliometers and ~5% for pyranometers (i.e., 15 Wm$^{-2}$), respectively (Augustine et al., 2000). Additionally, the radiation measurements obtained by Kipp & Zonen CNR1* 100 *and CNR4 instruments are with an expected accuracy of ±10% for daily totals (Wang and Dickinson, 2013). The radiation observations measured by Kipp & Zonen net radiometers (CNR1, 5-50 μm or CNR1-lite, 4.5-42 μm), are with uncertainty of ~10% at 95% confidence level for daily totals (Yamamoto et al., 2005). Besides, the uncertainties of the shortwave radiation measured by LI-COR Photodiode and R$_n$ observed by REBS Q\*7 are about 5 (5-15%) and 10 Wm$^{-2}$ (5-50%), respectively,* 105 *at monthly time scale (Box and Rinke, 2003; Steffen and Box, 2001).*

The corresponding content is also included in the revised manuscript.

The instruments applied to obtain surface radiation have different uncertainties. To be specific, the operational thermoelectric pyranometers are known for their high-accuracy performance, with a spectral response of 0.3-3.0 μm, a sensitivity of 7-

150 14μVW$^{-1}$ m$^2$, a thermal effect of less than 5%, and an annual stability of 5% (Lu et al., 2011; Jiang et al., 2019b). The Eppley Precision Infrared Radiometers (PIR, 3.5-50 μm) and Kipp & Zonen CG 4 pyrgeometers (4.5-42 μm) are applied to measure the surface radiation with a uncertainty of ± 6% or 15 Wm$^{-2}$ at the 95% confidence level (Philipona et al., 1998). The largest uncertainty for surface radiation measurements are ~2% for pyrheliometers and ~5% for pyranometers (i.e., 15 Wm$^{-2}$), respectively (Augustine et al., 2000). Additionally, the radiation measurements obtained by Kipp & Zonen CNR1 and CNR4

155 instruments are with an expected accuracy of ±10% for daily totals (Wang and Dickinson, 2013). The radiation observations measured by Kipp & Zonen net radiometers (CNR1, 5-50 μm or CNR1-lite, 4.5-42 μm), are with uncertainty of ~10% at 95% confidence level for daily totals (Yamamoto et al., 2005). Besides, the uncertainties of the shortwave radiation measured by LI-COR Photodiode and R$_n$ observed by REBS Q\*7 are about 5 (5-15%) and 10 Wm$^{-2}$ (5-50%), respectively, at monthly time scale (Box and Rinke, 2003; Steffen and Box, 2001). To deal with equipment and operational errors, daily mean surface R$_n$

175 **Monitoring for the Greenland Ice Sheet, SURFRAD: Surface Radiation Budget Network.**

| Network/Program | Sensors | Temporal Interval | Number of sites | |
|---|---|---|---|---|
| ARM | Kipp&Zonen CNR-1 | 10 minutes | 33 | |
| AsiaFlux | Kipp&Zonen CNR-1/EKO MS201 | 30 minutes | 31 | |
| BSRN | Kipp&Zonen CG4/Eppley PIR | 1 minutes | 15 | |
| CEOP | ↓ Eppley PIR/EKO MS202 | 30 minutes | 16 | |
| CEOP-Int | Kipp&Zonen CG4/Eppley PIR | 30 minutes | 8 | |
| ChinaFlux | Kipp&Zonen CNR-1 | 30/60 minutes | 3 | |
| EOL | Kipp&Zonen pyrgeometers, Eppley PIR | 30/60 minutes | 17 | |
| GCNEET | Li Cor Photodiode & REBS Q\*7 | 60 minutes | 18 | |
| GAME.ANN | EKO MS0202F | 30 minutes | 3 | |
| Global FluxNet | Kipp&Zonen CNR-1, etc | 30 minutes | 314 | |
| HiWATER | Kipp&Zonen CNR-1/CNR-4 | 10 minutes | 19 | |
| IMAU-Ktransect | Kipp&Zonen CNR-1 | 60 minutes | 4 | |
| LBA-ECO | Kipp&Zonen CG2/CNR-1 | 30 minutes | 8 | |
| PROMICE | Kipp&Zonen CNR-1/CNR-4 | 10 minutes | 24 | |
| SURFRAD | Eppley, Spectrosun | 1/3 minutes | 7 | |

110

**Comments 5:**

By "thermal effect," are you referring to the thermal offset of single black detector pyranometers? If so, there are references for this measurement error.

**Response 5:**

115    The "thermal effect" refers to the thermal offset of the thermopile pyranometers. The measurements error is referred to studies of Jiang et al. (2019) and Lu et al. (2011). The two references have been included in the revised manuscript.

> 150    14μVW$^{-1}$ m$^2$, a thermal effect of less than 5%, and an annual stability of 5% (Lu et al., 2011; Jiang et al., 2019b). The Eppley
>
> Precision Infrared Radiometers (PIR, 3.5-50 μm) and Kipp & Zonen CG 4 pyrgeometers (4.5-42 μm) are applied to measure

120

**Comment 6:**

169. What does "along with inverse navigation to relate a specific Earth location to each sensor's instantaneous field of view" mean?

**Response 6:**

125    The sentence of "along with inverse navigation to relate a specific Earth location to each sensor's instantaneous field of view" means geometric correction which is one of the three components of the AVHRR Land Pathfinder II processing system. The other two components are radiometric in-flight vicarious calibrations for the visible and near-infrared channels and atmospheric correction, respectively (Pedelty et al., 2007). Specifically, navigation is a process that relates an Earth location

130    to an instantaneous field of view (IFOV) of the sensor. The inverse navigation refers that the nearest IFOV scan number and position are determined for each grid cell of a predetermined geographic grid, which is a preprocessing for generating a consistent, long-term AVHRR data set at a resolution of 0.05° (El Saleous et al., 2000).

135

**Comment 7:**

180-194 In this description of the GLASS product, Rn is estimated from downward shortwave radiation, and other variables using multiple MARS learners. Where do the input data come from?

**Response 7:**

140    The shortwave radiation, albedo, and NDVI data are from GLASS products (Liang et al., 2020; Xiao et al., 2017). Other meteorological varibales come from MERRA2 reanalysis (Gelaro et al., 2017). The related information is included in the revised manuscript.

> 200    daytime lengths and land cover characteristics, which are designated based on the albedo and normalized difference vegetation
>
> index (NDVI). Multiple MARS learners were employed to establish efficient statistical relationships using GLASS downward
>
> shortwave radiation and MERRA2 meteorological variables, that allowed allowing land surface R$_n$ to be estimated from these

**Comment 8:**

217 The last phrase of this sentence "the diurnal variation of daily surface Rn." Does not make sense.

**Response 8:**

Thanks for your nice suggestion. The last phrase of "the diurnal variation of daily surface Rn" has been

 deleted in the revised manuscript.

> 230 surface $R_n$ and its radiative component provide outstanding accuracy and a reasonable spatial-temporal distribution compared
>
> to other reanalysis data. Therefore, MERRA2 $R_n$ data calculated from four surface radiative components were also used in this
>
> study to help retrieve accurate high-resolution surface $R_n$ estimates by providing average atmospheric information about the
>
> diurnal variation of daily surface $R_n$.

**Comment 9:**

 285 What does "when deeper networks converge" mean?

**Response 9:**

A deeper network converages, meaning the training and test errors no longer decrease with increasing training epochs. However, a degradation problem may expose that training accuray gets saturated and then degrades rapidly with network depth increasing. In other words, adding more layers to a suitably

 network leads to higher training error (He and Sun, 2015; Srivastava et al., 2015), though the deeper network starts converaging. The residual learning framework proposed by He et al. (2016) is thus applied to deal with the degradation problem.

**Comment 10:**

 323 This sentence does not make sense. Do you mean "Reliable and unreliable sites from each observation network, separated by a threshold ETC-derived correlation coefficient of 0.9, are listed in Table 5"?

**Response 10:**

 Thanks for your kind suggestion. Your understanding of this sentence is right! We have revised the sentence according to your expression. The number of reliable and unreliable sites for each observation network, identified by a threshold of 0.9 for the ETC-derived correlation coefficient, is listed in Table 5.

> 350 The reliable sites and unreliable sites identified a threshold of 0.9 for the ETC-derived correlation coefficient are summarized
>
> in Table 5 for each observation network. The number of reliable and unreliable sites for each observation network, identified
>
> by a threshold of 0.9 for the ETC-derived correlation coefficient, is listed in Table 5. A total of 275 267 sites could be

**Comment 11:**

334 - 337 Please include references for the ARM, SURFRAD, BSRN, and FluxNet networks.

**Response 11:**

We are grateful for the suggestion. The corresponding references for the four networks are included in the revised manuscript. ARM (Stokes and Schwartz, 1994), SURFRAD (Augustine et al., 2000), BSRN (Ohmura et al., 1998), and FluxNet (Wilson et al., 2002).

> 360     variations in $R_n$. Similar issues also exist in in situ measurements from the IMAU-Ktransect, HiWATER and LBA-ECO
>
> networks. In contrast, some of the international observational networks, such as BSRN (Ohmura et al., 1998) and FluxNet
>
> (Wilson et al., 2002), provide many ground-based measurements with sufficient spatial representativeness for $R_n$ at 5 km
>
> resolution. In addition, the ARM (Stokes and Schwartz, 1994) and SURFRAD (Augustine et al., 2000) networks were
>
> classified as containing reliable sites. In situ measurements from the SURFRAD (Augustine et al., 2000) network were well

**Comment 12:**

I assume the color bar represents a normalized count scaled to the most frequent count. Regardless, explain the color bar in the caption.

**Response 12:**

Thanks for your suggestion. The color bar illustrates the normalized density of samples. The corresponding explain has been included in the caption of the figure.

> 405     **Figure 5: Scatterplots of (a) mode training (fitting) accuracy and (b) model test accuracy for the reliable training and independent**
>
>         **validation sites. The color bar illustrates the normalized density of samples.**

**Comment 13:**

414 "under snow and ice surfaces" ? Perhaps use "for snow and ice surfaces" ?

**Response 13:**

Thanks for your suggestion. We have revised the phrase according to your comment.

> 445     RCNN model has less knowledge of $R_n$ dynamics  for snow and ice surfaces. The most significant difference for RMSE
>
> was observed over the ARM network, for which the mean RMSE value decreased by 2.1 $Wm^{-2}$ for the AVHRR $R_n$ retrievals

**Comment 14:**

453 Change phrasing to "…especially clouds that have significant impacts on shortwave…"

**Response 14:**

Thanks for your suggestion. We have revised the phrase according to your comment.

205

> 485 clouds and CWV control surface R$_n$ dynamics under cloud-sky conditions,  especially clouds that have significant impacts on shortwave and longwave radiation. Therefore, AOD, CWV, and cloud optical thickness (COT, as a surrogate for cloud optical properties) derived from MERRA2 were employed to analyze the sensitivity of the accuracy of the AVHRR and GLASS R$_n$ retrievals to variations in

**Comment 15:**

454 Where do cloud optical thickness (COT) and cloud water vapor (CWV) data come from?

210 **Response 15:**

The COT and CWV data come from MERRA2 reanalysis. We have added the information in the revised manuscript.

> 485 clouds and CWV control surface R$_n$ dynamics under cloud-sky conditions,  especially clouds that have significant impacts on shortwave and longwave radiation. Therefore, AOD, CWV, and cloud optical thickness (COT, as a surrogate for cloud optical properties) derived from MERRA2 were employed to analyze the sensitivity of the accuracy of the AVHRR and GLASS R$_n$ retrievals to variations in

215

**Comment 16:**

462 The sentence beginning with "Therefore, the performance…" does not make sense. Perhaps the end of that sentence should read: "…is comparable with regard to the accuracy of their Rn retrievals."

**Response 16:**

220 Thanks for your kind suggestion. We have revised the phrase according to your comment.

> 495 of the RCNN model and the MARS models used for the GLASS R$_n$ product (Jiang et al., 2016) is comparable  with regard to the accuracy of their R$_n$ retrievals. However, when the absorption and scattering effects are

**Comment 17:**

225 512 – 519 I don't understand your Figure 11. The AVHRR and GLASS Rn's as a function of COT are nearly on top of each other, yet the bias plotted on the same charts is significant. What am I missing here? Regardless, in the caption please define the bias and what the shading represents.

**Response 17:**

The absolute bias is defined as the absolute difference between daily mean AVHRR and GLASS R$_n$,
230 i.e., $\left| Rn_{avhrr} - Rn_{glass} \right|$. To more easily understand, the bias is replaced by the difference. The shading represents the variation range (stand deviation) of global daily AVHRR and GLASS R$_n$ retrievals and their absolute differences. The related information has been added in the figure caption.

**Figure 11: Variations in the spatial and temporal consistency of AVHRR and GLASS daily $R_n$ retrievals against cloud optical thickness (COT) in (a) January and (b) July 2008. The absolute difference is defined as $\left|Rn_{avhrr} - Rn_{glass}\right|$. The shading represents the variation range (stand deviation) of global daily AVHRR and GLASS $R_n$ retrievals and their absolute differences.**

The distributions of the absolute differences between daily mean AVHRR and GLASS $R_n$ values in January and July 2008 are shown in Figure 1. There exist large absolute differences in both January and July. It is reasonable that large absolute differences occur in Figure 11 in the manuscript. The reason for the AVHRR and GLASS $R_n$ values being nearly on top of each other (solid lines) is the effect of averaging operation over multiple land pixels within a certain COT range. However, the large variation range (shading) of daily $R_n$ retrievals means a single pixel within a certain COT range may have a large absolute difference under some specific conditions.

[Figure]

Figure 1: The distributions of absolute differences in January (left) and July (right), respectively.

**Comment 18:**

524 Please state how the difference is defined. Jan.- July or July – Jan.

**Response 18:**

The difference of cloud fraction is defined as Jan.-July. We have included the information in the revised manuscript.

555    information includes the properties of the entire cloud layer. Figure S2 shows the spatial distribution of the monthly mean cloud cover fraction (CF) at the global scale in January and July, and the corresponding differences in CFs (Jan.-July). In

**Comment 19:**

525 Do you mean "northern Australia" and "South America" ?

**Response 19:**

Note the areas where the CFs in July is larger than that in January 2008 (blue areas, Figure 2). The areas include Central Africa, Southern Asia, Southern Australia and Antarctica. We have revised the

corresponding part in the manuscript.

January, the CFs are higher over most land regions except in  Central Africa, Southern Asia, Southern Australia and Antarctica. However, most regions had small CFs in July. The differences in CFs for the two months are also marked; the positive differences demonstrate that more than 72% of the land

560     pixels had a higher CFs in January than in July. The spatial adjacent effects induced by clouds are more significant on surface

[Figure]

260

Figure 2: Spatial distribution of the global CFs differences between in January and in July 2008.

**Comment 20:**

265     545 "produced by NOAA" ? Should this read "replaced by NOAA"?

**Response 20:**

Thanks for your careful work. We have revised the mistake in the revised manuscript.

alternative update times of the NOAA-series satellites. For example, NOAA-11 was successfully  succeeded by NOAA-14 from 1994 to 1995. Similarly, NOAA-16 replaced NOAA-14 in 2000 for monitoring of the Earth's surface and

585     atmosphere. During these periods of satellite replacement, the corresponding AVHRR data contain large gaps. For example,

270

**Comment 21:**

555-560 The four timeseries in Fig. 12 after 2017 for all data sets may be well correlated but are obviously wrong and could not be used for climate studies. What does the shading represent in Fig. 12?

**Response 21:**

275     The LTDR project only uses afternoon satellite to generate the long-term AVHRR dataset because the atmospheric correction algorithm would produce high uncertainty when applied to low sun elevation pixels from the morning (am) satellites. Afternoon satellites include NOAA7, NOAA9, NOAA11, NOAA14, NOAA16, NOAA18 and NOAA19 (Figure 3). The use of these satellites alone inevitably leads to small gaps in the data in exchange for a higher accuracy in the atmospheric correction. The time

280     series is not fully complete and presents some observational gaps. The most important two were found in 1994 and from 2018 onwards. In the first case, important gaps and noise were found in the images from March to September and empty data from September to December, due to NOAA11 orbital degradation. From 2018 onwards the data quality has been degrading due to important gaps in the images and the presence of artefacts (Otón et al., 2021). This is why the $R_n$ timeseries after 2017 seem to be

285     abnormal. At several studies, authors suggest that 1994, 2018, 2019, and 2020 are not used due to the poor quality of AVHRR data (Hansen et al., 2020; Tian et al., 2015). The corrsponding contents were included in the revised manuscript.

    The shading represents the variation range of global monthly mean $R_n$. The information has been included in the revised manuscript.

Note that the LTDR project only uses afternoon satellite to generate the AVHRR product to do with the high uncertainty of the atmospheric correction algorithm when applied to low sun elevation pixels present in morning (am) satellites. Afternoon satellites include NOAA-7, NOAA-9, NOAA-11, NOAA-14, NOAA-16, NOAA-18, NOAA-19, and NOAA-20. The use of

580     these satellites alone inevitably leads to small gaps in the data in exchange for a higher accuracy in the atmospheric correction. The time series is not fully complete and presents some observational gaps. Specifically, some large discrepancies occur,

290

Figure 12: Long-term temporal variation of (a) monthly average $R_n$ and (b) monthly $R_n$ anomalies for the AVHRR, CERES,  GLASS and MERRA2 datasets, respectively. The shading represents the variation range (stand deviation) of the global monthly mean $R_n$.

[Figure]

Figure 3: Local overpass time of all NOA satellites containing the AVHRR sensor. Figure obtained from
295     Clerbaux et al. (2020).

**Comment 22:**

557 Has the AVHRR calibration across all satellites been applied to the AVHRR data shown in Fig. 12?

300   **Response 22:**

The LTDR product performs geolocation, calibration, and atmospheric and surface anisotropy correction for all AVHRR sensors aboard the NOAA afternoon (pm) satellites (Vermote and Saleous, 2006; Vermote and Kaufman, 1995; Otón et al., 2021; Franch et al., 2017). The calibration method proposed by Vermote and Kaufman (1995) is applied consistently across the AVHRR instruments onboard various NOAA
305     satellites. Relevant information refers to

https://landweb.modaps.eosdis.nasa.gov/cgi-bin/ltdr/ltdr/ltdrPage.cgi?fileName=avhrr_calib.

**Comment 23:**

310    572 Do you mean 0.708?

**Response 23:**

Thanks for your carefule examination. I looked back at the statistics, and the best R is 0.708, not 7.08. I have revised the mistake in the manuscript.

610    the sub-images denoted as B3 … B19 vary from 3 × 3 to 19 × 19, respectively, with an interval of 2 pixels. The true areas correspond to approximately 15 × 15 km² (B3) to 135 × 135 km² (B19) on the ground. The results are shown in Fig. 13. Overall, the average R increases from 0.61 to 7.08 0.708, and RMSE decreases from 50.12 to 46.17, respectively, for the MLR model.

315

**Comment 24:**

591 What is a wide overpass time?

**Response 24:**

320    The wide overpass time refers to a broad range of local time for satellite crossing over a particular location as shown in Fig. 3 from 13:00 to 20:00 LT for afternoon satellites.

635    of the atmosphere and clouds. King et al. (2013) acknowledged that the frequency of cloud variations is high at different times and locations based on twin MODIS cloud products. In view of the wide satellite overpass times over a particular location, e.g., equatorial crossing time generally ranges from 1300 to 1730 in local time, representing different instantaneous

325    **Comment 25:**

600 What Study?

**Response 25:**

Shupe et al. (2011) found annual cloud occurrence fractions are 58%–83% at the Arctic observatories, with a clear annual cycle wherein clouds are least frequent in the winter and most frequent in the late
330    summer and autumn. We have included the study in the revised manuscript.

640    estimation at high latitudes than that at middle and low latitudes.  Shupe et al. (2011) found annual cloud occurrence fractions are 58%–83% at the Arctic

observatories, with a clear annual cycle wherein clouds are least frequent in the winter and most frequent in the late summer and autumn.

**Technical corrections:**

335    **Comment 26:**

164 You don't need ", respectively" in this sentence.

**Response 26:**

Thanks for your suggestion. I have deleted the ", respectively" in the revised manuscript.

AVHRR TOA observations at five spectral channels (a visible band (0.55–0.68 μm), a near-infrared band (0.75–1.1 μm), a middle-infrared band (3.55–3.93 μm), and two thermal bands (10.5–11.3 and 11.5–12.5 μm) were utilized for

180    their comprehensive surface and atmospheric electromagnetic information. The National Aeronautics and Space

340

**Comment 27:**

"BSRN_DRA" site.

**Response 27:**

345    Thanks for your suggestion. I have revised the "BSRA_DRA" to "BSRN_DRA".

455    $R_n$ retrievals are more consistent with in situ measurements than the CERES-SYN and MERRA2 products. Specifically, the MERRA2 and CERES-SYN $R_n$ retrievals show higher values compared to the in situ measurements at the  BSRN_DRA site, especially during 140–200 day period. In comparison, the AVHRR and GLASS $R_n$ values closely match the

**Comment 28:**

350    483 "very low" not "vary low"

**Response 28:**

Thanks for your suggestion. We have revised the "vary low" to "very low".

> 515   for the RCNN training. In addition, the AVHRR $R_n$ retrievals show steady and  very low (close to zero) biases under
>
>        different conditions, while the biases of the GLASS $R_n$ retrievals show a high degree of variation. This illustrates that the

**Comment 29:**

543 1999-2000

**Response 29:**

Thanks for your careful examination. We have corrected the mistake.

> during some periods including 1994–1995, 1999–2000, 2007–2008, and 2018–2019. These periods correspond to the
>
> alternative update times of the NOAA-series satellites. For example, NOAA-11 was successfully  succeeded by
>
> NOAA-14 from 1994 to 1995. Similarly, NOAA-16 replaced NOAA-14 in 2000 for monitoring of the Earth's surface and
>
> 585   atmosphere. During these periods of satellite replacement, the corresponding AVHRR data contain large gaps. For example,

**Response to referee #2**

Title: A global long-term (1981–2019) daily land surface radiation budget product from AVHRR satellite data using a residual convolutional neural network

MS_No: ESSD-2021-250

Thanks very much for taking your time to review this manuscript. We really appreciate all your valuable comments and constructive suggestions! The specific responses to your all comments are listed below one by one.

**Major comments:**

**Comment 1:**

One of the real advantages I see with this dataset is the long record—since the dataset starts in 1981 and has an accuracy equal to or exceeding other satellite-based estimates, this extends observation-based estimates of the surface radiation budget significantly. That could be of significant value for long-term climate studies. ***The authors could highlight this advantage more strongly in the abstract and conclusions.***

**Response 1:**

Thanks for your kind suggestion. We have highlighted the advantage of the long-term record of AVHRR $R_n$ dataset in the abstract and conclusions, and the related content has been included in the revised manuscript.

> 25 improved for the snow/ice cover surface. The spatiotemporal analyses of these four $R_n$ datasets indicate that the AVHRR $R_n$ product reasonably replicates the spatial pattern and temporal evolution trends of $R_n$ observations. The long-term record (1981-2019) of the AVHRR $R_n$ product shows its value in climate change studies. This dataset is freely available at

> global climate change. Besides, compared to current satellite-derived $R_n$ products, e.g., CERES-SYN and GLASS (2000-present), a more long record (1981-2019) of the AVHRR $R_n$ dataset shows its value in climate change studies. However,
> 730

**Comment 2:**

Evaluation and training are done against multiple networks, but some of these networks are interconnected, for example, some ARM and all SURFRAD sites are included in the BSRN. As I look at the list of sites in Table S1, it appears that some of these stations are included multiple times. For example, BSRN_DRA is the same station as SF_DRA because the SURFRAD Desert Rock stations is submitted to the BSRN global network. ***This is particularly a problem if any of the independent validation stations are also included in the training dataset. Please look into this duplication.***

**Response 2:**

Thanks for your careful examination. After thoroughly reviewing the list of training (460) and validation (77) sites based on sites' geographic coordinates (i.e, latitude, longitude, elevation), we found several duplicate sites in the individual training sites group, including one ARM (ARM_E13) site and six SURFRAD (SF_TBL, SF_DRA, SF_PSU, SF_SXF, SF_FPK, SF_GCM) sites have corresponding duplicated sites in the BSRN and FluxNet network in the training group. Besides, several sites from the AsiaFlux, including FxMt_GCK, FxMt_MSE, FxMt_QHB, FxMt_TMK, FxMt_TSE, FxMt_QHB, may be identical to the corresponding sites of the Global FluxNet. However, the same site from different observation networks has different time periods of record, e.g., BSRN_DRA (1998-2017) and SF_DRA (1999-2019). After these duplicated sites were removed from the training dataset, the training statistics were almost the same probably because the duplicated samples are relatively small compared to the total sample population. Therefore, we deleted theses duplicated sites in BSRN and FluxNet networks from Table S1 and Table 2, and the corresponding Figure 5(a) unchanged. The corresponding number of each network was also revised.

175 Monitoring for the Greenland Ice Sheet, SURFRAD: Surface Radiation Budget Network.

| Network/Program | Sensors | Temporal Interval | Number of sites | |
|---|---|---|---|---|
| ARM | Kipp&Zonen CNR-1 | 10 minutes | 33 | |
| AsiaFlux | Kipp&Zonen CNR-1/EKO MS201 | 30 minutes | 31 | |
| BSRN | Kipp&Zonen CG4/Eppley. PIR | 1 minutes | 15 | |
| CEOP | Eppley. PIR/EKO MS202 | 30 minutes | 16 | |
| CEOP-Int | Kipp&Zonen CG4/Eppley. PIR | 30 minutes | 8 | |
| ChinaFlux | Kipp&Zonen CNR-1 | 30/60 minutes | 3 | |
| EOL | Kipp&Zonen pyrgeometers, Eppley. PIR | 30/60 minutes | 17 | |
| GCNEET | Li Cor Photodiode & REBS Q*7 | 60 minutes | 18 | |
| GAME.ANN | EKO MS0202F | 30 minutes | 3 | |
| Global FluxNet | Kipp&Zonen CNR-1, etc | 30 minutes | 307 | |
| HiWATER | Kipp&Zonen CNR-1/CNR-4 | 10 minutes | 19 | |
| IMAU-Ktransect | Kipp&Zonen CNR-1 | 60 minutes | 4 | |
| LBA-ECO | Kipp&Zonen CG2/CNR-1 | 30 minutes | 8 | |
| PROMICE | Kipp&Zonen CNR-1/CNR-4 | 10 minutes | 24 | |
| SURFRAD | Eppley. Spectrosun | 1/3 minutes | 7 | |

In addition, we found out that there are three same sites in both the training and independent validation groups, as shown in Table 1, although they are nominally administrated by different observation networks. The three training sites of Lath_CN-Ha2, Lath_KR-Hnm, and Lath_ID-Pag are of the Global FluxNET. For corresponding three validation sites, the CF_HB site belongs to the ChinaFlux network; the FxMt_HFK and the FxMt_PDF sites are of the AsiaFlux network. The ChinaFlux and AsiaFlux networks are sub-network of the FluxNET project. Therefore, we believe that the respective three sites in the training group and the validation group are the same.

**Table 1** Summary of duplicate site in both training and validation sites groups.

| Training site name | Latitude (°) | Longitude (°) | Elevation (m) | Validation site name | Latitude (°) | Longitude (°) | Elevation (m) |
|---|---|---|---|---|---|---|---|
| Lath_CN-Ha2 | 37.6086 | 101.3269 | 3203 | CF_HB | 37.6099 | 101.3224 | 3205 |

| | | | | | | | |
|---|---|---|---|---|---|---|---|
| Lath_KR-Hnm | 34.55 | 126.57 | 7 | FxMt_HFK | 34.55 | 126.57 | 13.74 |
| Lath_ID-Pag | 2.345 | 114.036 | 30 | FxMt_PDF | 2.345 | 114.0364 | 30 |

445

Besides, we also found that several validation sites have extremely similar geographic coordinates to the training sites (Table 2). These sites are from the same observation network at local scale. These sites do not belong to the same site at both training and validation sites groups, e.g., the Lath_US-Tw1 in the training group and Lath_US-Tw1 (-2, -3) in the validation group. To deal with

450 the issue, we have adopted the method that the mean values from these sites' measurements within 5-km extent were used to match the grid data, as mentioned in section 3 (Line 224).

**Table 2** Summary of sites with the similar geographic coordinates in training and validation groups.

| Training site name | Latitude (°) | Longitude (°) | Elevation (m) | Validation site name | Latitude (°) | Longitude (°) | Elevation (m) |
|---|---|---|---|---|---|---|---|
| FGI_MET0002 | 67.361866 | 26.637728 | 179 | FGI_VUO0002 | 67.361883 | 26.643233 | 180 |
| HAWS17 | 38.8451 | 100.36972 | 1559.63 | HAWS16 | 38.84931 | 100.36411 | 1564.31 |
| Lath_CA-SCB | 61.3089 | -121.2984 | 280 | Lath_CA-SCC | 61.3079 | -121.2992 | 285 |
| Lath_US-Tw1 | 38.1074 | -121.6469 | -9 | Lath_US-Tw2 | 38.1047 | -121.6433 | -5 |
| Lath_US-Tw1 | 38.1074 | -121.6469 | -9 | Lath_US-Tw3 | 38.1159 | -121.6467 | -9 |
| Lath_US-Tw1 | 38.1074 | -121.6469 | -9 | Lath_US-Tw4 | 38.10298 | -121.6414 | -5 |
| IMAU-S10 | 67.0005 | -47.0167 | 1850 | PM-KAN_U | 67.0003 | -47.0253 | 1840 |

To keep the independence of validation dataset from the training samples, we removed duplicate
455 three sites of the CF_HB, the FxMt_HFK, and the FxMt_PDF from the validation group. Note that we only use measurements of the sites with ETC coefficient of more than 0.9 to weaken upscaling errors of ground-based measurements. The ETC coefficients of the CF_HB and the FxMt_PDF are 0.7492 and 0.0337, respectively. Measurements from the two sites were previously not used in the validation activity. Therefore, we only need to delete the FxMt_HFK site with an ETC coefficient

460    of 0.9225 from the validation group to evaluate the performance of the RCNN model again. Figure
       1 shows the evaluated result based on the independent validation sites without/with the FxMt_HFK
       site. The uncertainty of $R_n$ retrievals at validation sites changes slightly with RMSE values from
       26.66 $Wm^{-2}$ to 26.77$Wm^{-2}$.

       Therefore, previous independent evaluation of $R_n$ retrievals at validation sites is reliable although
465    duplicate three sites are used in training and validation dataset simultaneously. We have revised
       Figure 5(b) and the corresponding content in the revised manuscript.

[Figure]

**Figure 1:** Evaluated results of RCNN model using independent validation dataset (a) without
FxMt_HFK and (b) with FxMt_HFK site measurements, respectively.

470

       **Comment 3:**

       I am curious whether the results shown in Figure 7 reflect the fact that some of these networks are
       included in training the AVHRR dataset. It isn't clear to me from the description whether training
475    stations were also used in this analysis, or whether this only includes independent testing stations
       and stations that didn't meet the reliability requirements. But even if these validation stations are
       independent from training data, the network of measurements around the ARM Southern Great
       Plains sites, for example, may be more similar to each other than a site that is located in a much
       different climate regime (e.g. independent sites ARM_E06 and ARM_E41 sites). That could lead to
480    overfitting. ***It would be helpful to understand how independent this validation dataset is.***

       **Response 3:**

       The collected sites come from several local observation networks and international networks.
       Generally, sites of the local networks are located at the small region (e.g., ARM, HiWATER), while
       sites of the international networks are distributed over the globe (e.g., BSRN, FluxNet). To fully
485    utilize these networks, we follow the idea of determination of training and validation sites in the
       GLASS $R_n$ estimation algorithm (Jiang et al., 2016). For an observation network with multiple sites,
       we randomly selected several sites to serve as independent sites and remaining sites are used as
       training sites. Regarding to a local network with less sites, all sites are used as training sites to ensure

the representativeness of the training dataset in characterizing spatiotemporal variation of surface
490    $R_n$. Based on the strategy, the training and validation sites are finally determined. Therefore, the
training and validation dataset both have great representation that reflect different surfaces (land
cover types, elevations) and atmospheric properties (climate zone), which is important for
evaluating model's robustness. Figure 2 shows the proportional distribution of training and
validation sites under different conditions. This figure is also included in the supplementary to help
495    readers more deeply understand the representations of the training and test datasets.

[Figure]

**Figure 2:** Proportional distributions of (a-c) training sites and (d-f) validation sites under different
climates, elevation ranges, and land covers, respectively. The value in the brackets is total number
of sites under specific condition.

500    The result in Figure 7 is obtained only using the independent validation dataset with ETC
coefficients > 0.9 (reliable). We can see that AVHRR and GLASS $R_n$ retrievals have comparable
accuracies over most observation networks, and the overall validation result also illustrates that the
difference of uncertainty in these two $R_n$ datasets is small (< 1.63%). Specifically, the RMSE
differences between AVHRR and GLASS $R_n$ are -2.03 (ARM), -1.31 (BSRN), -1.34 (CEOP), -0.32
505    (CEOP-Int), 1.41 (EOL), -0.84 (AsiaFlux), -1.32 (FluxNet), 0.22 (PROMICE), and -0.99
(SURFRAD) $Wm^{-2}$, respectively. The performance of RCNN model over ARM network is better
than other networks as ARM is a local network with extremely similar conditions for training and
validation, which may reveal a false performance of RCNN model. However, some results from
BSRN, FluxNet, EOL networks can reflect more information about RCNN model robustness at a
510    larger spatiotemporal extent.

For the small regional network, measurements only reflect the spatiotemporal variation of $R_n$ at a
local extent. It is unsuitable to select many sites from local networks as validation sites to evaluate
RCNN's independent performance when we want to retrieve surface $R_n$ at global scale. Therefore,
more sites from the international networks should be used as the validation sites. Fortunately, the
515    number of site from the local networks is small in the validation group. Most validation sites were
used come from the networks at continental or global scales. Specifically, the number of sites from
the continental and global networks is more than 89%, including BSRN (2), CEOP (5), EOL (5),
AsiaFlux (10), FluxNet (39), and PROMICE (7). Conversely, the number of sites from local
networks is small with a proportion < 10%, including ARM (2), HiWATER (1), GAME.ANN (1).

520 Besides, based on the response 2, there is no duplicate site in training and validation site groups, except the CF_HB, the FxMt_HFK, and the FxMt_PDF. Therefore, the independence of validation dataset is adequate to evaluate the overall performance of RCNN model at validation sites.

> independent test datasets to evaluate the model performance. Similar and comprehensive surface and atmospheric conditions
>
> between training and validation sites illustrate the good representations of the training and test datasets (Fig. S1). More than
>
> 165 89% of validation sites come from the continental and international networks, including BSRN, FluxNet, CEOP, EOL, AsiFlux,
>
> PROMICE, which ensure the independence of the test dataset.

525

**Comment 4:**

Does Figure 14 show local time? Please label for clarity.

**Response 4:**

Thanks for your nice suggestion. Figure 14 shows the local time of NOAA-series satellites crossing.
530 We have added the information in the caption of figure 14 and the corresponding phrase.

> 635 times in local time. The improved effect is slightly more significant during the afternoon than in the morning when more over-
>
> land clouds are present (King et al., 2013). This improvement is also more pronounced during the night. The AVHRR $R_n$
>
> **Figure 14: Effect of daily mean MERRA2 $R_n$ on AVHRR $R_n$ retrievals at different satellite overpass times in local time. The bars indicate RMSE and lines indicates absolute biases.**

535 **Minor comments:**

**Comment 1:**

Line 50: "RT-based physical methods show a great generalization" I am not sure what this phrase means, please revise for clarity.

**Response 1:**

540 The phrase refers that different from empirical methods, the application of RT-based physical methods is not subjected to the limitation of training samples at a regional scale; in other words, the RT-based physical models are more applicable to a larger spatiotemporal extent. The phrase has been revised in the revised manuscript.

> empirical statistical methods.  RT-based physical methods are more
>
> applicable to a larger spatiotemporal extent because they consider the physical processes of solar radiation from the top of the

545

**Comment 2:**

Line 310: should it be: "data was *then* removed"?

**Response 2:**

550 Thanks for your careful examination. We have corrected the mistake in the revised manuscript.

> and divided into ten groups. One group of these data was  then removed as a hold-out or validation dataset and the
>
> remaining nine groups of data were treated as the training datasets. The training datasets were used to fit the RCNN model,

**Comment 3:**

555 Line 346: "for in surface radiation estimations." Wording doesn't seem quite right here.

**Response 3:**

Thanks for your careful work. We have revised the phrase in the revised manuscript.

> 375 identified as unreliability due to the presence of large water bodies within the satellite footprint. Thus, the processing of
>
> identifying reliable sites highlights the need to pay more attention to such areas for  surface radiation estimations.

560

**Comment 4:**

Lines 360-361: This sentence is awkwardly written and should be revised. Changing consistently to consistent, and site to sites would improve readability.

**Response 4:**

565 Thanks for your kind suggestion. We have revised the phrase according to your comments.

> 390 implementation of ETC for the selection of reliable site ensures more  consistent spatial representativeness of
>
> ground-based measurements and AVHRR data, which improves the accuracy of $R_n$ retrievals. Indeed, the CV-derived average

**Comment 5:**

570 Line 483: should be very instead of "vary"

**Response 5:**

Thanks for your careful examination. We have corrected the mistake.

> 515 for the RCNN training. In addition, the AVHRR $R_n$ retrievals show steady and  very low (close to zero) biases under
>
> different conditions, while the biases of the GLASS $R_n$ retrievals show a high degree of variation. This illustrates that the

575

**Comment 6:**

Line 545: should "produced" be replaced?

**Response 6:**

Thanks for your careful examination. We have revised the phrase.

alternative update times of the NOAA-series satellites. For example, NOAA-11 was successfully  succeeded by
NOAA-14 from 1994 to 1995. Similarly, NOAA-16 replaced NOAA-14 in 2000 for monitoring of the Earth's surface and
585    atmosphere. During these periods of satellite replacement, the corresponding AVHRR data contain large gaps. For example,

580

**Comment 7:**

Line 563: GLASS is misspelled GALSS

585    **Response 7:**

Thanks for your careful examination. We have corrected the misspelling.

**Figure 12: Long-term temporal variation of (a) monthly average $R_n$ and (b) monthly $R_n$ anomalies for the AVHRR, CERES,
GLASS and MERRA2 datasets, respectively. The shading represents the variation range (stand deviation) of the global monthly
mean $R_n$.**

590    **Comment 8:**

Line 572: I think that 7.08 must be 0.78. Please check.

**Response 8:**

Thanks for your careful examination. After looking back at the evaluated result, the R-value is 0.78, not 7.08. We have corrected the mistake.

the average R increases from 0.61 to  0.708, and RMSE decreases from 50.12 to 46.17, respectively, for the MLR model.

595    As the valid spatial extent increases, essential and complete spatial features are exposed and incorporated into the MLR model,

**Comment 9:**

Line 690: I think "satellite replacement works" should be satellite replacement work if you are
600    referring to times when there is no satellite data because it the satellites are being worked on.

**Response 9:**

Thanks for your kind suggestion. We have revised the phrase according to your comments.

740    Second, an effective data gap-filling method or multi-source data-fusion algorithm is required to fill the data gaps over land,

especially during periods of satellite replacement works. Third, coupled with spatially adjacent information, real-time temporal

**Comment 10:**

Line 697: should be "covered surfaces".

**Response 10:**

Thanks for your careful work. We have corrected the mistake.

740    covered surfaces. To address this problem, more physical knowledge is needed to fully utilize data-driven modeling to estimate

surface $R_n$ under different atmospheric and surface conditions. In particular, more attention should be paid to understanding

**Responses to Editor**

Title: A global long-term (1981–2019) daily land surface radiation budget product from AVHRR satellite data using a residual convolutional neural network
MS_No: ESSD-2021-250

Thank you for your careful work. I have included some guidelines in the revised manuscript according to your comments. The response is listed below.

**Comment 1:**
Reviewer #2 has given some very helpful and valuable suggestions on how and which reference data should be used for such applications. I hoped that you include some of these guidelines in your manuscript. Instead, you have only updated the numbers of the stations. Thus, please include the suggestions for higher-quality reference products and how to use them into your data section.

**Response 1:**
Reviewer #2 provided two useful suggestions in selecting in situ measurements. The first suggestion is that we should pay more attention to the duplicated sites when determining training and validation sites because some sites are included in multiple observation networks. The second suggestion is to ensure the representation and independence of the validation sites, which indeed shows the ability of model in estimating global surface Rn.

With respect to the two suggestions, we have updated the site measurements and corresponding results in the manuscript. Also, some useful suggestions according to Reviewer #2's comments are included in the section 2.1 in the revised manuscript to let other readers know the problems when conducting similar studies.

As shown in Fig. 1, the surface $R_n$ measurements from 448 stations were used to train the proposed RCNN model (red circles), while the measurements from the remaining 75 stations (blue circles) were selected as independent test datasets to evaluate the model performance. To well illustrate the performance of the model in estimating global surface $R_n$, more sites from international observation networks should be determined as the independent validation sites rather than regional observation networks with similar climate regimes (e.g., ARM) to ensure the independence of the test dataset, which avoids overfitting in model training. Similar and comprehensive surface and atmospheric conditions between training and validation sites illustrate the good representations of the training and test datasets (Fig. S1). In this study, More more than 89% of validation sites come from the continental and international networks, including BSRN, FluxNet, CEOP, EOL, AsiFlux, and PROMICE, which ensure the independence of the test dataset. Additionally, similar and comprehensive surface and atmospheric conditions

between the training and validation sites illustrate the good representations of the both training and independent test datasets in global surface $R_n$ variability (Fig. S1), which detects the ability of model in estimating global surface $R_n$. Note that some current regional and international networks are interconnected, for example, some ARM and all BSRN sites are included in the BSRN networks. When determining the training and validation sites, more attention should be paid to these duplicate sites in multiple observation networks to ensure the independence of the validation sites from the training sites. Finally, as shown in Fig. 1, the surface $R_n$ measurements from 448 stations were used to train the proposed RCNN model (red circles), while the measurements from the remaining 75 stations (blue circles) were selected as the independent test dataset to evaluate the model performance.

**References**

Augustine, J. A., DeLuisi, J. J., and Long, C. N.: SURFRAD–A national surface radiation budget network for atmospheric research, Bulletin of the American Meteorological Society, 81, 2341-2358, 2000.

Box, J. E. and Rinke, A.: Evaluation of Greenland ice sheet surface climate in the HIRHAM regional climate model using automatic weather station data, Journal of Climate, 16, 1302-1319, 2003.

Clerbaux, N., Akkermans, T., Baudrez, E., Velazquez Blazquez, A., Moutier, W., Moreels, J., and Aebi, C.: The Climate Monitoring SAF Outgoing Longwave Radiation from AVHRR, Remote Sensing, 12, 929, 2020.

El Saleous, N., Vermote, E., Justice, C., Townshend, J., Tucker, C., and Goward, S.: Improvements in the global biospheric record from the Advanced Very High Resolution Radiometer (AVHRR), International Journal of Remote Sensing, 21, 1251-1277, 2000.

Franch, B., Vermote, E. F., Roger, J.-C., Murphy, E., Becker-Reshef, I., Justice, C., Claverie, M., Nagol, J., Csiszar, I., Meyer, D., Baret, F., Masuoka, E., Wolfe, R., and Devadiga, S.: A 30+ Year AVHRR Land Surface Reflectance Climate Data Record and Its Application to Wheat Yield Monitoring, Remote Sensing, 9, 296, 2017.

Gelaro, R., McCarty, W., Suárez, M. J., Todling, R., Molod, A., Takacs, L., Randles, C. A., Darmenov, A., Bosilovich, M. G., and Reichle, R.: The modern-era retrospective analysis for research and applications, version 2 (MERRA-2), Journal of climate, 30, 5419-5454, 2017.

Hansen, M., Song, X., DiMiceli, C., Carroll, M., Sohlberg, R., Kim, D., and Townshend, J.: MEaSURES Vegetation Continuous Fields ESDR Algorithm Theoretical Basis Document (ATBD) Version 2.0, 2020.

He, K. and Sun, J.: Convolutional neural networks at constrained time cost, Proceedings of the IEEE conference on computer vision and pattern recognition, 5353-5360,

He, K., Zhang, X., Ren, S., and Sun, J.: Deep Residual Learning for Image Recognition, 2016 IEEE Conference on Computer Vision and Pattern Recognition (CVPR), 770-778, 10.1109/CVPR.2016.90, 2016.

Jiang, B., Liang, S., Ma, H., Zhang, X., Xiao, Z., Zhao, X., Jia, K., Yao, Y., and Jia, A.: GLASS daytime all-wave net radiation product: Algorithm development and preliminary validation, Remote Sensing, 8, 222, 2016.

Jiang, H., Lu, N., Qin, J., Tang, W., and Yao, L.: A deep learning algorithm to estimate hourly global solar radiation from geostationary satellite data, Renewable & Sustainable Energy Reviews, 114, 109327, 2019.

Liang, S., Cheng, J., Jia, K., Jiang, B., Liu, Q., Xiao, Z., Yao, Y., Yuan, W., Zhang, X., and Zhao, X.: The Global LAnd Surface Satellite (GLASS) product suite, Bulletin of the American Meteorological Society, 1-37, 2020.

Lu, N., Qin, J., Yang, K., and Sun, J.: A simple and efficient algorithm to estimate daily global solar radiation from geostationary satellite data, Energy, 36, 3179-3188, 2011.

Ohmura, A., Dutton, E. G., Forgan, B., Fröhlich, C., Gilgen, H., Hegner, H., Heimo, A., König-Langlo, G., McArthur, B., and Müller, G.: Baseline Surface Radiation Network (BSRN/WCRP): New precision radiometry for climate research, Bulletin of the American Meteorological Society, 79, 2115-2136, 1998.

Otón, G., Lizundia-Loiola, J., Pettinari, M. L., and Chuvieco, E.: Development of a consistent global long-term burned area product (1982–2018) based on AVHRR-LTDR data, International Journal of Applied Earth Observation and Geoinformation, 103, 102473, https://doi.org/10.1016/j.jag.2021.102473, 2021.

710 Pedelty, J., Devadiga, S., Masuoka, E., Brown, M., Pinzon, J., Tucker, C., Vermote, E., Prince, S., Nagol, J., Justice, C., Roy, D., Junchang, J., Schaaf, C., Jicheng, L., Privette, J., and Pinheiro, A.: Generating a Long-term Land Data Record from the AVHRR and MODIS Instruments, 2007 IEEE International Geoscience and Remote Sensing Symposium, 23-28 July 2007, 1021-1025, 10.1109/IGARSS.2007.4422974,

Philipona, R., Fröhlich, C., Dehne, K., DeLuisi, J., Augustine, J., Dutton, E., Nelson, D., Forgan, B., Novotny, P., and Hickey, J.: The Baseline Surface Radiation Network pyrgeometer round-robin
715 calibration experiment, Journal of Atmospheric and Oceanic Technology, 15, 687-696, 1998.

Shupe, M. D., Walden, V. P., Eloranta, E., Uttal, T., Campbell, J. R., Starkweather, S. M., and Shiobara, M.: Clouds at Arctic atmospheric observatories. Part I: Occurrence and macrophysical properties, Journal of Applied Meteorology and Climatology, 50, 626-644, 2011.

Srivastava, R. K., Greff, K., and Schmidhuber, J.: Highway networks, arXiv preprint arXiv:1505.00387,
720 2015.

Steffen, K. and Box, J.: Surface climatology of the Greenland ice sheet: Greenland Climate Network 1995–1999, Journal of Geophysical Research: Atmospheres, 106, 33951-33964, 2001.

Stokes, G. M. and Schwartz, S. E.: The Atmospheric Radiation Measurement (ARM) Program: Programmatic background and design of the cloud and radiation test bed, Bulletin of the American
725 Meteorological Society, 75, 1201-1222, 1994.

Tian, F., Fensholt, R., Verbesselt, J., Grogan, K., Horion, S., and Wang, Y.: Evaluating temporal consistency of long-term global NDVI datasets for trend analysis, Remote Sensing of Environment, 163, 326-340, 2015.

Vermote, E. and Kaufman, Y.: Absolute calibration of AVHRR visible and near-infrared channels using
730 ocean and cloud views, International Journal of Remote Sensing, 16, 2317-2340, 1995.

Vermote, E. and Saleous, N.: Calibration of NOAA16 AVHRR over a desert site using MODIS data, Remote sensing of Environment, 105, 214-220, 2006.

Wang, K. and Dickinson, R. E.: Global atmospheric downward longwave radiation at the surface from ground-based observations, satellite retrievals, and reanalyses, Reviews of Geophysics, 51, 150-185,
735 2013.

Wang, T., Shi, J., Husi, L., Zhao, T., Ji, D., Xiong, C., and Gao, B.: Effect of solar-cloud-satellite geometry on land surface shortwave radiation derived from remotely sensed data, Remote Sensing, 9, 690, 2017.

Wilson, K., Goldstein, A., Falge, E., Aubinet, M., Baldocchi, D., Berbigier, P., Bernhofer, C., Ceulemans,
740 R., Dolman, H., and Field, C.: Energy balance closure at FLUXNET sites, Agricultural and Forest Meteorology, 113, 223-243, 2002.

Wyser, K., O'Hirok, W., Gautier, C., and Jones, C.: Remote sensing of surface solar irradiance with corrections for 3-D cloud effects, Remote Sensing of Environment, 80, 272-284, https://doi.org/10.1016/S0034-4257(01)00309-1, 2002.

745 Xiao, Z., Liang, S., Tian, X., Jia, K., Yao, Y., and Jiang, B.: Reconstruction of long-term temporally continuous NDVI and surface reflectance from AVHRR data, IEEE Journal of Selected Topics in Applied Earth Observations and Remote Sensing, 10, 5551-5568, 2017.

Yamamoto, S., Saigusa, N., Gamo, M., Fujinuma, Y., Inoue, G., and Hirano, T.: Findings through the AsiaFlux network and a view toward the future, Journal of Geographical Sciences, 15, 142-148, 2005.

750